# Global variability in atmospheric new particle formation mechanisms

Bin Zhao[1,2,3 ✉], Neil M. Donahue[4,5,6,7], Kai Zhang[3], Lizhuo Mao[1], Manish Shrivastava[3], Po-Lun Ma[3], Jiewen Shen[1], Shuxiao Wang[1,2], Jian Sun[8], Hamish Gordon[4,5], Shuaiqi Tang[3], Jerome Fast[3], Mingyi Wang[9], Yang Gao[10], Chao Yan[11], Balwinder Singh[3], Zeqi Li[1], Lyuyin Huang[1], Sijia Lou[11], Guangxing Lin[3,13], Hailong Wang[3], Jingkun Jiang[1,2], Aijun Ding[11], Wei Nie[11], Ximeng Qi[11], Xuguang Chi[11] & Lin Wang[12]

A key challenge in aerosol pollution studies and climate change assessment is to understand how atmospheric aerosol particles are initially formed[1,2]. Although new particle formation (NPF) mechanisms have been described at specific sites[3–6], in most regions, such mechanisms remain uncertain to a large extent because of the limited ability of atmospheric models to simulate critical NPF processes[1,7]. Here we synthesize molecular-level experiments to develop comprehensive representations of 11 NPF mechanisms and the complex chemical transformation of precursor gases in a fully coupled global climate model. Combined simulations and observations show that the dominant NPF mechanisms are distinct worldwide and vary with region and altitude. Previously neglected or underrepresented mechanisms involving organics, amines, iodine oxoacids and $HNO_3$ probably dominate NPF in most regions with high concentrations of aerosols or large aerosol radiative forcing; such regions include oceanic and human-polluted continental boundary layers, as well as the upper troposphere over rainforests and Asian monsoon regions. These underrepresented mechanisms also play notable roles in other areas, such as the upper troposphere of the Pacific and Atlantic oceans. Accordingly, NPF accounts for different fractions (10–80%) of the nuclei on which cloud forms at 0.5% supersaturation over various regions in the lower troposphere. The comprehensive simulation of global NPF mechanisms can help improve estimation and source attribution of the climate effects of aerosols.

Atmospheric aerosol particles cause more than three million premature deaths worldwide every year[8] and act as a key modulator of Earth's climate. NPF from condensable gas molecules is the fundamental source of most atmospheric particles[1,9]. The subsequent growth of these particles is thought to contribute approximately half of the global number of cloud condensation nuclei (CCN)[9,10], substantially affecting cloud properties and Earth's radiative balance[1,11]. Understanding the mechanisms of regional and global NPF is necessary for accurate estimation of aerosols' climatic effects and for attribution of such effects to controllable sources of primary particles and gases.

Despite its atmospheric importance, NPF has long been among the least understood components of atmospheric chemistry. Recent observational studies have revealed NPF mechanisms at specific sites through direct detection of molecular clusters (intermediates for particle formation)[3–6]. However, NPF mechanisms in most regions and at most altitudes remain a mystery. This is largely because current

atmospheric models—tools indispensable for understanding the mechanisms and impacts of NPF on global and regional scales—lack the ability to represent many critically important processes. Most widely used global models are built on traditional binary and ternary particle nucleation processes involving sulfuric acid ($H_2SO_4$), ammonia ($NH_3$) and ions[1,10], which underpredict both the NPF rate and the particle numbers in most atmospheric environments, often by one order of magnitude or more[3,7,12,13]. Some recent experimental and/or observational studies suggested that 'modern' NPF mechanisms, including amine–$H_2SO_4$ nucleation[3,4,14], synergistic $HNO_3$–$H_2SO_4$–$NH_3$ nucleation[15] and iodine oxoacids nucleation[5,16,17], are important at certain locations[3–5,15–17], but such mechanisms have seldom, if ever, been incorporated in atmospheric models[15,18,19]. Another complicated mechanism is organic-mediated nucleation, which is driven by ultralow and extremely low volatility organic compounds (ULVOCs and ELVOCs, respectively) with saturation vapour concentration ($C^*$) of less than

[1]State Key Joint Laboratory of Environmental Simulation and Pollution Control, School of Environment, Tsinghua University, Beijing, China. [2]State Environmental Protection Key Laboratory of Sources and Control of Air Pollution Complex, Beijing, China. [3]Pacific Northwest National Laboratory, Richland, WA, USA. [4]Center for Atmospheric Particle Studies, Carnegie Mellon University, Pittsburgh, PA, USA. [5]Department of Chemical Engineering, Carnegie Mellon University, Pittsburgh, PA, USA. [6]Department of Chemistry, Carnegie Mellon University, Pittsburgh, PA, USA. [7]Department of Engineering and Public Policy, Carnegie Mellon University, Pittsburgh, PA, USA. [8]National Center for Atmospheric Research, Boulder, CO, USA. [9]Division of Chemistry and Chemical Engineering, California Institute of Technology, Pasadena, CA, USA. [10]Key Laboratory of Marine Environment and Ecology, Ministry of Education, Ocean University of China, Qingdao, China. [11]Joint International Research Laboratory of Atmospheric and Earth System Sciences, School of Atmospheric Sciences, Nanjing University, Nanjing, China. [12]Shanghai Key Laboratory of Atmospheric Particle Pollution and Prevention (LAP³), Department of Environmental Science and Engineering, Fudan University, Shanghai, China. [13]Present address: College of Ocean and Earth Sciences, Xiamen University, Xiamen, China. ✉e-mail: bzhao@mail.tsinghua.edu.cn

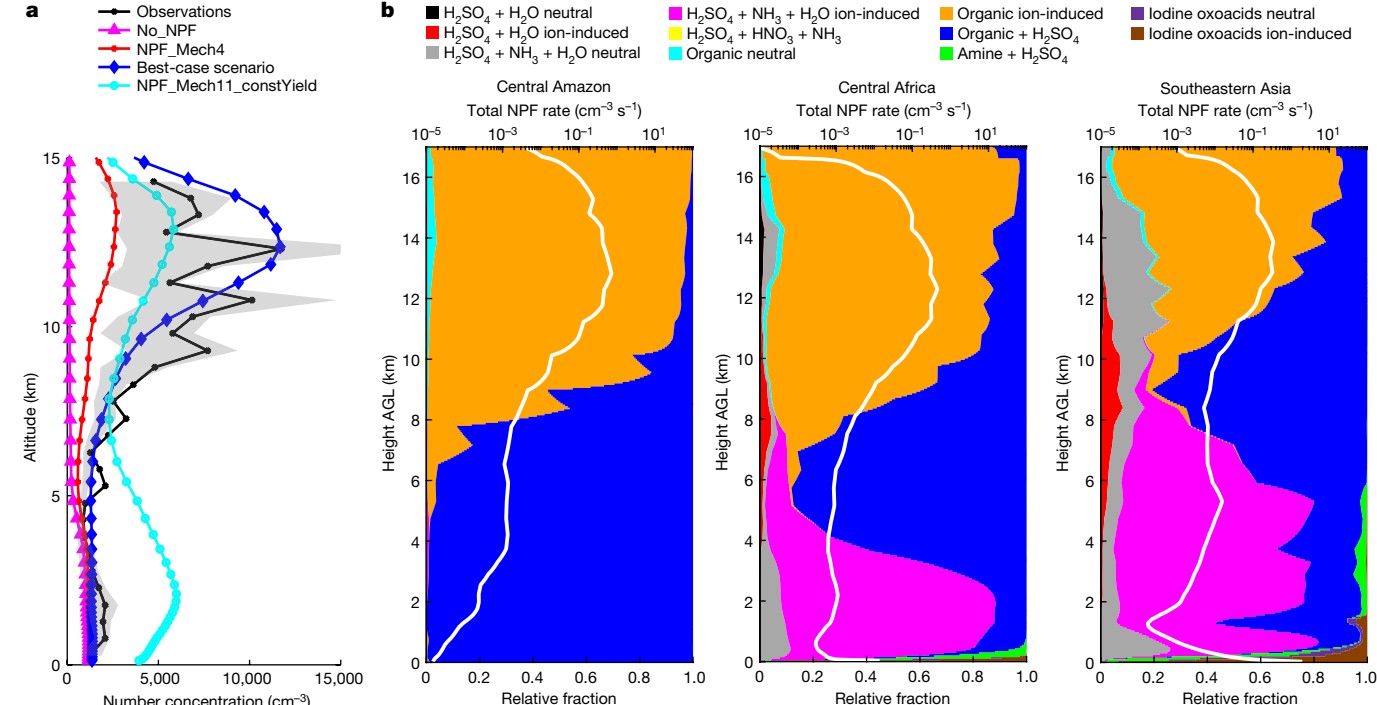

**Fig. 1 | Mechanisms of NPF and constraints from observations over rainforests. a**, Comparison of simulated particle number concentrations with aircraft measurements obtained over the Amazon during the ACRIDICON-CHUVA campaign in September 2014. Both simulations and observations are for particles >10 nm near the surface and 20 nm above the altitude of 13.8 km, with smooth transition between. The lines represent mean concentrations within each vertical bin and the shaded areas represent the 25th to 75th percentiles of the observations. All particle number concentrations are normalized to standard temperature and pressure (273.15 K and 101.325 kPa). Definitions of the model scenarios are given in the main text and Supplementary Table 1. **b**, NPF rates as a function of height AGL over the Central Amazon, Central Africa and Southeastern Asia. White lines represent the total NPF rates of all mechanisms at a diameter of 1.7 nm ($J_{1.7}$, on a log scale) and the coloured areas represent the relative contributions of different mechanisms, both averaged in 2016 over the regions specified in Extended Data Fig. 1b.

$3 \times 10^{-9}$ and $3 \times 10^{-5}$ μg m$^{-3}$, respectively[20,21]. Recent modelling studies that considered organic-mediated nucleation consistently oversimplified the processes by assuming that the nucleating organics represent a fixed fraction of all oxidation products[9,22–24] or that only a few individual molecules are involved in nucleation[25]. This is by contrast to the latest understanding that ULVOCs and ELVOCs encompass numerous species[20,26,27] and their yields vary by several orders of magnitude depending on temperature and NO$_x$ concentration[26,28,29]. The wide gap in model representation of NPF processes prevents holistic understanding of the mechanisms and impacts of NPF globally.

Here we synthesized molecular-level laboratory experiments to develop comprehensive model representations of NPF and the chemical transformation of precursor gases in a fully coupled global climate model. The model considers 11 nucleation mechanisms, among which four crucial mechanisms were largely overlooked previously, including iodine oxoacids neutral and ion-induced nucleation, synergistic H$_2$SO$_4$–HNO$_3$–NH$_3$ nucleation and amine–H$_2$SO$_4$ nucleation. Furthermore, we transformed previous model representations of pure-organic and organic–H$_2$SO$_4$ nucleation by implementing in the model an advanced experimentally constrained Radical Two-Dimensional Volatility Basis Set (R2D-VBS) to simulate the temperature-dependent and NO$_x$-dependent formation chemistry and thermodynamics of ULVOCs and ELVOCs[13,26]. Our new model greatly improves particle number simulations over the world's particle hotspots, often by one order of magnitude or more. This allows the explanation of worldwide NPF mechanisms that vary greatly with region and altitude (see schematic in Extended Data Fig. 1a). Below, we first discuss the NPF mechanisms over rainforests, anthropogenically polluted regions and oceans (see their respective spatial extent in Extended Data Fig. 1b), which cover most of the world's areas with either high particle concentrations or

large aerosol–cloud radiative forcing[7,30–36]. We then provide a global overview of the mechanisms and impacts of NPF. We show that previously underrepresented mechanisms probably dominate NPF over most of the above key regions, which could substantially reshape the understanding of NPF. Our systematic sensitivity analysis suggests that the quantifiable uncertainties at present are unlikely to change our main findings but might affect the exact quantitative contributions of individual mechanisms. Furthermore, potential uncertainties beyond our current knowledge might further refine and possibly modify the findings we present.

## NPF mechanisms over rainforests

The upper troposphere above rainforests, including the Central Amazon, Central Africa and Southeastern Asia, is among the largest reservoirs of particles globally on a number basis, as shown in Extended Data Fig. 2a. Figure 1a compares simulated vertical profiles of particles with those obtained from aircraft measurements over the Central Amazon during the ACRIDICON-CHUVA campaign[30]. As well as our comprehensive 'best-case' scenario that included all 11 nucleation mechanisms and the R2D-VBS module, we conducted two sensitivity simulations: the first did not consider any NPF process ('No_NPF'), whereas the second considered only traditional neutral and ion-induced H$_2$SO$_4$–H$_2$O nucleation and H$_2$SO$_4$–NH$_3$–H$_2$O nucleation ('NPF_Mech4'), which resembles the NPF treatment adopted in commonly used climate models[1,10]. Simulation results from the two sensitivity experiments underestimate the high particle number concentrations in the upper troposphere by nearly one order of magnitude. By contrast, our best-case scenario successfully reproduces the observed strong peak in particle number concentrations in the upper troposphere.

The much-improved performance and global coverage of our model allow explanation of the NPF mechanisms over the three main rainforest areas mentioned above, as shown in Fig. 1b. Organic-mediated nucleation (pure-organic and organic–$H_2SO_4$ nucleation) consistently dominates in the upper troposphere of the three regions according to our model. In particular, pure-organic ion-induced nucleation dominates at altitudes above 10–12 km, at which the largest NPF rates occur. Organic-mediated nucleation mechanisms are mainly driven, in our model, by ULVOCs and ELVOCs formed through oxidation of monoterpene emissions from rainforests; these monoterpene emissions are lifted to the upper troposphere by frequent and strong tropical convection. NPF rates are greatly enhanced in the upper troposphere relative to those in the lower troposphere because the low temperatures of the former increase the overall yields of ULVOCs/ELVOCs as a combined effect of chemistry and volatility changes and enhance the stability of new particles. Our new R2D-VBS module captures the variable yields of ULVOCs/ELVOCs, representing a notable advantage over previous simplified modelling approaches[9,22,37] that assume that a fixed fraction of the organic oxidation products drives nucleation. A sensitivity simulation ('NPF_Mech11_constYield'; see Supplementary Table 1) shows that such simplified approaches would produce too many particles at low altitude and too few particles at high altitude, unlike the distribution revealed by observations over the Amazon. Furthermore, the simulated concentrations of monoterpenes, which are precursors to the ULVOCs/ELVOCs, generally agree with observations acquired during two field campaigns over the Amazon[38] and Southeastern Asia[39] (0.19 ppb versus 0.13 ppb in the Amazon and 0.13 ppb versus 0.17 ppb in Southeastern Asia), further enhancing confidence in the model simulations.

## NPF mechanisms in human-polluted regions

Another group of global particle hotspots is the boundary layer of anthropogenically polluted regions, such as Eastern China, India and parts of Europe and the United States, as well as the upper troposphere in the Asian monsoon region that spans Eastern China and India[15,31,32]. Recent observations detected abundant amine–$H_2SO_4$ clusters during NPF events and suggested the dominant role of amine–$H_2SO_4$ nucleation at several surface sites in Beijing and Shanghai[3,4]; however, NPF mechanisms on the regional scale and their vertical distributions remain to be explored. Figure 2a evaluates simulated particle number size distributions against available observations at three sites in China (see statistics in Extended Data Fig. 3d). Simulations without NPF ('No_NPF') or with only traditional inorganic nucleation mechanisms ('NPF_Mech4') underestimate the number concentrations of ultrafine particles (diameter < 100 nm) by more than 60%. Our best-case scenario with all 11 mechanisms successfully reproduces the observed ultrafine particle concentrations with a normalized mean bias within ±10% (within ±33% for individual sites).

On the basis of the markedly improved model, Fig. 2b illustrates NPF mechanisms as a function of height over the four polluted regions mentioned above. The NPF rates in these regions are highest near the surface and mainly driven by amine–$H_2SO_4$ nucleation in our model, which is ultimately attributed to strong anthropogenic amine and $SO_2$ emissions. Notably, amine–$H_2SO_4$ nucleation is confined to a reasonably shallow layer below about 500 m, which is primarily attributable to sharp reduction in concentrations of amines with height owing to their short lifetime. To ensure that the model correctly captures amine–$H_2SO_4$ nucleation, we compared simulated concentrations of $H_2SO_4$ and dimethylamine (DMA) with observations acquired in China, Europe and the United States (Extended Data Fig. 3a,b). The simulated concentrations are generally within a factor of 3 (2.5) of the observed values for $H_2SO_4$ (DMA), indicating reasonable model performance, especially considering the difficulty in accurate simulation of these species recognized in previous studies[40,41].

A secondary maximum of NPF rate occurs in the upper troposphere (9–14 km) in all four regions. Over Eastern China and India, which have relatively high upper-tropospheric NPF rates among the four regions, the recently discovered synergistic $H_2SO_4$–$HNO_3$–$NH_3$ mechanism[15] is important and often dominant, as shown in Fig. 2b, whereas $H_2SO_4$–$NH_3$–$H_2O$ neutral nucleation plays a secondary but sometimes comparable role. The rate of $H_2SO_4$–$HNO_3$–$NH_3$ nucleation is highly sensitive to the abundance of ammonia, which is lifted to the upper troposphere by vigorous Asian monsoon convection[42,43]. The simulated $NH_3$ concentration in the Asian monsoon upper troposphere during summer mostly varies in the range 10–40 ppt and occasionally reaches 60 ppt (Extended Data Fig. 4), consistent with large-scale satellite observations (mostly 10–35 ppt, occasionally 150 ppt)[42,43]. Notably, the actual rate of $H_2SO_4$–$HNO_3$–$NH_3$ nucleation is probably higher than our baseline simulation because the real-world $NH_3$ concentration is non-uniform within a 1° × 1° model grid (see related sensitivity simulations in Methods). Hence, $H_2SO_4$–$HNO_3$–$NH_3$ nucleation is probably the leading mechanism in the upper troposphere above Eastern China and India. In Europe and the Eastern United States, the upper-tropospheric NPF rates are comparatively smaller; the dominant mechanism in our model is organic-mediated nucleation (organic–$H_2SO_4$ and pure-organic nucleation; Fig. 2b), although $H_2SO_4$–$NH_3$–$H_2O$ neutral nucleation may also make certain contributions. Compared with rainforest regions, the dominant role of organic-mediated nucleation is less pronounced, introducing a certain level of uncertainty. The mechanisms in Europe and the Eastern United States differ from those over Eastern China and India because: (1) the weaker convection in Europe and the Eastern United States limits convective transport of $NH_3$ and (2) the stronger emission of biogenic volatile organic compounds is favourable for organic-mediated nucleation.

## NPF mechanisms over the oceans

Among all regions globally, cloud radiative effects are most susceptible to CCN availability over the oceans[44,45]. Moreover, the tropical oceanic upper troposphere, as well as fragmented areas in the oceanic boundary layer, have been shown to be particle hotspots by the Atmospheric Tomography Mission (ATom) campaign[7,46] carried out over the Pacific and Atlantic oceans, but the mechanisms of NPF remain unclear. Figure 3a shows that, without considering NPF ('No_NPF'), the model captures the observed coarse-mode concentrations but largely misses the nucleation-mode and Aitken-mode particles. By contrast, our best-case simulation with 11 NPF mechanisms reasonably reproduces the observed distributions of nucleation-mode and Aitken-mode particles, especially the extensive upper-tropospheric hotspots and scattered boundary-layer hotspots.

With reasonable model–observation agreement, the NPF mechanisms over the Pacific and Atlantic oceans are shown in Fig. 3b and Extended Data Fig. 8. In the marine boundary layer, the dominant NPF mechanism in our model is mostly neutral and ion-induced nucleation of iodine oxoacids, which is directly associated with oceanic emissions of iodine-containing species. Ion-induced nucleation is more important in the tropics, whereas neutral nucleation is more important in colder mid-to-high-latitude regions because of its stronger temperature dependence. To ensure that the model reasonably simulates iodine oxoacids nucleation, we further show that the simulated $HIO_3$ concentrations vary between 80% below and 100% above the observed values at ten oceanic or coastal sites worldwide (Extended Data Fig. 3c). This is deemed reasonable performance because our study, to our knowledge, represents the first time that $HIO_3$ formation chemistry has been simulated in three-dimensional models.

In the upper troposphere above the Pacific and Atlantic oceans, organic–$H_2SO_4$ nucleation and $H_2SO_4$–$NH_3$–$H_2O$ neutral nucleation are most probably the two dominant mechanisms of nucleation according to our model. The relative importance of the two mechanisms varies

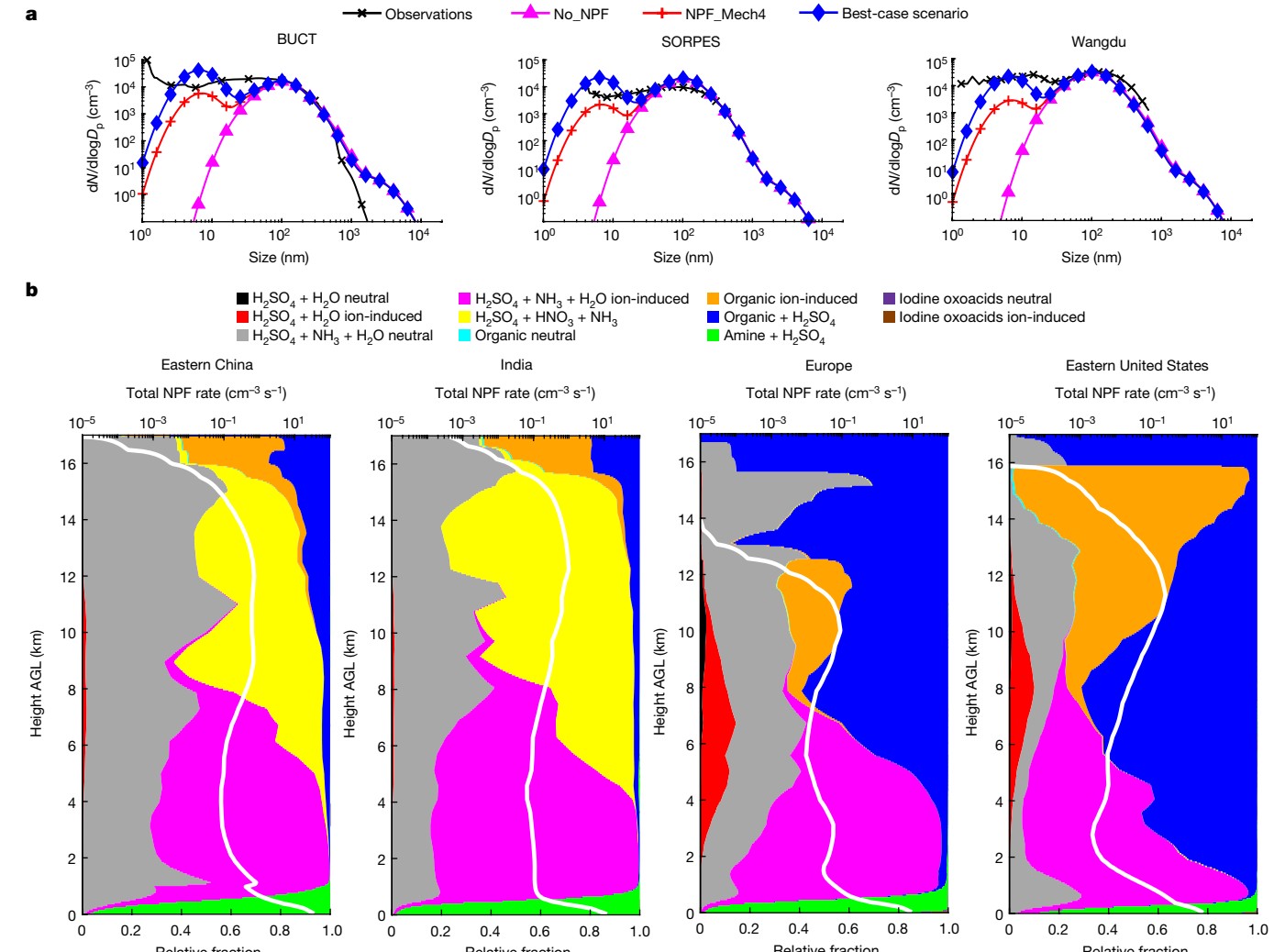

**Fig. 2 | Mechanisms of NPF and constraints from observations over anthropogenically polluted regions. a**, Comparison of simulated particle number size distributions with observations obtained at three sites in China: BUCT (Beijing University of Chemical Technology), Beijing; SORPES (Station for Observing Regional Processes of the Earth System), Nanjing; Wangdu, Hebei. All particle number size distributions are normalized to standard temperature and pressure. Definitions of the model scenarios are given in the main text and Supplementary Table 1. We do not expect the model to exactly capture the shape of the ultrafine number size distribution because the model uses a mode approach to represent particle size. **b**, NPF rates as a function of height AGL over Eastern China, India, Europe and the Eastern United States. White lines represent the total NPF rates at a diameter of 1.7 nm ($J_{1.7}$, on a log scale) and the coloured areas represent the relative contributions of different mechanisms, both averaged in 2016 over the regions specified in Extended Data Fig. 1b.

with latitude. Our simulation suggests that organic–$H_2SO_4$ nucleation is important across wide latitude ranges, whereas $H_2SO_4$–$NH_3$–$H_2O$ neutral nucleation could be important in certain mid-latitude areas with obvious anthropogenic influences. Notably, the $H_2SO_4$ concentration involved in nucleation traces back to both oceanic dimethyl sulfide (DMS) emissions and continental anthropogenic $SO_2$ emissions (Supplementary Fig. 1).

## Global overview and sensitivity analysis

In Fig. 4, which illustrates the global zonal mean NPF rates of each of the 11 mechanisms, global NPF mechanisms are shown to be largely governed by the hotspot regions analysed above. In the tropical upper troposphere, particle formation is dominated by organic-mediated nucleation in our model, especially pure-organic ion-induced nucleation that dominates above the altitude of 11 km, at which the largest NPF rates occur. This reflects the NPF mechanisms over the rainforests and tropical oceans. In the mid-latitude upper troposphere, the

$H_2SO_4$–$HNO_3$–$NH_3$, organic–$H_2SO_4$ and $H_2SO_4$–$NH_3$–$H_2O$ mechanisms all make important contributions to particle formation, reflecting a combination of NPF mechanisms over the mid-latitude oceans and Asian monsoon regions. In the boundary layer, amine–$H_2SO_4$ nucleation dominates in mid-latitude areas of the Northern Hemisphere in our model, characteristic of anthropogenically polluted regions; iodine oxoacids nucleation dominates at other latitudes because of the vast oceans.

We designed sensitivity experiments to test key factors of uncertainty that might potentially affect the leading NPF mechanism in one or more regions. Detailed descriptions and results of the sensitivity experiments are provided in Methods and summarized in Supplementary Table 1, Extended Data Figs. 5–9 and Supplementary Figs. 2–9. Briefly, the sensitivity experiments consisted of changing emissions or concentrations of $SO_2$, DMS, $H_2SO_4$, monoterpenes, DMA, $NH_3$ and $HIO_3$, the temperature dependence of organic-mediated nucleation, the parameterizations of organic–$H_2SO_4$ and amine–$H_2SO_4$ nucleation, the subgrid $NH_3$ concentration distributions and key parameters

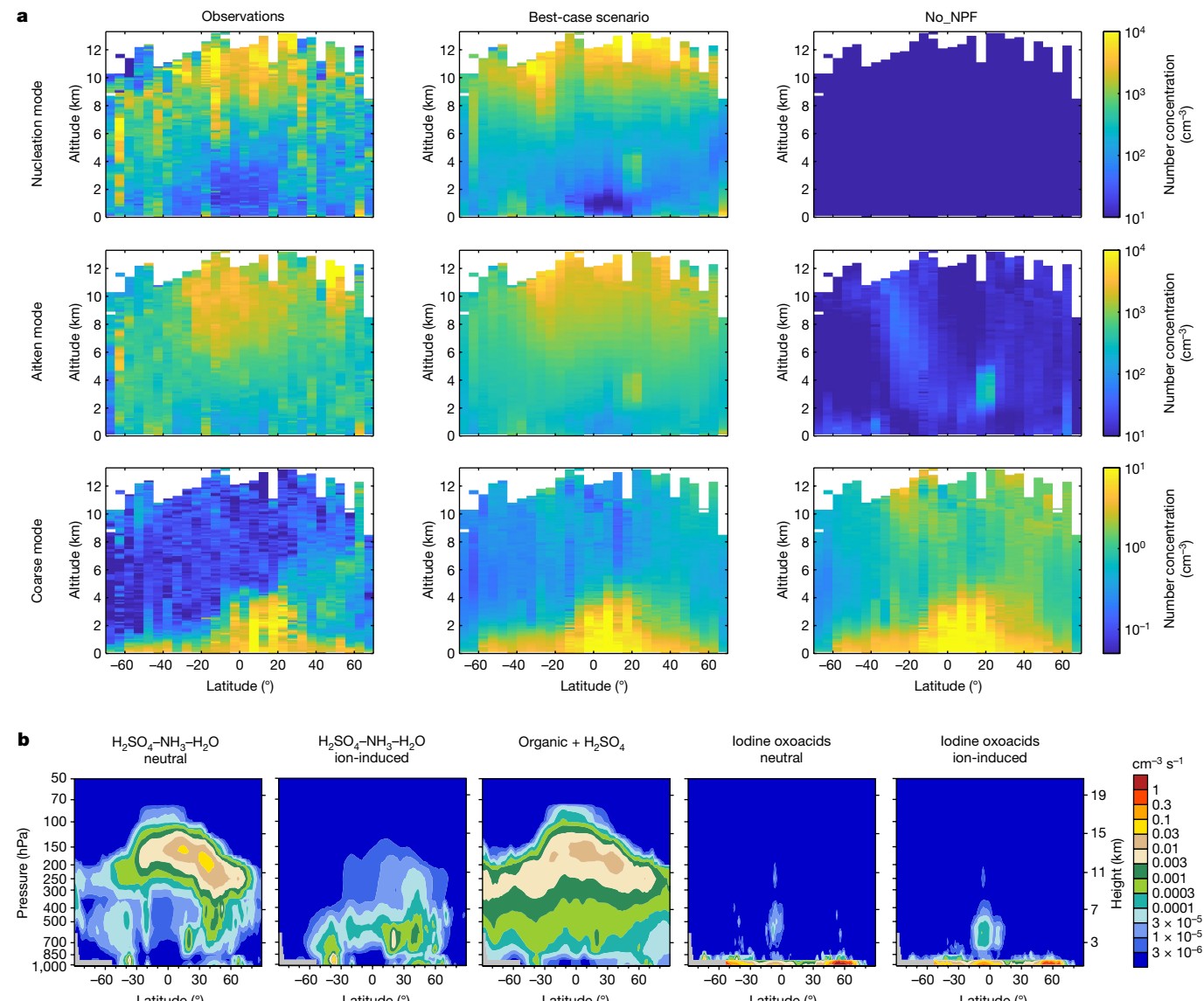

**Fig. 3 | Mechanisms of NPF and constraints from observations over oceans.**
**a**, Comparison of simulated number concentrations of nucleation-mode, Aitken-mode and coarse-mode particles with aircraft observations obtained over the Pacific and Atlantic oceans. The observations were obtained during the ATom campaign in July–August 2016, January–February 2017, September–October 2017 and April–May 2018. Simulation results are matched to individual observational data based on time and location. Model–observation pairs are grouped into two-dimensional bins defined by latitude (every 5°) and altitude

(every 100 m) and the average particle number concentrations in each bin are calculated and plotted. All particle number concentrations are normalized to standard temperature and pressure. **b**, Zonal mean NPF rates of individual mechanisms over the Pacific Ocean (170° E–150° W) in 2016. Only five NPF mechanisms are shown because the other six mechanisms are negligible in these regions. NPF rates over the Atlantic Ocean are shown in Extended Data Fig. 8 (first row).

governing convective transport based on our best-case scenario. The results indicate that our main findings about the leading NPF mechanisms hold true under these sensitivity simulations in all key regions of interest. Nevertheless, these sources of uncertainty might affect the precise quantitative contributions of individual mechanisms, both in the above key regions and in other areas not discussed in detail individually.

## Contribution of NPF to particles and CCN

On the basis of our new model, we show in Fig. 5 and Extended Data Fig. 2 the fractions of particles and CCN at 0.5% supersaturation (CCN0.5%) caused by NPF at different heights, as determined by the contrast between the 'best-case' and 'No_NPF' scenarios. In the lower troposphere (from the surface to 1 km, approximately the low-cloud

level), the fractions of particles and CCN0.5% caused by NPF vary regionally. Over the tropical and mid-latitude oceans at which cloud radiative effects are highly susceptible to CCN availability, NPF generally accounts for 70–90% of the particles and for 50–80% of the CCN0.5% concentration. For anthropogenically polluted regions, the fractions caused by NPF are comparatively large over Europe and the Eastern United States (80–95% of particle numbers and 30–65% of the CCN0.5% concentration) and small over Eastern China and India (35–85% of particle numbers and 10–35% of the CCN0.5% concentration). The difference is probably because China and India have relatively large emissions of primary particles that act as a strong coagulation sink of freshly formed particles. For rainforests, the fractions of particles and CCN0.5% from NPF also vary greatly with location, depending inversely on the strength of biomass-burning emissions. More discussion is provided in Methods.

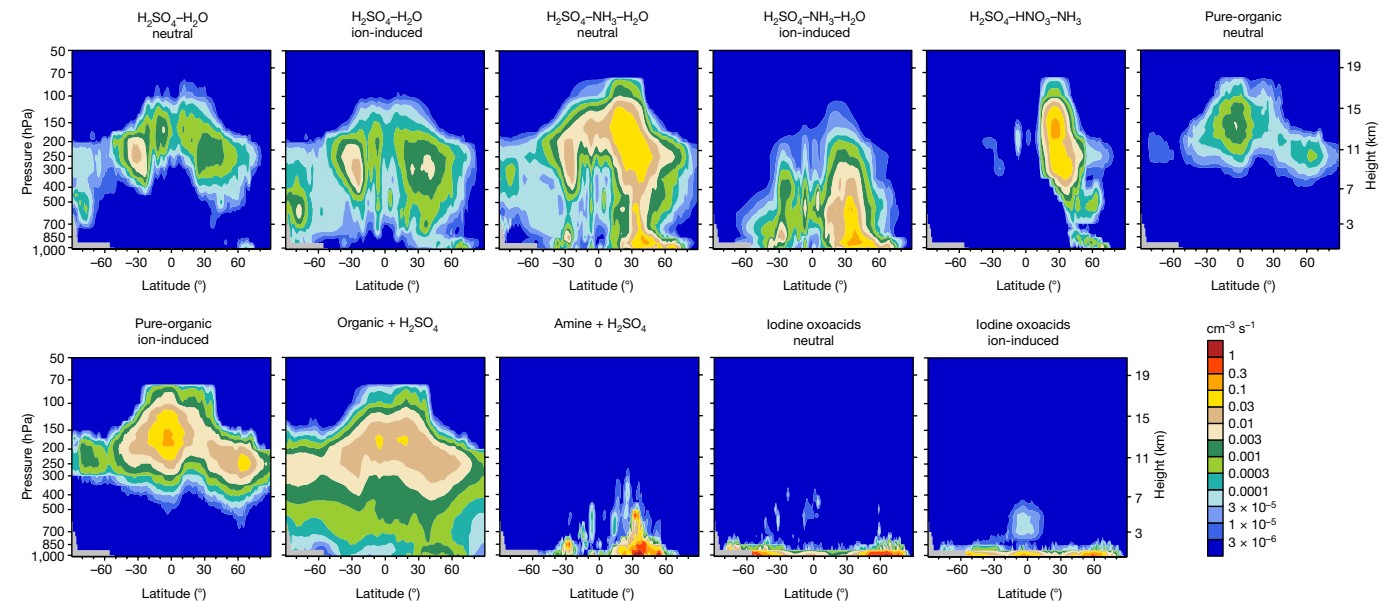

**Fig. 4 | Zonal mean NPF rates globally in 2016.** Each panel displays the rate of an individual NPF mechanism.

**a**
CCN concentration (0.5% supersaturation)

13 km AGL

**b**
Fraction of CCN from NPF

**c**
1 km AGL

**d**

**e**
Surface

**f**

**Fig. 5 | CCN concentrations and fractions of CCN caused by NPF at different vertical levels in 2016. a,c,e,** Spatial distribution of CCN concentrations at 0.5% supersaturation (CCN0.5%) at 13 km AGL (**a**), 1 km AGL (approximately at the low-cloud level) (**c**) and surface level (**e**). **b,d,e,** Fractions of CCN0.5% caused by NPF at 13 km AGL (**b**), 1 km AGL (**d**) and surface level (**f**). All concentrations are normalized to standard temperature and pressure. Maps were created using the NCAR Command Language (version 6.6.2), https://doi.org/10.5065/D6WD3XH5.

## Discussion

In this study, we developed a comprehensive global model representation of the physicochemical processes underlying NPF, which represents substantial advances compared with previous modelling studies (see the 'Discussion in the context of previous global models' section in Methods). The much-improved model, combined with observational constraints, provides a comprehensive overview of global NPF mechanisms that vary greatly with region and altitude (see schematic in Extended Data Fig. 1a). We found that particle formation in most aerosol-rich or cloud-susceptible regions is most probably dominated by 'modern' NPF mechanisms discovered or quantified in the past decade (that is, nucleation involving organics, amines, iodine oxoacids and $HNO_3$), rather than by traditional mechanisms involving only $H_2SO_4$, $NH_3$ and $H_2O$. The importance of these modern mechanisms on regional and global scales has been underappreciated because they were mostly missing or substantially biased in previous models. Our sensitivity experiments showed that the findings are robust across 13 key sources of uncertainty, but uncertainty might remain in aspects not covered by these experiments. Of note, our model has not included all NPF mechanisms exhaustively. New atmospherically relevant NPF mechanisms have been identified recently, especially those involving the synergistic effects of several compounds[47–50]. For example, amines and $NH_3$ have been found to nucleate synergistically with $H_2SO_4$, especially in environments with insufficient amines to fully stabilize large $H_2SO_4$ clusters[48,49]. $HNO_3$ has been shown to enhance DMA–$H_2SO_4$ nucleation under favourable conditions with relatively high $HNO_3$ and DMA concentrations[50]. It is more than likely that new synergistic effects or other new NPF mechanisms will be identified in future experiments that better mimic the atmospheric composition. Parameterizing the emerging new NPF mechanisms and incorporating them into the model could further refine and possibly modify the picture we present here; therefore, such work is needed in the future. Furthermore, our sensitivity experiments only tested one source of uncertainty at a time; combinations of numerous sources might lead to larger uncertainty in some parts of the multiparameter space. Nevertheless, our improved prediction of NPF processes could substantially reshape current understanding of global aerosol loadings and budgets. For example, the inclusion of modern NPF mechanisms in our model almost tripled the global mean surface particle number concentrations relative to those based on traditional NPF mechanisms involving $H_2SO_4$, $NH_3$ and $H_2O$ (Extended Data Fig. 2e,g). Our model evaluation over key regions suggests that the changes in number concentrations are realistic, but further evaluation using more observations would be invaluable. In particular, simultaneous measurements of nucleation precursors and particle size distributions are greatly needed, especially in the upper troposphere above Southeastern Asia, Central Africa, the Asian monsoon regions, the Eastern United States and Europe. Furthermore, direct detection of molecular clusters is encouraged in various regions of the world, especially in the upper troposphere.

The comprehensive representation of NPF mechanisms in this study will also facilitate detailed source apportionment of particles and CCN globally. This will further help with accurate attribution of aerosols' climatic effects to emission sources of precursor gases and primary particles, with implications for the development of targeted control policies. Furthermore, clarification of NPF mechanisms has important implications for assessments of historical and future climate change, because particles arising from different mechanisms may undergo markedly different changes. For example, during the industrial era, $H_2SO_4$–$HNO_3$–$NH_3$ nucleation has probably become enhanced substantially owing to the increase in anthropogenic emissions, whereas pure-organic and organic–$H_2SO_4$ nucleation have probably experienced more complex changes attributable to changes in both biogenic and anthropogenic emissions. This could have resulted in notably different aerosol loading changes in the upper troposphere above Asian monsoon regions, rainforests and tropical oceans during the industrial era. For another example, whether NPF in oceanic boundary layers is governed by iodine oxoacids or $H_2SO_4$ means that the climatic effects of aerosols will evolve differently under future policy interventions. In view of the regionally and vertically distinct NPF mechanisms (as well as their different past and future changes), it is imperative to adequately represent all of the main NPF mechanisms in climate simulations and projections, especially those considered in the assessment reports of the Intergovernmental Panel on Climate Change.

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

## Methods

### NPF module with 11 nucleation mechanisms

We developed an NPF module with 11 nucleation mechanisms and incorporated it in the Energy Exascale Earth System Model (E3SM) version 1 (refs. 51,52), which is a fully coupled Earth system model. The 11 mechanisms included $H_2SO_4$–$H_2O$ neutral and ion-induced nucleation, $H_2SO_4$–$NH_3$–$H_2O$ neutral and ion-induced nucleation, synergistic $H_2SO_4$–$HNO_3$–$NH_3$ nucleation, pure-organic neutral and ion-induced nucleation, organic–$H_2SO_4$ nucleation, amine–$H_2SO_4$ nucleation and iodine oxoacids neutral and ion-induced nucleation. The parameterizations of the NPF rates at 1.7-nm diameter ($J_{1.7}$) were mostly developed on the basis of experiments in the CLOUD (Cosmics Leaving Outdoor Droplets) chamber[15,20,21,53–56], with the assistance of cluster kinetic simulations and quantum chemistry simulations.

The parameterizations of seven nucleation mechanisms ($H_2SO_4$–$H_2O$ neutral and ion-induced mechanisms, $H_2SO_4$–$NH_3$–$H_2O$ neutral and ion-induced mechanisms, pure-organic neutral and ion-induced mechanisms and organic–$H_2SO_4$ mechanisms) represented an improved version of our previous regional modelling study[13], which was again revised on the basis of Gordon et al.[9]. There are three main updates compared with Gordon et al.[9]. First, we refitted the parameterization of organic–$H_2SO_4$ nucleation based on recent CLOUD experimental data reported in Lehtipalo et al.[57]. By contrast, Zhao et al.[13] and Gordon et al.[9] used the parameterization of Riccobono et al.[21], which was derived from laboratory experiments in which the precursor gases were not measured directly and was thus subject to large uncertainty. To fit the new parameterization, we used the experiments with a certain amount of $H_2SO_4$ concentration ($>2 \times 10^5$ cm$^{-3}$) and without the presence of $NH_3$; we excluded the experiments with extremely low $H_2SO_4$ ($<2 \times 10^5$ cm$^{-3}$) because organic–$H_2SO_4$ nucleation was probably not the dominant mechanism in those experiments. We tried two different forms of equations for NPF rates (in cm$^{-3}$ s$^{-1}$) that have been used previously[21,58]: $J_{\text{organic–H2SO4}} = k[H_2SO_4]^2[ORG]$ and $J_{\text{organic–H2SO4}} = k[H_2SO_4][ORG]$, in which $[H_2SO_4]$ represents the concentration of $H_2SO_4$ (cm$^{-3}$) and $[ORG]$ represents the concentration of non-nitrate HOM (highly oxygenated organic molecule) dimer (cm$^{-3}$), which was found to correlate best with NPF rate among all HOM categories investigated by Lehtipalo et al.[57]. We finally adopted the second equation form because it results in a higher correlation ($R^2 = 0.80$) than the first form ($R^2 = 0.61$). The fitted parameterization was $J_{\text{organic–H2SO4}} = 1.85 \times 10^{-14}[H_2SO_4][ORG]$. Second, whereas the original parameterizations of pure-organic nucleation used HOMs as inputs, we followed Zhao et al.[13] to use ULVOCs with a certain level of polarity (O:C > 0.4) as inputs because they have been shown to be better indicators of nucleating organics, especially considering the large range of variations in atmospheric temperature[26]. Accordingly, the original parameterizations were adjusted to suit the new inputs. Note that the ULVOC concentrations were calculated by summing the organic species with $C^*$ values of less than $3 \times 10^{-9}$ µg m$^{-3}$ in the R2D-VBS framework described in the subsequent section. Similarly, we revised the above-mentioned organic–$H_2SO_4$ nucleation parameterization to use the sum of ULVOCs and ELVOCs ($C^* < 3 \times 10^{-5}$ µg m$^{-3}$) with O:C > 0.4 as inputs, so as to match the outputs of the R2D-VBS framework. This is justified because ULVOCs and ELVOC largely overlap with non-nitrate HOM dimer at the temperature (278 K) under which the experiments of Lehtipalo et al.[57] were conducted. Third, we considered the temperature dependence of pure-organic and organic–$H_2SO_4$ nucleation rates. The temperature-dependence function, $J_T = J_{278K}\exp(-(T - 278)/13.0)$ (representing an increase in the NPF rate by a factor of 2.15 per 10 K temperature decrease), was determined by combining quantum chemistry calculations and the buffering effect of the volatility shift with temperature[13]. To examine its uncertainty, we conducted a sensitivity simulation using a much weaker temperature-dependence function, $J_T = J_{278K}\exp(-(T - 278)/20.0)$, which translates to an increase in the NPF rate by a factor of only 1.6 per 10 K temperature decrease.

For amine–$H_2SO_4$ nucleation, we focused on nucleation induced by DMA because it is most efficient in stabilizing $H_2SO_4$ clusters and is considered the key amine species driving NPF in the urban atmosphere[3,59,60]. The starting point of our parameterization was that derived from chamber experiments reported by Almeida et al.[54], who obtained an NPF rate at 3.2 nm ($J_{3.2}$) using a condensation particle counter (CPC) and scanning mobility particle sizer (SMPS) measurements, which they then extrapolated to 1.7 nm using the Kerminen and Kulmala[61] equation; however, their extrapolation did not include the effect of self-coagulation among particles. Kürten et al.[62] introduced a new method to retrieve NPF rates at sizes below the detection threshold of the instrument, which explicitly considers the effect of self-coagulation. The re-evaluated $J_{1.7}$ values are a factor of 10 faster than the estimates of Almeida et al.[54] and agree almost perfectly with rates calculated from the output of a kinetic aerosol model. For this reason, we applied a scaling factor of 10 to the parameterization of Almeida et al.[54]. Furthermore, previous studies showed that amine–$H_2SO_4$ nucleation depends substantially on temperature, which is not considered in the parameterization of Almeida et al.[54]. In the current study, we derived the following temperature-dependence function using the cluster kinetic model developed by Cai et al.[4] and applied it to the above parameterization:

$$J_T = J_{278K}(1.576\exp(-((T - 250.6)/23.18)^2) + 0.6956\exp(-((T - 273.1)/13.01)^2))$$

in which $T$ is temperature (unit: K).

For the synergistic $H_2SO_4$–$HNO_3$–$NH_3$ mechanism, Wang et al.[15] conducted experiments in the CLOUD chamber at 223 K, which is a temperature typical of the upper troposphere in the Asian monsoon region (205–230 K according to our model). They used $H_2SO_4$, $NH_3$ and $HNO_3$ concentrations of 0.05–0.5 ppt, 19–80 ppt and 0.03–0.2 ppb, respectively, that is, all similar to real-world conditions in the upper troposphere of the Asian monsoon region according to observations and our model simulations (Extended Data Fig. 4, Supplementary Figs. 2 and 3 and Supplementary Fig. 10). Wang et al.[15] also suggested applying a temperature-dependence function derived for the $HNO_3$–$NH_3$ system to the $H_2SO_4$–$HNO_3$–$NH_3$ system because of the lack of direct measurements for the latter. However, this function would lead to unrealistically large NPF rates near the surface of anthropogenically polluted regions. Given that the experiments for $H_2SO_4$–$HNO_3$–$NH_3$ nucleation were conducted only at 223 K and that previous studies have shown that $H_2SO_4$–$HNO_3$–$NH_3$ nucleation is not detectable at warm temperatures[63], we assumed that the parameterization of Wang et al.[15], as well as the temperature-dependence function, should be applied only to temperatures <238 K. At higher temperatures, the NPF rate was set to zero and smooth transition was implemented near 238 K to avoid abrupt change.

For iodine oxoacids nucleation, we derived parameterizations of neutral and ion-induced NPF rates by fitting the CLOUD experimental data reported by He et al.[55]. The challenge was to obtain the temperature-dependence function because the experiments by He et al.[55] did not cover the full temperature range but instead focused only on +10 °C and −10 °C. For neutral nucleation, the NPF rate increases substantially when the temperature changes from +10 °C to −10 °C. We assumed that the temperature-dependence function has the Arrhenius form and we fitted the function to the experimental data at +10 °C and −10 °C. Moreover, previous quantum chemistry calculations[64] have shown that the neutral NPF rate increases minimally with further temperature reduction below −10 °C because the iodine oxoacid clusters are already highly stable at −10 °C. Therefore, we assumed that the fitted temperature-dependence function only applies to temperatures above −10 °C and that the neutral NPF rate is independent of temperature below −10 °C. For ion-induced nucleation, we derived the NPF rates using the difference between the measurements under neutral and galactic cosmic ray conditions; the NPF rates can be derived

accurately at +10 °C but not at −10 °C because the NPF rates for neutral and galactic cosmic ray conditions are very similar at −10 °C. However, He et al.[55] proved that ion-induced nucleation proceeds at the kinetic limit below +10 °C. Therefore, we did not consider the temperature dependence of ion-induced nucleation below +10 °C. The temperature dependence above +10 °C was assumed to be the same as that of neutral nucleation. The parameterizations of neutral and ion-induced NPF rates are given below:

$$J_{\text{iodine oxoacids neutral}}$$
$$= 2.57 \times 10^{-32}[\text{HIO}_3]^{4.23}$$
$$(1.40 \times 10^{-46}\exp(29,900/\max(T, 263)))$$

$$J_{\text{iodine oxoacids ion−induced}}$$
$$= 1.28 \times 10^{-18}[\text{HIO}_3]^{2.48}([\text{ION}]/700)$$
$$(1.40 \times 10^{-46}\exp(29,900/\max(T, 283)))$$

in which $\text{HIO}_3$ is the iodic acid concentration (unit: $\text{cm}^{-3}$), [ION] is the ion concentration (unit: $\text{cm}^{-3}$) and $T$ is temperature (unit: K).

For all nucleation mechanisms, we calculated the NPF rates in the model using time-step mean concentrations of precursors, including $\text{H}_2\text{SO}_4$, ULVOCs, ELVOCs, DMA, $\text{HIO}_3$, $\text{NH}_3$ and $\text{HNO}_3$. The time-step mean concentrations of $\text{H}_2\text{SO}_4$, DMA and $\text{HIO}_3$ were calculated on the basis of a pseudo-steady-state assumption[65], which is reasonable given the time step of 30 min. For ULVOCs and ELVOCs, because the model calculates the condensation/evaporation of organic gases with adaptive time stepping, as will be described below, we calculated the mean concentrations over each condensation/evaporation sub-time step by averaging the initial and final concentrations and subsequently calculated the mean concentrations over the entire time step (30 min) by taking the weighted average of each sub-time step. The concentrations of $\text{NH}_3$ and $\text{HNO}_3$ in the model were prescribed (see detailed descriptions below). The new particles at 1.7-nm diameter were injected into the nucleation mode. The amount of gas precursors consumed in the nucleation process was subtracted from the gas-phase concentrations.

## R2D-VBS and its parameterizations

A substantial challenge in the simulation of organic-mediated nucleation is the high complexity of the formation processes of ULVOCs and ELVOCs. To address this challenge, we incorporated the R2D-VBS framework in the E3SM to simulate the chemical transformation and volatility distribution of monoterpene oxidation products (including ULVOCs and ELVOCs) in the full atmospheric temperature range, with the R2D-VBS parameters optimized against laboratory experiments[13,26,66]. The R2D-VBS explicitly treats the peroxy radical ($\text{RO}_2$) chemistry and tracks the distribution of radical termination products with the two-dimensional space defined by $C^*$ (from $10^{-10}$ to $10^6$ $\mu$g $\text{m}^{-3}$ separated by intervals of one order of magnitude) and the oxygen-to-carbon ratio (O:C, from 0 to 1.3 separated by intervals of 0.1). Specifically, the reactions begin with oxidation of monoterpenes with OH, $\text{O}_3$ and $\text{NO}_3$, producing $\text{RO}_2$. Then, $\text{RO}_2$ undergoes either autoxidation or termination. Autoxidation produces a more-oxygenated $\text{RO}_2$, which will further undergo autoxidation or termination. Termination proceeds through unimolecular termination or reactions with NO, $\text{HO}_2$ or another $\text{RO}_2$. The cross-reactions of $\text{RO}_2$ produce either dimer or non-dimer products. The non-dimer products of $\text{RO}_2$ cross-reactions, as well as the products of unimolecular termination and reaction with NO, will undergo either functionalization or fragmentation, with a branching ratio ($\beta$) between the two that depends on the O:C ratio of the RO intermediates produced in the termination processes. Then we distributed the stable molecules from each of the $\text{RO}_2$ termination pathways to a series of species in the $C^*$–O:C space by means of kernels. Kernels define the rule for mapping a reactant ($\text{RO}_2$ in this case) with a given $C^*$ and O:C ratio to a distribution of reaction products in the

$C^*$–O:C space through a specific termination pathway. The parameters of the R2D-VBS were determined from experimental chemical kinetics literature whenever possible. However, there are some tunable parameters that are either not measurable directly or represent the mean state of many species or reactions. The values of these tunable parameters (a set of which is called 'parameterization') were optimized by simulating a series of laboratory experiments with the R2D-VBS and minimizing the differences between the simulated and measured HOMs and secondary organic aerosols (SOA). These experiments involved oxidation of monoterpenes by OH and $\text{O}_3$ in smog chambers and oxidation flow reactors under different temperatures. Notably, isoprene might suppress the NPF triggered by monoterpene oxidation products at low altitudes[67–69], whereas sesquiterpenes might contribute to NPF despite their much smaller concentrations than those of monoterpenes[70]; these effects were not considered in the model, although the formation of SOA from isoprene oxidation was considered. Further details of the R2D-VBS and the parameter optimization are described in our previous works[13,26,66].

When we incorporated the experimentally constrained R2D-VBS in the E3SM, we simplified it to reduce the computational burden associated with many R2D-VBS species. Specifically, we summed the species in the same $C^*$ bin and different O:C bins, which equivalently condensed the R2D-VBS to a 1D-VBS with $C^*$ ranging from $10^{-10}$ to $10^6$, thereby reducing the number of species advected in the model. Meanwhile, we retained all original chemical reactions within the R2D-VBS such that the simplification did not affect the simulation results. It is noted that we only included species with an O:C ratio of >0.4 in the condensed 1D-VBS. Thus, the total ULVOC and ELVOC concentrations within the condensed 1D-VBS were used to drive organic-mediated nucleation. The remaining less-oxygenated compounds (that is, with O:C ≤ 0.4) do not contribute to nucleation in our model but they might still contribute to the formation of SOA. We simulated the SOA formation associated with those less-oxygenated compounds using the original SOA parameterization in the E3SM[71], which was based on a simple 1D-VBS involving five surrogate species with C* ranging from $10^{-1}$ to $10^3$ $\mu$g $\text{m}^{-3}$. Then, we parameterized the SOA fraction formed from the less-oxygenated compounds by fitting a series of box-model simulations under various temperatures following Zhao et al.[13], and we applied the parameterized fraction below to the original SOA formation parameterization in the E3SM:

$$f = 1.0 − 0.37/(\exp(−0.0597 \times T + 14.02) + 1.0)$$

in which $T$ is temperature (unit: K).

## Representation of the source and sinks of DMA

We calculated the DMA concentrations by implementing the sources and sinks of DMA in the E3SM. Considering that bottom-up DMA emission inventories are lacking at present owing to insufficient source measurements, we estimated the DMA emissions based on $\text{NH}_3$ emissions (see the subsequent section) and DMA/$\text{NH}_3$ ratios. In this study, we used the source-dependent DMA/$\text{NH}_3$ emission ratios derived by Mao et al.[72] through source apportionment analysis based on simultaneous ambient observations of DMA, $\text{NH}_3$, $\text{NO}_x$, $\text{SO}_2$ and meteorological parameters at a suburban site in Nanjing, China. The DMA/$\text{NH}_3$ ratio for maritime emissions was derived from a recent campaign in offshore areas of China[73]. We used specific DMA/$\text{NH}_3$ ratios of 0.0070, 0.0018, 0.0015, 0.0100, 0.0009 and 0.0144 for chemical−industrial, other industrial, agricultural, residential, transport and maritime sources, respectively.

We explicitly represented the removal of DMA through gas-phase oxidation, aerosol uptake and wet deposition. The oxidation of DMA by •OH was assumed to proceed at a constant rate of $6.49 \times 10^{-11}$ $\text{cm}^{-3}$ $\text{s}^{-1}$ ($5.85–7.13 \times 10^{-11}$ $\text{cm}^{-3}$ $\text{s}^{-1}$ in the literature[74]), whereas reactions with $\text{O}_3$ and $\text{NO}_3$ were neglected owing to their much slower rates. The aerosol

uptake of DMA is subject to higher uncertainty because the uptake coefficient varies over a relatively large range in the literature ($5.9 \times 10^{-4}$ to $4.4 \times 10^{-2}$)[75,76]. Therefore, we assumed the uptake coefficient to be 0.001, which is approximately the median value determined from recent laboratory measurements[75,76]. The wet deposition of DMA was calculated on the basis of Henry's law with a Henry's law constant of $5.7 \times 10^4$ mol m$^{-3}$ atm$^{-1}$ ($3.0 \times 10^4$ to $6.1 \times 10^4$ mol m$^{-3}$ atm$^{-1}$ in the literature[77]). We designed a sensitivity experiment to examine the impact of the uncertainty in DMA concentration simulations (see detailed descriptions below).

### Representation of sources and sinks of iodine oxoacids

We incorporated in the model 14 iodine species: $I_2$, HOI, HI, I, IO, OIO, IOIO, $I_2O_3$, $I_2O_4$, $IOIO_4$, $HIO_3$, INO, $INO_2$ and $IONO_2$. We calculated the emission fluxes of $I_2$ and HOI online using surface wind speed, surface $O_3$ concentration and sea-surface temperature following the parameterizations reported by Karagodin-Doyennel et al.[78]. The emissions of organic iodine species were not included because they are much smaller than the emissions of inorganic iodine and thus contribute less to reactive iodine species in the atmosphere[79,80]. We simulated the iodine chemistry by incorporating in the E3SM the chemical box model described by Finkenzeller et al.[81], which considers gas-phase radical reactions, thermal decomposition reactions and photochemical reactions (see a full list of the reactions and their rate coefficients in Supplementary Table 2). This box model includes the previously missing $HIO_3$ formation reactions found by Finkenzeller et al.[81], that is, the conversion of IOIO to $IOIO_4$ and then to $HIO_3$. We further added the reaction of OIO with OH to produce $HIO_3$, following Plane et al.[82]. Moreover, we considered the aerosol uptake of the iodine species using the uptake coefficients listed in Supplementary Table 3. Field observations and laboratory experiments showed that particulate $IO_3^-$ is readily reduced, effectively recycling iodine to the gas phase[83,84]. Because the detailed chemistry for $IO_3^-$ reduction is complicated and not entirely clear, we assumed that the condensed $HIO_3$ instantaneously reemitted into the gas phase as HOI, following Finkenzeller et al.[81]. We did not explicitly consider iodine recycling following the uptake of other iodine species; the effect of these processes, together with other uncertainties in iodine chemistry, was considered in the sensitivity simulations described below. We used Henry's law to calculate the wet deposition of iodine species using the Henry's law constants summarized in Supplementary Table 3. To reduce the computational amount associated with the representation of iodine chemistry, we constructed a simplified version of the iodine chemistry in which minor species ($I_2O_3$, $I_2O_4$, INO, $INO_2$ and $IONO_2$) as well as the associated chemical reactions were removed from the complete version. The simulated monthly mean $HIO_3$ concentrations by the complete and simplified versions closely match each other on the ocean.

### Configuration of the revised E3SM

The E3SM simulates aerosol processes with the five-mode modal aerosol module (MAM5) (K.Z., J.S. & P.-L.M., manuscript in preparation), which is a revised version of the four-mode modal aerosol module (MAM4)[85,86]. The MAM5 represents the particle number size distributions with five lognormal size modes: nucleation, Aitken, accumulation, coarse and primary carbon modes. Aerosols are assumed to be internally mixed in each mode but externally mixed between different modes. In comparison with the MAM4, the nucleation mode is newly added in the MAM5 to better represent NPF and the number size distribution of ultrafine particles. Also added are physical processes related to the nucleation mode, including condensation, renaming (conversion from smaller modes to larger modes), coagulation, transport, dry deposition and wet deposition. Condensation is a key process driving the growth of newly formed particles to CCN size. The model explicitly represents the condensation of $H_2SO_4$ and organic vapours across the entire volatility range (including ULVOCs and ELVOCs).

The condensation of $H_2SO_4$ is treated dynamically as an irreversible process, using standard mass-transfer expressions that are integrated over the size distribution of each mode[87]. The condensation and evaporation of organic vapours are treated dynamically as reversible processes and calculated using a semi-implicit Euler approach with adaptive time stepping on the basis of Zaveri et al.[88]. The Kelvin effect is accounted for in the calculation of condensation rates. Condensation can result in smaller-mode particles growing into the size range of the next larger mode. Thus, after condensation is calculated, the renaming module reallocates the number and mass concentrations of the subset of smaller-mode particles that exceed a specified threshold diameter to the next larger mode. The threshold diameter is defined as the geometric mean of the characteristic diameters of two neighbouring modes, in which the characteristic diameter of a mode was assumed to be the nominal volume mean diameter (determined by the nominal number median diameter and the geometric standard deviation) for that mode (ref. 86 and K.Z., J.S. & P.-L.M., manuscript in preparation).

The modal approach to represent particle number size distribution has been used in more than 85% of the models involved in the Coupled Model Intercomparison Project phase 6 (CMIP6)[89], owing to a good balance between accuracy and computational efficiency. Moreover, modal modules with a nucleation mode have been widely used to simulate NPF and its climate impacts[24,90–94]. Mann et al.[95] systematically compared particle simulations using a modal aerosol microphysics module with a nucleation mode against those simulated by a 'sectional' module, which represents particle number size distribution with discrete size bins in the same host global model. They concluded that the modal and sectional modules generally perform similarly against observed size-resolved particle number concentrations, with the differences between the two modules being much smaller than model–observation differences. The particle simulations of the modal module, as well as the degree of agreement with the sectional module and observations, are affected by several parameters defining the structure of the modal module, including the mode width (geometric standard deviation) and threshold diameter for renaming[95]. Here we perturbed these parameters to examine the sensitivity of the simulation results to the structure of the model. In the first sensitivity simulation ('Small_Mode_Width'), the mode widths of the nucleation, Aitken and accumulation modes were reduced from 1.6, 1.6 and 1.8 in the best-case scenario to 1.45, 1.45 and 1.4, respectively, to roughly capture the range reported in previous number size distribution observations and modelling studies (Mann et al.[95] and references therein). Accordingly, in the second sensitivity run ('Large_Mode_Width'), the mode widths of the three modes were increased to 1.8, 1.8 and 2.0, respectively. The third sensitivity run ('New_Threshold_Diameter') perturbed the threshold diameter for renaming; specifically, the default threshold diameter described above was modified to the upper size limit for the smaller mode involved in renaming (8.7 nm and 53.5 nm for nucleation and Aitken modes, respectively), which was smaller than the default threshold diameter, following several previous studies[96,97]. Considering that the relative importance of different NPF mechanisms, the main focus of this work, is hardly affected by the above parameters, we concentrate on the impact of these parameters on the fraction of CCN caused by NPF, as shown in Supplementary Fig. 11. The general pattern of the region-dependent and altitude-dependent fraction of CCN from NPF remain unchanged in all sensitivity simulations. The magnitude of the fraction of CCN from NPF could vary by up to 5% in the lower troposphere and 10% in the upper troposphere relative to the best-case scenario, but the fractions of CCN from NPF in almost all regions are still within the ranges described in the main text. Therefore, the uncertainty in the modal module structure should not affect the main conclusion of this study.

The treatment of SOA formation follows Lou et al.[71], except that the treatment of monoterpene SOA was replaced by the R2D-VBS module implemented in this work. Further details about the representations of aerosol processes other than SOA formation are available in

Wang et al.[85]. The model uses a unified treatment of the convective transport of aerosols and gases, as well as the convective wet removal of aerosols, with consideration of secondary activation for aerosols above the cloud base[85,98].

For gas chemistry, the model explicitly represents the oxidation of DMS to $SO_2$ and subsequently to $H_2SO_4$ (ref. 99). The model digests prescribed monthly mean concentrations of oxidants (OH, $O_3$ and $NO_3$) as well as $NH_3$, NO and $HNO_3$, which are derived from the outputs of CAM-Chem[100] with $NH_3$ concentrations in the Pacific and Atlantic upper troposphere scaled to match large-scale satellite retrievals[42]. Note that $O_3$ is prescribed only in the troposphere but is predicted above the troposphere. The diurnal variation of these prescribed species is calculated online following Lou et al.[71]. The wet deposition of gas species, such as the organic oxidation products (including the lumped R2D-VBS species), is simulated on the basis of Henry's law; this is similar to the treatment of the wet deposition of DMA and iodine species described in the previous two sections. The Henry's law constants for the organic oxidation products are estimated as a function of volatility and precursor type following Hodzic et al.[101].

We used the 'ne30' grid configuration with horizontal resolution of approximately 1° (5,400 spectral elements) and 72 vertical levels. We nudged the wind fields to the Modern-Era Retrospective analysis for Research and Applications, version 2 (MERRA-2) reanalysis data (available at 3-h intervals), with a relaxation timescale of 6 h. We applied nudging every model time step (that is, 30 min) and linearly interpolated the 3-h MERRA-2 data to model time steps to constrain model simulations, following Sun et al.[102]. Simulations for the comprehensive best-case scenario were conducted for eight consecutive years (2013–2020), with the first year used as the spin-up period. Such long-term simulations facilitated comparison with observational data, which were mostly distributed between 2014 and 2020. The duration of sensitivity simulations conducted to quantify the NPF contributions or to analyse the uncertainty was 15 months (October 2015 to December 2016) and the simulations were initialized from the output of the longer best-case simulation. The simulation results for 2016 were used for most of the analysis presented in this paper.

We used anthropogenic and biomass-burning emissions as well as concentrations of greenhouse gases from CMIP6 emission data[103] with further corrections[104]. We estimated emissions of intermediate volatility organic compounds from anthropogenic and biomass burning to be four times greater than the emissions of primary organic aerosol, following Lou et al.[71], but with the same spatiotemporal distribution as that of the primary organic aerosol. We obtained DMS emissions from a coupled-model simulation with detailed representation of DMS formation in seawater[105]. We used the 2014 emissions in our simulations. An exception was the $SO_2$ emissions in China, for which we used year-by-year emissions from ABaCAS-EI, a local emission inventory developed by Tsinghua University[106,107]. This is because $SO_2$ emissions in China have changed substantially during the simulation periods and because $SO_2$ emissions are closely tied to $H_2SO_4$ concentrations and, hence, to NPF rates. Emissions of biogenic volatile organic compounds were calculated offline using the Model of Emissions of Gases and Aerosols from Nature[108]; average emissions for 2006–2010 are used in our simulations.

### Descriptions of observational data

We used a series of observational data to evaluate our model and to support our findings. We obtained particle measurements over the Amazon rainforest from the ACRIDICON-CHUVA (Aerosol, Cloud, Precipitation, and Radiation Interactions and Dynamics of Convective Cloud Systems–Cloud processes of the main precipitation systems in Brazil: a contribution to cloud resolving modeling and to the GPM) campaign[30,109], which was conducted in September 2014. Specifically, we used total particle number concentration measurements made by a CPC (modified GRIMM CPC 5.410 by GRIMM Aerosol Technik)

aboard the German High Altitude and LOng range (HALO) aircraft. The CPC has a nominal cutoff diameter of 4 nm, but owing to inlet losses, the effective cutoff diameter is approximately 10 nm near the surface and increases to approximately 20 nm at 150 hPa (approximately 13.8 km). Because monoterpenes have established connections to organic-mediated NPF, we also evaluated the model against monoterpene concentrations measured by an Ionicon quadrupole high-sensitivity proton-transfer-reaction mass spectrometer (PTR-MS) aboard the G-1 aircraft during the GoAmazon (Observations and Modeling of the Green Ocean Amazon) campaign[38]. For China, we used particle number size distribution measurements obtained at three sites: the BUCT (Beijing University of Chemical Technology) site in Beijing, the SORPES (Station for Observing Regional Processes of the Earth System) site in Nanjing[110,111] and the Wangdu site in Hebei. The particle number size distributions from 1 nm to 10 μm at BUCT were measured during January–May and October–December 2018 using two home-made systems: a diethylene glycol scanning mobility particle sizer (DEG-SMPS) and a particle number size distribution system that included another SMPS for a larger size range and an aerodynamic particle sizer[112–114]. The particle number size distributions at SORPES were measured during January–December 2019 using a SMPS with a nano-differential mobility analyser and a long-differential mobility analyser, covering a size range of 4–500 nm (ref. 115). The number size distributions at Wangdu were measured using a combination of a nano condensation nucleus counter system (model A11, Airmodus), a nano-SMPS (consisting of a differential mobility analyser 3085 and a CPC3776, TSI), a long-SMPS (consisting of a differential mobility analyser 3081 and a CPC3775, TSI) and a neutral cluster and air ion spectrometer (NAIS, Airel Ltd.) for the size range 1.34–661.2 nm during December 2018 and January 2019 (Y. Q. Lu, manuscript in preparation). We also compared our simulations with a series of DMA measurements made by chemical ionization mass spectrometry (CIMS) or using a Vocus proton-transfer-reaction time-of-flight mass spectrometer (Vocus PTR-TOF) and $H_2SO_4$ measurements made by CIMS in China, Europe and the United States. The time periods and sources of the DMA and $H_2SO_4$ data at individual sites are given in the caption of Extended Data Fig. 3. Over the Pacific and Atlantic oceans, we used particle number size distribution measurements acquired during the ATom campaign (July–August 2016, January–February 2017, September–October 2017 and April–May 2018)[7,46], during which four flights profiled altitudes between approximately 0.18 km and 11–13 km at latitudes of 81° N to 65° S. The number size distributions from 2.7 nm to 4.8 μm were measured using a suite of instruments, including a nucleation-mode aerosol size spectrometer for diameters of 2.7–60 nm, an ultrahigh-sensitivity aerosol spectrometer for diameters of 60–500 nm and a laser aerosol spectrometer that extended this distribution to 4.8 μm. Finally, we used $HIO_3$ concentration observations summarized by He et al.[55] for ten oceanic or coastal sites around the world.

### Sensitivity analysis of NPF mechanisms

To test the robustness of our main conclusions about NPF mechanisms, we conducted sensitivity simulations with respect to key uncertainty factors. A list of the sensitivity experiments is presented in Supplementary Table 1.

$H_2SO_4$ is a key species involved in seven of the 11 NPF mechanisms considered in our model. $H_2SO_4$ is mainly formed from the oxidation of $SO_2$ emissions in terrestrial regions and from the oxidation of both $SO_2$ and DMS emissions in oceanic regions. The $SO_2$ emissions used in this study were obtained from the CMIP6 emission dataset[104], based on the Community Emissions Data System (CEDS) emission inventory[103]. Among all air pollutants, the emissions of $SO_2$ can be most accurately estimated on the basis of the mass balance of sulfur. Smith et al.[116] estimated the overall global uncertainty in $SO_2$ emissions of the CEDS inventory to be 8–14% and regional uncertainties to be within 30%. Another estimate[117] based on a similar EDGARv4.3.2 emission inventory reported

that $SO_2$ emissions have uncertainty of 14.4–47.6% at the regional level. Overall, we conclude, with confidence, that the uncertainty of $SO_2$ emissions at the regional level should be well within 50%. The marine DMS emissions used in this study were obtained from Wang et al.[105,118], who reported an annual marine DMS emission of 20.4 TgS year$^{-1}$ based on surface ocean DMS concentrations and a parameterization of sea-to-air gas transfer velocity. Using generally the same method, Lana et al.[119] derived an annual marine DMS emission of 28.1 TgS year$^{-1}$. They further showed that the uncertainty in the parameterization of sea-to-air gas transfer velocity resulted in a DMS emission uncertainty range of 17.6–34.4 TgS year$^{-1}$, whereas the uncertainty in the underlying data used to derive surface ocean DMS concentrations resulted in a DMS emission uncertainty range of 24.1–40.4 TgS year$^{-1}$. Taken together, the uncertainty of the marine DMS emission estimate should be within 100% (that is, a factor of 2). To evaluate the impact of such uncertainty, we conducted two sensitivity experiments: the first one ('1.5*$SO_2$_2*DMS'), compared with the best-case simulation, increased the $SO_2$ and DMS emissions by a factor of 1.5 and 2, respectively, and the second one ('0.67*$SO_2$_0.5*DMS') reduced the $SO_2$ and DMS emissions by a factor of 1.5 and 2, respectively. We also looked at the uncertainty from a different perspective. Our model evaluation showed that the simulated $H_2SO_4$ concentration was generally within a factor of 3 of that of the observations across various polluted regions (Extended Data Fig. 3) and that this uncertainty range encompassed not only the uncertainty in the precursor emissions but also the uncertainty in other chemical and physical processes. Accordingly, we conducted two more sensitivity simulations ('0.33*$H_2SO_4$' and '3*$H_2SO_4$') by reducing and increasing the $H_2SO_4$ concentrations simulated in the best-case scenario by a factor of 3 to cover the uncertainty in $H_2SO_4$ concentration. Extended Data Figs. 5–8 show that the four sensitivity scenarios have limited impact on the relative contributions of individual NPF mechanisms over our main regions of interest, largely because the increase or decrease in $H_2SO_4$ concentration causes simultaneous changes in the rates of most NPF mechanisms.

ULVOCs and ELVOCs drive organic-mediated nucleation (pure-organic and organic–$H_2SO_4$ nucleation), which—according to our model—is the dominant mechanism in the upper troposphere above three rainforest regions (that is, the Amazon, Southeastern Asia and Central Africa) and probably one of the two primary mechanisms in the upper troposphere above the Pacific and Atlantic oceans. A recent box-model study also showed that organic–$H_2SO_4$ nucleation could potentially reproduce the most NPF events in the upper troposphere above the Pacific Ocean among different nucleation mechanisms, although the precursor concentrations were tuned rather than predicted in that work[120]. ULVOCs and ELVOCs in our model are mainly formed through the oxidation of monoterpene emissions that are lifted to the upper troposphere by convection. Guenther et al.[121] estimated that monoterpene emissions from the Model of Emissions of Gases and Aerosols from Nature that was used in this study have uncertainty of a factor of 3. To better evaluate the uncertainty in monoterpene emissions, we compared simulated monoterpene concentrations with field observations[38,39] in two of the three main rainforest regions, that is, the Amazon and Southeastern Asia, which are the largest sources of monoterpene globally[121] and key regions of interest in this work. Average simulated concentrations during the periods of the field campaigns (February–March and September–October 2014 in the Amazon and April–July 2008 in Southeastern Asia) are 0.19 and 0.13 ppb, respectively, which are comparable with the observed values of 0.13 and 0.17 ppb; the model–observation differences are well within the uncertainty of a factor of 3 estimated by Guenther et al.[121]. To test the potential impact of emission uncertainty, we designed a sensitivity experiment ('0.33*MT') in which the monoterpene emissions are reduced by a factor of 3; this probably represents an extreme case given the above-mentioned good model–observation agreement of monoterpene concentrations. We did not test higher monoterpene

concentrations because they would not challenge the leading roles of organic-mediated nucleation over the regions of interest. Extended Data Figs. 5, 7 and 8 show that, even with reduced monoterpene emissions, organic-mediated nucleation remains the largest NPF mechanism in the upper troposphere above the three rainforest regions and one of the two primary mechanisms in the upper troposphere above the Pacific and Atlantic oceans.

We also designed two sensitivity experiments to examine the parameterizations of organic-mediated nucleation. The first experiment ('org-weak-T-dependence') assumed a much weaker temperature dependence of pure-organic and organic–$H_2SO_4$ nucleation rates ($J_T = J_{278K}\exp(-(T-278)/20.0)$) than that in the best-case scenario. We did not test stronger temperature dependence because it would lead to larger NPF rates over the regions of interest and thus would not challenge the crucial roles of organic-mediated nucleation. The second experiment ('organic-$H_2SO_4$_Riccobono') set organic–$H_2SO_4$ nucleation parameterization to that reported in Riccobono et al.[21]; this was because the parameterization of Riccobono et al.[21] has been used in previous modelling studies[13,22,23,53], although it was subject to uncertainty because it was derived from laboratory experiments in which the precursor gases were not detected directly. Extended Data Fig. 5 shows that, in the upper troposphere over the rainforest, none of the sensitivity experiments affected our key conclusion that organic-mediated nucleation plays a dominant role; in particular, pure-organic ion-induced nucleation is the largest above 10–12 km, at which the highest NPF rates occur. In the upper troposphere above the Pacific and Atlantic oceans (Extended Data Figs. 7 and 8), organic–$H_2SO_4$ nucleation and $H_2SO_4$–$NH_3$–$H_2O$ neutral nucleation remain the two primary nucleation mechanisms under all sensitivity experiments. It also holds that organic–$H_2SO_4$ nucleation prevails over wide latitude ranges, whereas $H_2SO_4$–$NH_3$–$H_2O$ neutral nucleation could be important in certain mid-latitude areas.

DMA is a key precursor involved in amine–$H_2SO_4$ nucleation, which our study suggests is the leading NPF mechanism near the surface in anthropogenically polluted regions (Eastern China, India, Europe and Eastern United States). Here we examined the uncertainty of simulated DMA concentrations by means of evaluation against observations in different anthropogenically polluted regions (Extended Data Fig. 3). The evaluation results indicate that the simulated concentrations are within a factor of 2.5 of the observed values. Thus, we designed a sensitivity experiment ('0.4*DMA') that reduces the DMA concentrations by a factor of 2.5 relative to the best-case simulation. Note that this across-the-board reduction probably represents an extreme case according to the above evaluation results, because the model-to-observation ratios at different sites vary between 2.5 and 1/2.5, with less than half of the sites exhibiting an overestimation. Similar to the case of ULVOCs/ELVOCs, we did not test higher DMA concentrations because they would not alter the leading roles of amine–$H_2SO_4$ nucleation. Besides, we conducted another experiment ('amine-$H_2SO_4$_Almeida') that used the NPF rate parameterization derived directly from CLOUD chamber experiments reported by Almeida et al.[54]. In comparison, the parameterization used in the best-case scenario also considered the self-coagulation of small particles and the temperature dependence of NPF rates. Extended Data Fig. 6 illustrates that, under both sensitivity scenarios, amine–$H_2SO_4$ nucleation remains the dominant nucleation mechanism near the surface of the four anthropogenically polluted regions. Furthermore, the results on the dominant role of amine–$H_2SO_4$ nucleation are consistent with recent observational studies that directly measured molecular clusters at a few polluted sites[3,4,60], which at least lends some support to our assessment results.

NPF mechanisms involving $NH_3$ have been shown by our model to play key roles in the upper troposphere above the Asian monsoon regions (including Eastern China and India) and the Pacific and Atlantic oceans. Specifically, our model shows that synergistic $H_2SO_4$–$HNO_3$–$NH_3$ nucleation is probably a leading mechanism in the upper

troposphere above Eastern China and India, whereas organic–$H_2SO_4$ nucleation and $H_2SO_4$–$NH_3$–$H_2O$ neutral nucleation are the two primary NPF mechanisms in the upper troposphere above the Pacific and Atlantic oceans. In the upper troposphere of the Asian monsoon region, the simulated $NH_3$ concentration during summer mostly varies in the range 10–40 ppt and it occasionally reaches 60 ppt (Extended Data Fig. 4), consistent with large-scale satellite observations (that is, mostly 10–35 ppt, occasionally 150 ppt)[42,43]. Observations have revealed that the upper-tropospheric $NH_3$ concentration is highly non-uniform because $NH_3$ is mainly uplifted by deep convection. Therefore, even with average $NH_3$ concentrations similar to those observed, our results represent a lower-limit estimate of the $H_2SO_4$–$HNO_3$–$NH_3$ nucleation rate, because the $H_2SO_4$–$HNO_3$–$NH_3$ nucleation rate is much more strongly dependent on the $NH_3$ concentration than linear. Nevertheless, the NPF rate at high $NH_3$ concentration will be limited by the availability of $H_2SO_4$. Here we considered an extreme case ('nonuniform-$NH_3$') in which the $NH_3$ concentration was set at 1 ppb (consistent with observations in convective outflow hotspots by Höpfner et al.[43]) in [average $NH_3$]/1 ppb of the area of each model grid and zero in the remaining area of the model grid in the upper troposphere. For the areas with the presence of $NH_3$, we assumed that $H_2SO_4$ is exhausted by nucleation. In this sensitivity case, the fractional contribution of $H_2SO_4$–$HNO_3$–$NH_3$ nucleation in the upper troposphere above Eastern China and India increased moderately by approximately 10–25% from the best-case levels (Extended Data Fig. 9), which reveals strong limitation of $H_2SO_4$ availability in this case. For a scenario intermediate between the best-case simulation and the above sensitivity simulation, the contribution of $H_2SO_4$–$HNO_3$–$NH_3$ nucleation could be even higher to some extent, because the limitation of $H_2SO_4$ availability may not be as strong.

In the upper troposphere above the Pacific and Atlantic oceans, accurate large-scale $NH_3$ observations are unavailable; therefore, we evaluated the uncertainty associated with $NH_3$ from the perspective of emission uncertainty. In our model, the $NH_3$ concentrations were prescribed using the outputs of CAM-Chem[100], which were further driven by anthropogenic and marine $NH_3$ emission inputs. The uncertainty in anthropogenic $NH_3$ emissions has been estimated to be 125% at the global level and 186–294% at the regional level[117,122]. The marine $NH_3$ emissions used in the CAM-Chem simulation were obtained from Bouwman et al.[123], who reported a marine emission estimate of 8 TgN year$^{-1}$ associated with an uncertainty range of a factor 2–3, although the relative source distribution might be more reliable than the absolute emissions[123]. Most other studies (for example, Fowler et al.[124], Paulot et al.[125] and references therein) estimated marine $NH_3$ emissions at 5.6–13 TgN year$^{-1}$, with the most extreme estimates of 2.5 and 23 TgN year$^{-1}$. Therefore, the emission estimate of Bouwman et al.[123] with uncertainty of a factor 3 broadly encompasses the range of values reported in related studies. Considering the above-mentioned uncertainties in anthropogenic and marine $NH_3$ emissions, as well as the linear relationship between $NH_3$ emissions and concentrations, we designed a sensitivity experiment ('0.33*$NH_3$') by reducing $NH_3$ concentrations in the best-case simulation by a factor of 3. Similar to the discussion on organic-mediated nucleation, we did not test higher $NH_3$ concentrations because they would not challenge the important roles of $H_2SO_4$–$HNO_3$–$NH_3$ nucleation over the regions of interest. Extended Data Figs. 7 and 8 indicate that, even with low $NH_3$ concentration, in the sensitivity experiment, $H_2SO_4$–$HNO_3$–$NH_3$ nucleation remains one of the two primary NPF mechanisms in the troposphere over the Pacific and Atlantic oceans, although its contribution would be smaller than that of organic–$H_2SO_4$ nucleation.

$HNO_3$ is a key precursor involved in $H_2SO_4$–$HNO_3$–$NH_3$ nucleation, which our study suggests is an important and often dominant NPF mechanism in the upper troposphere above the Asian monsoon region. Here we evaluate the simulated concentrations of $HNO_3$ against observations from the Microwave Limb Sounder (MLS) aboard the Aura satellite[126]. We used the level 3 monthly $HNO_3$ product for the

evaluation[127]. Supplementary Fig. 10 shows simulated and observed 2016 mean $HNO_3$ concentrations at 150 hPa (approximately 13 km), corresponding broadly to the location with the highest NPF rate in our model. It also represents one of the few vertical levels provided by the Aura MLS level 3 product within the upper troposphere, at which NPF rates are notably high. The simulations generally agree well with observations, with average concentrations of 0.562 and 0.544 ppb, respectively, over the Asian monsoon region (8°–40° N, 60°–130° E). Given that the $H_2SO_4$–$HNO_3$–$NH_3$ nucleation rate has relatively weak (quadratic) dependence on $HNO_3$ concentration[15], the uncertainty in $HNO_3$ concentration should not change our conclusion about the role of $H_2SO_4$–$HNO_3$–$NH_3$ nucleation in the upper troposphere above the Asian monsoon region.

$HIO_3$ is the key driver of iodine oxoacids nucleation, which has been shown by our study to be the dominant NPF mechanism near the surface over oceans. The uncertainties in emissions, gas-phase chemistry, wet deposition, aerosol uptake and cycling of iodine species all contribute to the uncertainty in the simulated $HIO_3$ concentrations. Because the simulated $HIO_3$ concentrations vary between 80% below and 100% above the observed values, we designed two sensitivity experiments ('0.5*$HIO_3$' and '5*$HIO_3$') that set $HIO_3$ concentrations to 0.5 and 5 times those of the best-case simulation results. Again, this across-the-board reduction or increase probably represents an extreme case because the biases at different locations vary between the two bounds. Extended Data Figs. 7 and 8 show that, in both sensitivity simulations, iodine oxoacids nucleation remains the dominant nucleation mechanism near the surface of the Pacific and Atlantic oceans.

Atmospheric dynamics and transport may be a source of uncertainty for the precursor concentrations and NPF mechanisms in the upper atmosphere. The transport of precursors to the upper troposphere occurs by means of two main processes in the model: large-scale transport resolved by the model grids and unresolved convective transport that must be parameterized. To accurately simulate large-scale transport, as described above, we nudged the wind fields to the MERRA-2 reanalysis data, following Sun et al.[102]. Reanalysis data represent a blend of observations and weather simulations realized through data assimilation and they provide the most accurate and complete picture of past weather and climate that is available at present. Sun et al.[102] showed that nudging E3SM simulations to the MERRA-2 reanalysis data could produce simulated grid-scale winds that closely resemble those of MERRA-2, with spatial and temporal correlations >0.9 for both the lower and the upper troposphere. Therefore, we believe that the simulation of large-scale transport, based on the best available and highly mature method, should not cause large uncertainty in precursor concentrations and NPF mechanisms.

The simulation of subgrid convective transport is comparatively uncertain because of the difficulty in directly applying observational constraints. In the E3SM model, deep convection and the associated convective transport were simulated using the Zhang and McFarlane (ZM) convection scheme[128]. We used an improved version, as described by Wang et al.[85,98], which uses a unified treatment of the convective transport of aerosols and gases. To understand the uncertainty associated with the deep convection scheme, Qian et al.[129] performed numerous sensitivity simulations in which seven main parameters of the ZM convection scheme and certain other parameters in the E3SM were perturbed simultaneously within their possible ranges using the Latin hypercube sampling method. They showed that more than 85% of the total variance of precipitation, an indicator of convection development that controls convective transport, could generally be explained by two out of the seven ZM parameters: (1) the timescale for the consumption rate of convective available potential energy (hereafter, denoted by 'tau') and (2) the fractional mass entrainment rate (hereafter, denoted by 'dmpdz'), defined as the fractional air mass flux entrained into a volume of cloudy air per unit height[130]. Yang et al.[131] reached almost the same conclusion about the governing

parameters for the ZM convection scheme implemented in the Community Atmosphere Model version 5. To better address the reviewer's concern, we further performed two sensitivity simulations ('upper_tau' and 'lower_tau') that set the value of tau to the upper and lower bounds (14,400 and 1,800 s, respectively) of the possible ranges specified by Qian et al.[129], as compared with the optimized value of 3,600 s in the best-case simulation. Similarly, we conducted two sensitivity simulations for dmpdz ('upper_dmpdz' and 'lower_dmpdz'), which changed the value of dmpdz from $0.7 \times 10^{-3}\,\mathrm{m}^{-1}$ in the best-case simulation to $2.0 \times 10^{-3}\,\mathrm{m}^{-1}$ and $0.1 \times 10^{-3}\,\mathrm{m}^{-1}$, respectively. Supplementary Figs. 6–9 summarize the contributions of different NPF mechanisms in these sensitivity simulations over the main regions of interest in this study, including rainforests, anthropogenically polluted regions and the Pacific and Atlantic oceans. The results indicate that the sensitivity scenarios have limited influence on the relative contributions of individual NPF mechanisms in the upper troposphere over these regions, largely because the perturbation of convective transport simultaneously changes the concentrations of several nucleation precursors, which subsequently causes simultaneous change in the rates of most NPF mechanisms.

## Further discussion about NPF mechanisms in different regions

Figure 1 shows that, near the surface of rainforests, the NPF rates are low in the Amazon but are high in Southeastern Asia and moderate in Central Africa. The low NPF rate in the Amazon is because of low $SO_2/H_2SO_4$ and high temperatures that cause low organic nucleation rates; this is consistent with observations showing rare NPF events in the pristine Amazon boundary layer[132,133]. The high surface NPF rate in Southeastern Asia is driven by iodine oxoacids nucleation (typical of oceanic regions) and amine–$H_2SO_4$ nucleation (typical of anthropogenically polluted regions), as shown in Fig. 1b. This is because Southeastern Asia is affected by not only biogenic emissions but also oceanic emissions of iodine species and anthropogenic emissions of $SO_2$ and amines, and thus it possesses some NPF features typical of oceanic and polluted regions. Central Africa is more affected by anthropogenic and oceanic emissions than the Amazon but is less affected than Southeastern Asia, leading to the moderate NPF rate there.

In the upper troposphere of all three rainforest regions, pure-organic nucleation is the dominant NPF mechanism because its rate is greatly enhanced at low temperatures owing to the marked decrease of volatility and increase of cluster stability. It is noted that nucleation of $H_2SO_4$ and $NH_3$ contributes <2% of the NPF rate in the upper troposphere of the Amazon but contributes a larger fraction of 10–15% in the upper troposphere of Southeastern Asia owing to more abundant $H_2SO_4$ from anthropogenic sources. However, according to our model, this mechanism still cannot compete with organic-mediated nucleation.

Moreover, our results show that $H_2SO_4$–$HNO_3$–$NH_3$ nucleation is important in the upper troposphere over the Asian monsoon region (spanning China and India; see Fig. 2b) but not over other regions with notable anthropogenic pollution such as Southeastern Asia. This is mainly because of the much larger $NH_3$ concentration in the former than the latter. Observations (Figs. 4 and 5 of Höpfner et al.[42] and Supplementary Fig. 5 of Höpfner et al.[43]) and our simulation (Extended Data Fig. 4) both revealed notable hotspots of $NH_3$ concentrations (10–40 ppt, occasionally >60 ppt) in the upper troposphere of the Asian monsoon region in summer but not over other regions or during different seasons. This is because of the following: (1) $NH_3$ emissions in China and India (23.0 Mt in 2014 (ref. 103)) are much larger than those in Southeastern Asia (4.7 Mt in 2014 (ref. 103)) and (2) the Asian summer monsoon is especially favourable for upward transport of $NH_3$ to the heights of interest. For similar reasons, the mean $H_2SO_4$ concentrations in the upper troposphere of the Asian monsoon region (0.2–0.5 ppt) are also larger than those over Southeastern Asia (0.04–0.10 ppt), as shown in Supplementary Figs. 2 and 3, further contributing to the high $H_2SO_4$–$HNO_3$–$NH_3$ nucleation rate over the Asian monsoon region.

## Further results and discussion about the NPF contribution to particles and CCN

Figure 5 and Extended Data Fig. 2 show the fractions of particles and CCN caused by NPF at three representative heights: 13 km above ground level (AGL; the upper troposphere), 1 km AGL (approximately at the low-cloud level) and at the surface. Although supersaturation levels vary under different cloudy conditions, our analysis primarily focuses on CCN0.5% to evaluate the potential impact of NPF. In the upper troposphere, NPF contributes more than 90% of the particle number concentration and 60–85% of the CCN0.5% concentration in most regions; the absolute contribution in terms of numbers per cubic centimetre is highest over the rainforests and in the Asian monsoon regions because of the regionally highest NPF rates. The results in the lower troposphere (from the surface to the low-cloud level) have been described in the main text. It is noted that, in the lower troposphere over rainforests, the fractions of particles and CCN0.5% from NPF vary greatly with location, depending inversely on the strength of biomass-burning emissions. This is consistent with previous regional simulation results that suggested that NPF contributes most of the particles and the CCN in the Amazon boundary layer in the pristine wet season but contributes only a small fraction in the dry season when wildfires prevail[134].

Supplementary Fig. 12 further shows that, from a zonal mean perspective, NPF generally contributes more than 90% of the particle number concentration above 3–4 km in the troposphere and 50–90% below 3–4 km. The contributions of NPF to CCN0.5% are usually more than 60% above 3–4 km in the troposphere and 30–60% below 3–4 km.

## Discussion in the context of previous global models

The understanding of atmospheric NPF mechanisms and the representations of these mechanisms in global models have been advancing rapidly. The binary nucleation involving $H_2SO_4$ and $H_2O$ was the earliest identified NPF mechanism and thus the NPF processes in most early global models were parameterized from the classical nucleation theory for $H_2SO_4$–$H_2O$ nucleation[1,135]. $NH_3$ and ions were later found to accelerate $H_2SO_4$–$H_2O$ nucleation by stabilizing the $H_2SO_4$–$H_2O$ clusters. Hence many global modelling studies have incorporated parameterizations of $H_2SO_4$–$NH_3$–$H_2O$ neutral nucleation as well as $H_2SO_4$–$H_2O$ or $H_2SO_4$–$NH_3$–$H_2O$ ion-induced nucleation developed on the basis of either classical nucleation theory or kinetic nucleation models[10,136–138]. These modelling studies showed varied performance in reproducing atmospheric particle number observations, mainly because of the uncertainty in the theoretical calculations of the NPF rate. Recent box-model estimates or three-dimensional simulations[3,7,12,13] using experiment-based parameterizations have shown that the mechanisms involving $H_2SO_4$, $NH_3$ and $H_2O$ fail to explain observed NPF rates or particle numbers in wide-ranging atmospheric environments, including those in the boundary layer and the upper troposphere; the underestimation often occurs by an order of magnitude or more.

To better reproduce observed NPF rates in the boundary layer, some global modelling studies used empirical nucleation parameterizations, which simply assumed NPF rates to be proportional to the $H_2SO_4$ concentration raised to the power of one to two with a rate coefficient tuned on the basis of observational data in certain regions of the continental boundary layer[37,90,92,137,139–142]. These parameterizations were further classified as the activation nucleation parameterization[143], for which the NPF rate is linearly related to $H_2SO_4$ concentration and the kinetic nucleation parameterization[144], for which the NPF rate has a quadratic relationship with $H_2SO_4$ concentration. These parameterizations have shown some success in reproducing observed NPF rates and particle number concentrations in the continental boundary layer, especially over regions similar to those for which the parameterizations were derived[90,92,141,142]. However, they often fail to reproduce the particle number concentrations in many other regions; for example, they have been shown to substantially overpredict particle number

concentrations over the oceans and above the boundary layer[1,90,140], which limits their range of application. Moreover, the actual chemical mechanism of NPF cannot be clarified with these parameterizations owing to their empirical nature. Here we conducted a sensitivity simulation ('NPF_Mech4_scaled') following the principles of the empirical nucleation parameterizations. In this simulation, we applied a fixed scaling factor to the NPF rates in NPF_Mech4 (which only includes four traditional nucleation mechanisms involving $H_2SO_4$, $NH_3$ and $H_2O$) such that its globally averaged NPF rate matched that of the best-case simulation. Here the globally averaged NPF rate is defined as the average of the NPF rates across all model grid boxes (both horizontally and vertically), weighted by the volumes of those grid boxes. Extended Data Fig. 10 illustrates the zonal mean NPF rates in the best-case and 'NPF_Mech4_scaled' scenarios. Despite using the same globally averaged NPF rates in both scenarios, the NPF rates for specific regions and altitudes differ greatly. Moreover, because the dominant NPF mechanism differs between the two scenarios, the NPF rate would respond differently to the perturbation of precursor concentrations in the future.

Recent studies have revealed that ULVOCs and ELVOCs can trigger NPF either with or without $H_2SO_4$ (refs. 20,21). Several global modelling studies have considered organic-mediated nucleation, based on parameterizations either derived from laboratory experiments or empirically fitted using field measurements of NPF rates and proxies for nucleating organics[9,22,24,37,58,92,93,145]. Zhu et al.[25] simulated organic-mediated nucleation in the CESM/IMPACT model using a different approach, which assumed that only a few compounds (diacyl peroxide, pinic acid, pinanediol and selected oxidation products of pinanediol) drive nucleation and thus tracked the formation of these molecules. Most of these studies showed improved model performance against observed NPF rates or particle number concentration[24,58,92,93]. However, these studies consistently simplified organic-mediated nucleation by assuming either that the nucleating organics represent a fixed fraction of all oxidation products[9,22,24,37,58,92,93,145] or that only a few individual molecules are involved in nucleation[25]. This is by sharp contrast to the latest understanding that organic-mediated nucleation is driven by a large variety of ULVOC and ELVOC species[20,26,27], whose yields vary by several orders of magnitude depending on temperature and $NO_x$ concentration[26,28,29]. Evaluation against observations over the Amazon has shown that the 'fixed-fraction' assumption leads to unrealistically low particle number concentrations in the upper troposphere and unrealistically high concentrations in the boundary layer[13].

Two global modelling studies evaluated the potential impact of individual nucleation mechanisms that have recently received attention, that is, amine–$H_2SO_4$ nucleation[18] and $HNO_3$–$H_2SO_4$–$NH_3$ nucleation[15]. Bergman et al.[18] incorporated amine–$H_2SO_4$ nucleation in the ECHAM-HAMMOZ global model and found that amines might substantially enhance nucleation in terrestrial regions, but their model predicted unrealistically high NPF rates (approximately 1,000 $cm^{-3}$ $s^{-1}$) compared with typical observations in some notable polluted regions. Wang et al.[15] investigated the potential role of $HNO_3$–$H_2SO_4$–$NH_3$ nucleation in upper-tropospheric particle formation based on chamber experiments conducted in that study. Nevertheless, these studies did not consider most other important nucleation mechanisms and thus could not clarify the relative importance of different mechanisms over various regions.

Different from most earlier modelling studies that included only one or two NPF mechanisms, Dunne et al.[53] and Gordon et al.[9] presented by far the most systematic global modelling study by integrating NPF mechanisms involving $H_2SO_4$, $NH_3$ and organics in a single model. They used parameterizations of $H_2SO_4$–$H_2O$ and $H_2SO_4$–$NH_3$–$H_2O$ neutral and ion-induced nucleation based on CLOUD chamber experiments under wide temperature ranges, effectively reducing the uncertainty of earlier studies using theoretically derived nucleation parameterizations. They also included organic-mediated nucleation but they simplistically assumed a fixed fraction of all oxidation products to drive nucleation, as described above. Their simulations revealed that nearly all nucleation throughout the present-day atmosphere involves ammonia or biogenic organic compounds, as well as $H_2SO_4$. However, essentially all organic-mediated nucleation in their model occurred in the lower troposphere, contrary to the latest understanding[13,25], probably because their simplified 'fixed-fraction' approach produced too many particles at low altitudes and too few particles at high altitudes. Besides, they did not consider some crucial nucleation mechanisms such as iodine oxoacids nucleation and $HNO_3$–$H_2SO_4$–$NH_3$ nucleation that were speculated to prevail in certain regions with either high particle number concentrations or large aerosol–cloud radiative forcing.

In this study, we synthesized the latest laboratory experiments to develop comprehensive model representations of NPF and the chemical transformation of precursor gases in a fully coupled global climate model. The main advances compared with previous modelling studies consisted of the following. (1) The model comprehensively included 11 nucleation mechanisms based on the latest information available from laboratory experiments, covering practically all those mechanisms thought to be atmospherically relevant at present. Among the 11 mechanisms, four crucial ones were usually overlooked in previous global models, including iodine oxoacids neutral and ion-induced nucleation, synergistic $H_2SO_4$–$HNO_3$–$NH_3$ nucleation and amine–$H_2SO_4$ nucleation. (2) We transformed previous global model representations of pure-organic and organic–$H_2SO_4$ nucleation by implementing in the model an advanced, experimentally constrained R2D-VBS to simulate the temperature-dependent and $NO_x$-dependent formation chemistry and thermodynamics of ULVOCs and ELVOCs. (3) We incorporated systematic treatment of the sources and sinks of iodine oxoacids, including precursor emissions, detailed gas-phase reactions, particle uptake and recycling, thereby facilitating reasonable simulation of iodine oxoacids nucleation. (4) We assessed the sensitivity of the results to the most important known uncertainties in a reasonably comprehensive manner. The new model, although inevitably still bearing uncertainties, greatly improves the simulation of particle number concentrations over the world's particle hotspots and allows explanation of worldwide NPF mechanisms that vary greatly with region and altitude, which have important implications for accurate estimation of both aerosol radiative forcing and anthropogenic effects on climate.

## Data availability

The observational data over the Amazon from the ACRIDICON-CHUVA and GoAmazon campaigns are publicly available at https://www.arm.gov/research/campaigns/amf2014goamazon and https://halo-db.pa.op.dlr.de/mission/5. The observational data over the Pacific and Atlantic oceans, acquired during the ATom campaign, are publicly available at https://daac.ornl.gov/ATOM/campaign/. Other relevant data are available at https://figshare.com/s/71bf2a48657a2f5deb76. Model outputs were processed and Figs. 3b, 4 and 5 and Extended Data Figs. 2, 4, 7, 8 and 10 were plotted using the NCAR Command Language (version 6.6.2), https://doi.org/10.5065/D6WD3XH5. Source data are provided with this paper.

## Code availability

The model codes developed in this work as well as sample datasets used to test the codes are available at https://figshare.com/s/71bf2a48657a2f5deb76.

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

**Acknowledgements** This study was supported by the US Department of Energy, Office of Science, Office of Biological and Environmental Research, Earth System Model Development (ESMD) programme area as part of the Enabling Aerosol–cloud interactions at GLobal convection-permitting scalES (EAGLES) project (project no. 74358), and by the National Natural Science Foundation of China (22188102). This study was also supported by the National Natural Science Foundation of China (42275110) and the National Key R&D Program of China (2022YFC3701000, Task 5). N.M.D. was supported by the National Science Foundation (AGS2132089). M.S. was supported by the US Department of Energy Office of Science, Office of Biological and Environmental Research, through the Early Career Research Program. This research used the resources of the National Energy Research Scientific Computing Center (NERSC), which is a US Department of Energy Office of Science User Facility located at Lawrence Berkeley National Laboratory, operated under contract no. DE-AC02-05CH11231 using NERSC awards ALCC-ERCAP0016315, BER-ERCAP0015329, BER-ERCAP0018473 and BER-ERCAP0020990. The Pacific Northwest National Laboratory is operated for the US Department of Energy by Battelle Memorial Institute under contract DE-AC05-76RL01830.

**Author contributions** B.Z., N.M.D., P.-L.M., M.S. and K.Z. conceived the study. B.Z., K.Z., N.M.D., M.S., J.Sh. and C.Y. developed the methods used in this study. B.Z., L.M., K.Z., N.M.D., M.S., J.Sh., J.Su., S.T., H.G., B.S., Z.L., L.H., S.L., G.L., Y.G., J.J., A.D., W.N., X.Q., X.C. and L.W. carried out the research. B.Z. wrote the original draft of the manuscript. N.M.D., Y.G., M.S., K.Z., J.Su., S.W., P.-L.M., S.T., J.F., H.G., Z.L., M.W., S.L., C.Y., G.L., H.W., J.J., A.D., W.N. and L.W. revised the manuscript.

**Competing interests** The authors declare no competing interests.

**Additional information**
**Correspondence and requests for materials** should be addressed to Bin Zhao.

A

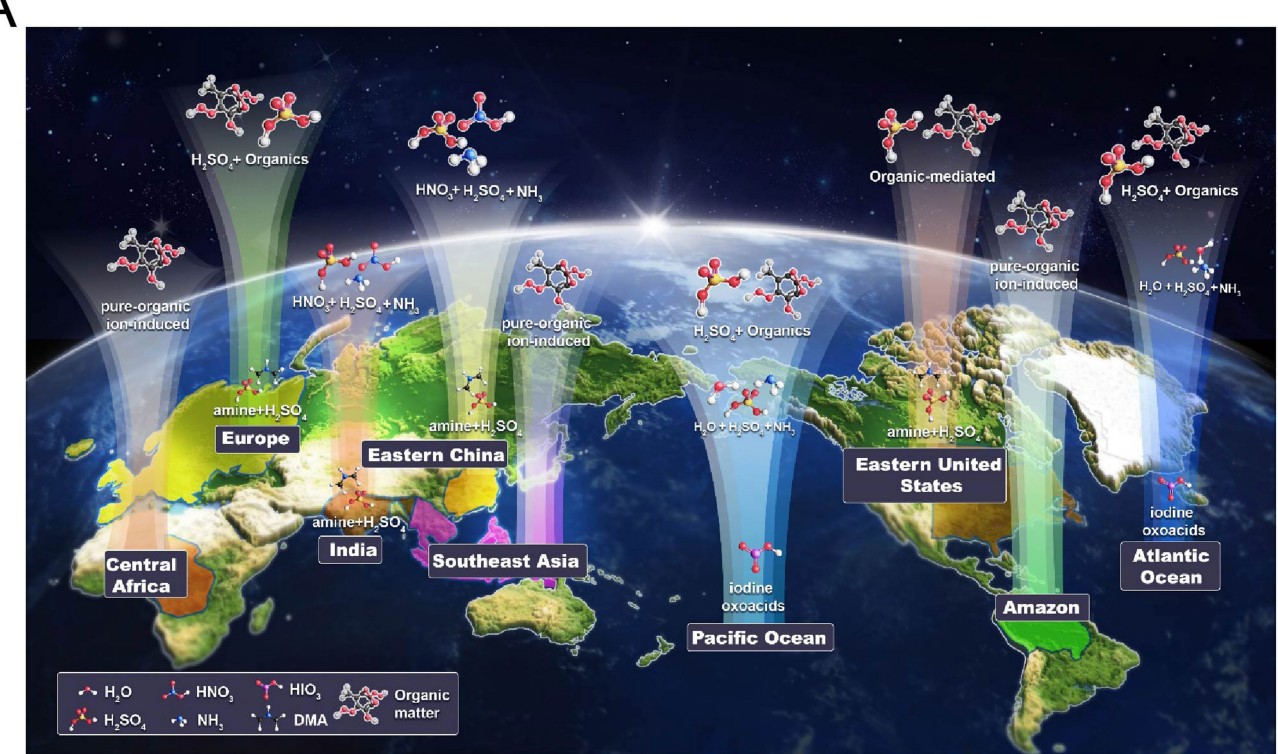

B

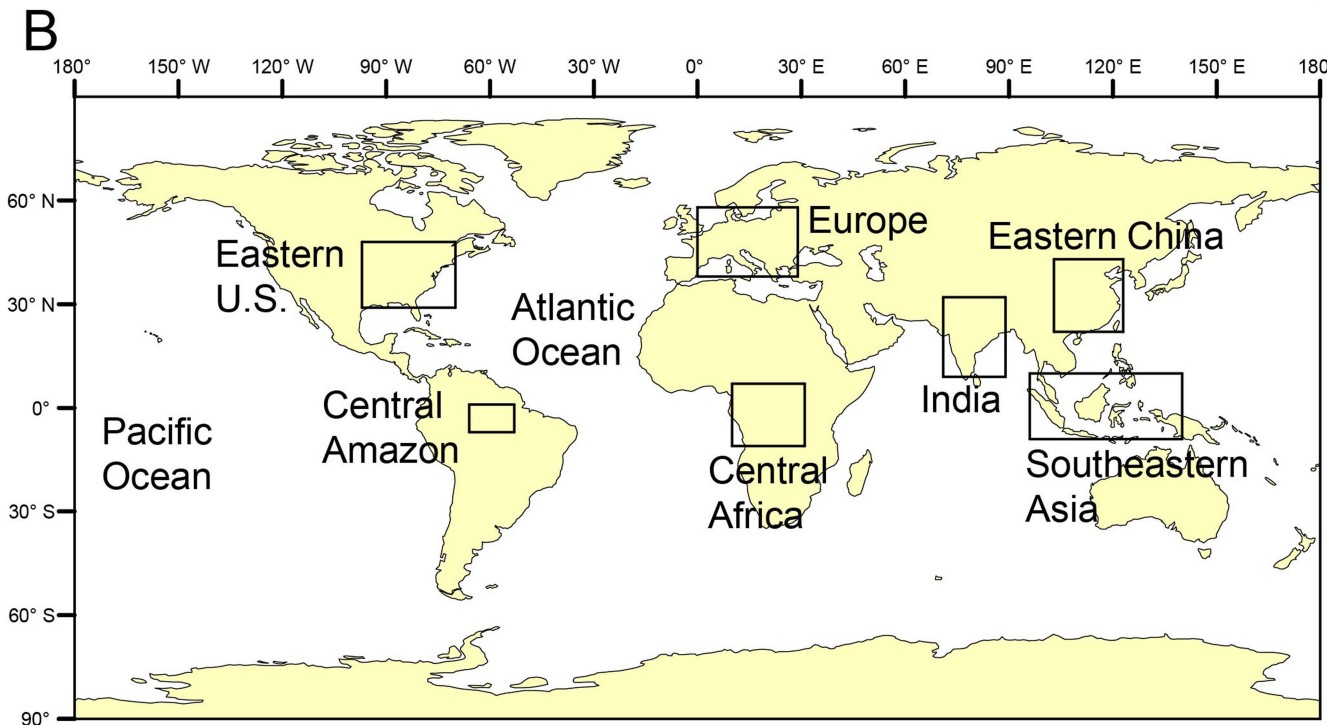

**Extended Data Fig. 1 | Schematic of regionally leading NPF mechanisms and spatial extent for quantitative analyses. a**, Schematic of the leading NPF mechanisms in the boundary layer and upper troposphere of the regions of interest. Note, in the upper troposphere above the Pacific and Atlantic oceans, organic–$H_2SO_4$ nucleation and $H_2SO_4$–$NH_3$–$H_2O$ neutral nucleation are identified as two primary NPF mechanisms. **b**, Spatial extent of the regions of interest used in our quantitative analyses. Panel **b** was created with ArcGIS 10 using free map data made by Natural Earth (https://www.naturalearthdata.com/).

## Particle number concentration

## Fraction of particle number from NPF

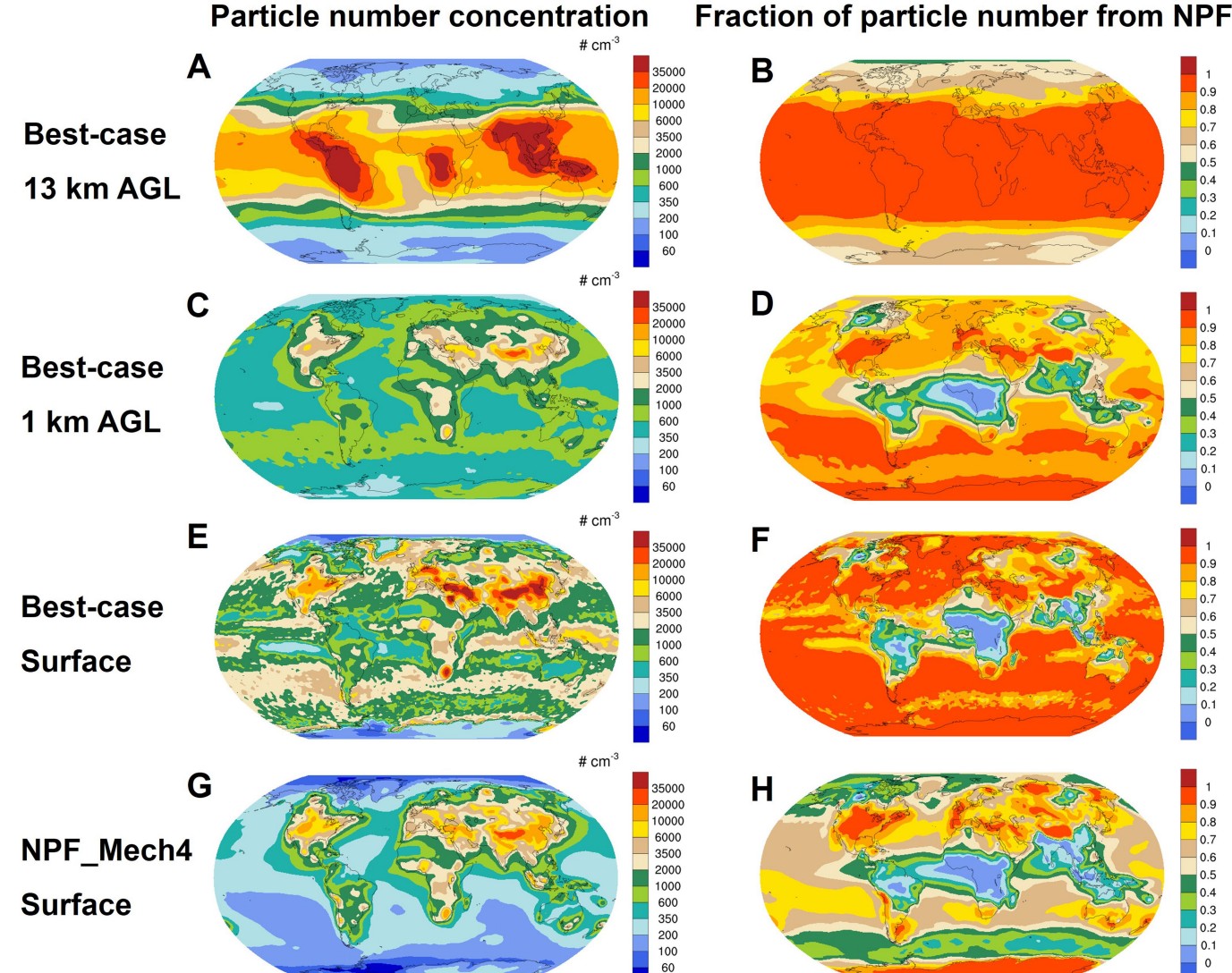

**Best-case 13 km AGL**

**Best-case 1 km AGL**

**Best-case Surface**

**NPF_Mech4 Surface**

**Extended Data Fig. 2 | Number concentrations of particles across the entire size range and the fractions caused by NPF at different vertical levels in 2016. a,c,e,** Spatial distribution of particle number concentrations at 13 km AGL (**a**), 1 km AGL (approximately at the low-cloud level) (**c**) and surface level from the best-case simulation (**e**). **b,d,f,** Fractions of particle number concentrations from NPF at 13 km AGL (**b**), 1 km AGL (**d**) and surface level, based on the difference between the best-case and No_NPF scenarios (**f**). **g,** Spatial distribution of particle number concentrations at surface level from the NPF_Mech4 scenario. **h,** Fractions of particle number concentrations from NPF at surface level, based on the difference between the NPF_Mech4 and No_NPF scenarios. Definitions of the scenarios are presented in Methods and Supplementary Table 1. Particle number concentrations cover the entire size range (note that field observations are mostly made for particles larger than a certain cutoff size) and are normalized to standard temperature and pressure (273.15 K and 101.325 kPa). The zonal mean particle number concentrations and the fractions caused by NPF are presented in Supplementary Fig. 12. Maps were created using the NCAR Command Language (version 6.6.2), https://doi.org/10.5065/D6WD3XH5.

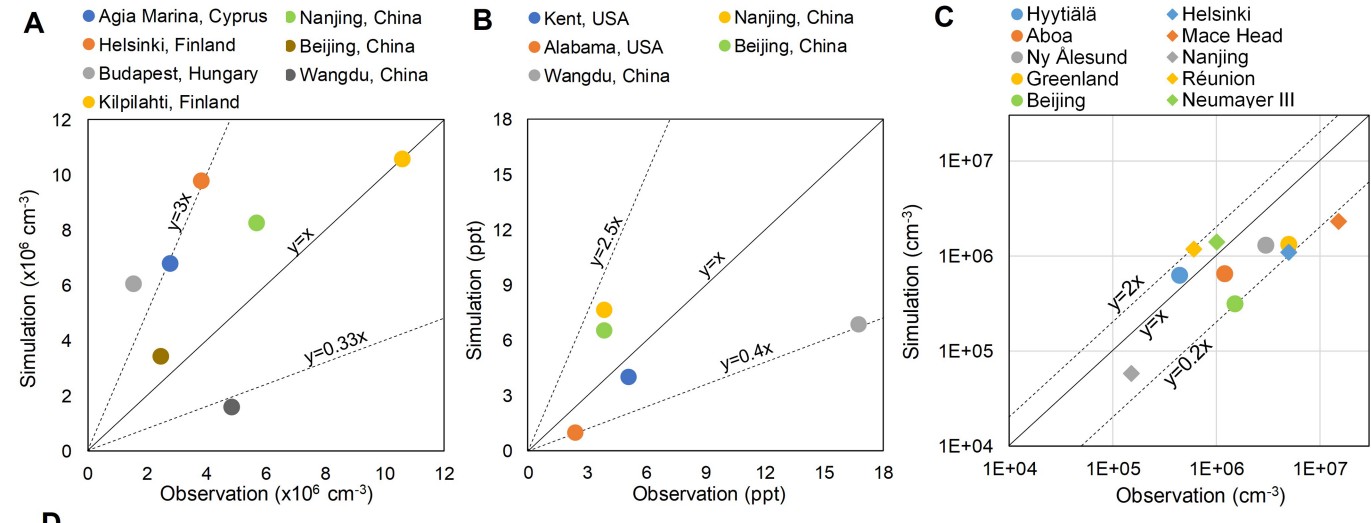

| Site | Observation | Simulated number concentrations (# cm$^{-3}$) | | | Normalized bias | | |
|------|-------------|--------|-----------|-----------|--------|----------|-----------|
| | | No_NPF | NPF_Mech4 | Best-case | No_NPF | NPF_Mech4 | Best-case |
| BUCT | 25178 | 4929 | 8833 | 27739 | −80% | −65% | 10% |
| SORPES | 13862 | 6116 | 7957 | 17545 | −56% | −43% | 27% |
| Wangdu | 34854 | 10481 | 12362 | 23298 | −70% | −65% | −33% |
| Average | 24631 | 7175 | 9717 | 22861 | −71% | −61% | −7% |

**Extended Data Fig. 3 | Further evaluation of model performance at observational sites over oceanic and human-polluted continental regions.** Comparison of simulated H$_2$SO$_4$ (**a**) and DMA (**b**) concentrations with observations in anthropogenically polluted regions. We list the time ranges and sources of the observational data as follows. **a**, H$_2$SO$_4$: Agia Marina, Cyprus, February 2018, Dada et al.[146]; Helsinki, Finland, July 2019, Dada et al.[146]; Budapest, Hungary, March–April 2018, Dada et al.[146]; Kilpilahti, Finland, June 2012, Dada et al.[146]; Nanjing, China, January, April, July, November 2018, Yang et al.[147]; Beijing, China, January–April and October–December 2018, Deng et al.[148]; Wangdu, China, December 2018 and January 2019, Wang et al.[149]. **b**, DMA: Kent,

USA, November 2011 and August–September 2013, You et al.[150]; Alabama, USA, June 2013, You et al.[150]; Wangdu, China, December 2018 and January 2019, Wang et al.[149]; Nanjing, China, August–September 2012, Zheng et al.[151]; Beijing, China, January–March and October–December 2018, Cai et al.[4]. **c**, Comparison of simulated HIO$_3$ concentrations with observations at ten oceanic or coastal sites worldwide. Simulated concentrations are averaged over the nine model grids encompassing an observational site. Locations of the sites and time periods of observations are summarized in He et al.[55]. **d**, Statistics of simulated and observed number concentrations of ultrafine particles (diameter <100 nm) at three observational sites in China.

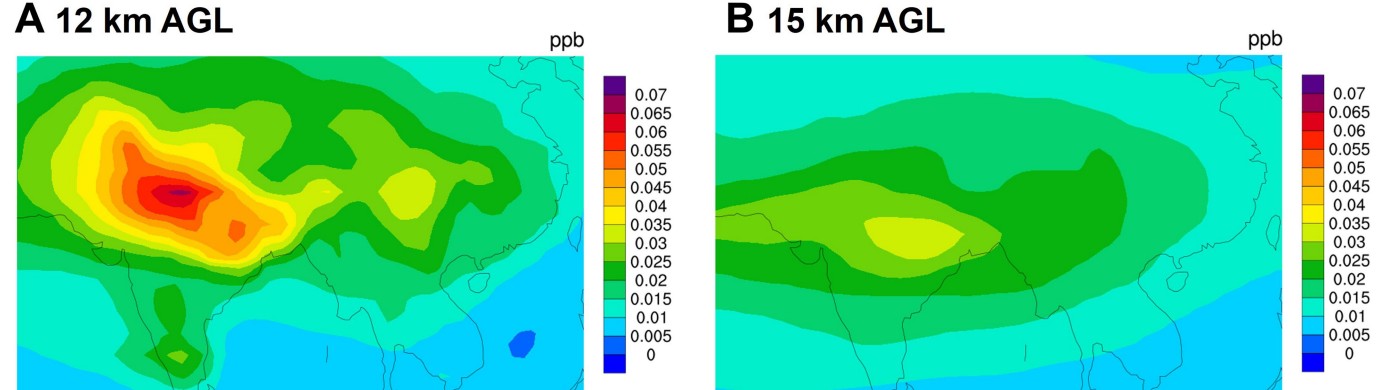

**Extended Data Fig. 4 | Simulated NH₃ concentrations in the upper troposphere over the Asian monsoon region in June–August 2016.** Concentrations at 12 and 15 km AGL are shown to facilitate comparison with satellite observations reported by Höpfner et al.[42] (their Fig. 5) and Höpfner et al.[43] (their Supplementary Fig. 5).

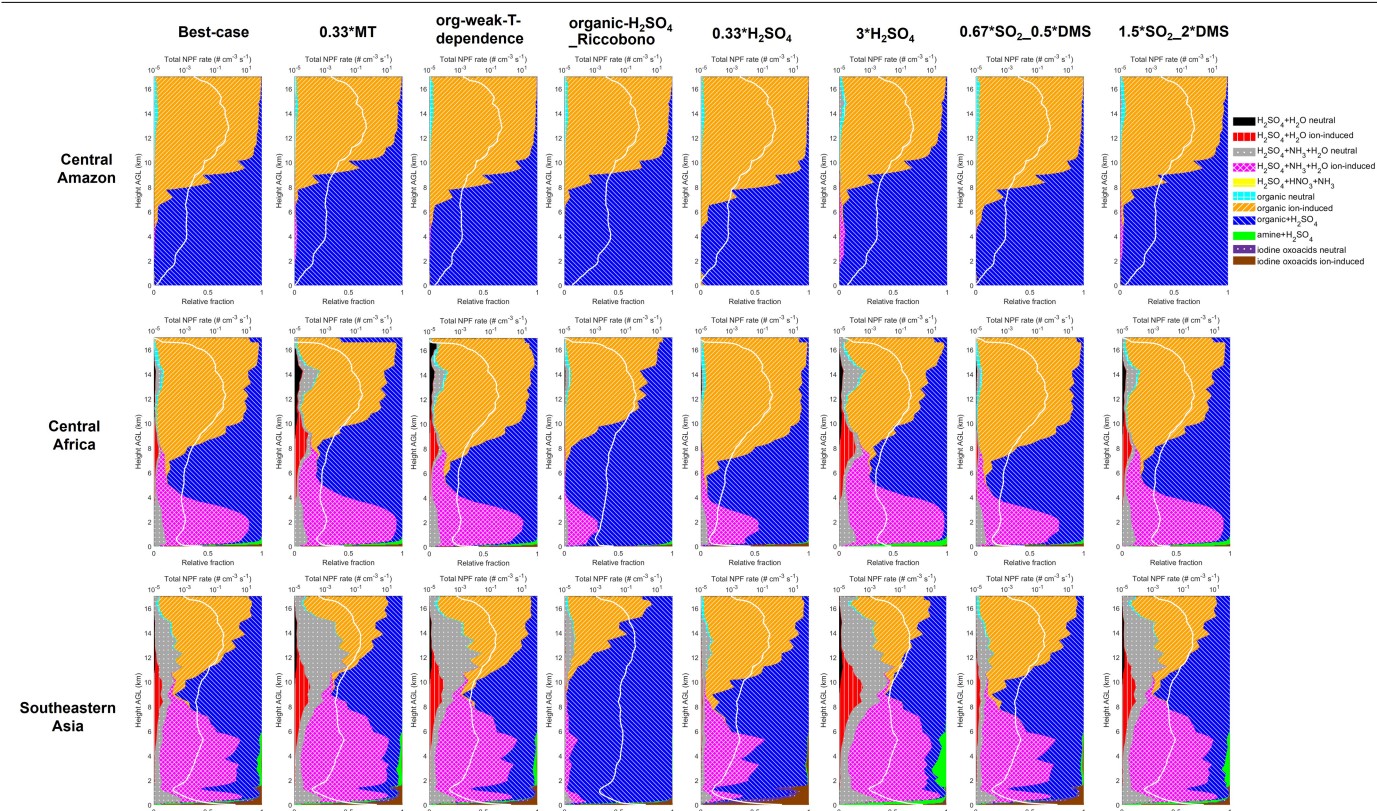

**Extended Data Fig. 5 | NPF rates as a function of height AGL over rainforests under the best-case and sensitivity scenarios.** White lines represent the total NPF rates of all mechanisms at a diameter of 1.7 nm ($J_{1.7}$, on a log scale) and the coloured areas represent the relative contributions of different mechanisms, both averaged in 2016 over the regions specified in Extended Data Fig. 1b. Definitions of the sensitivity experiments are presented in Methods and Supplementary Table 1.

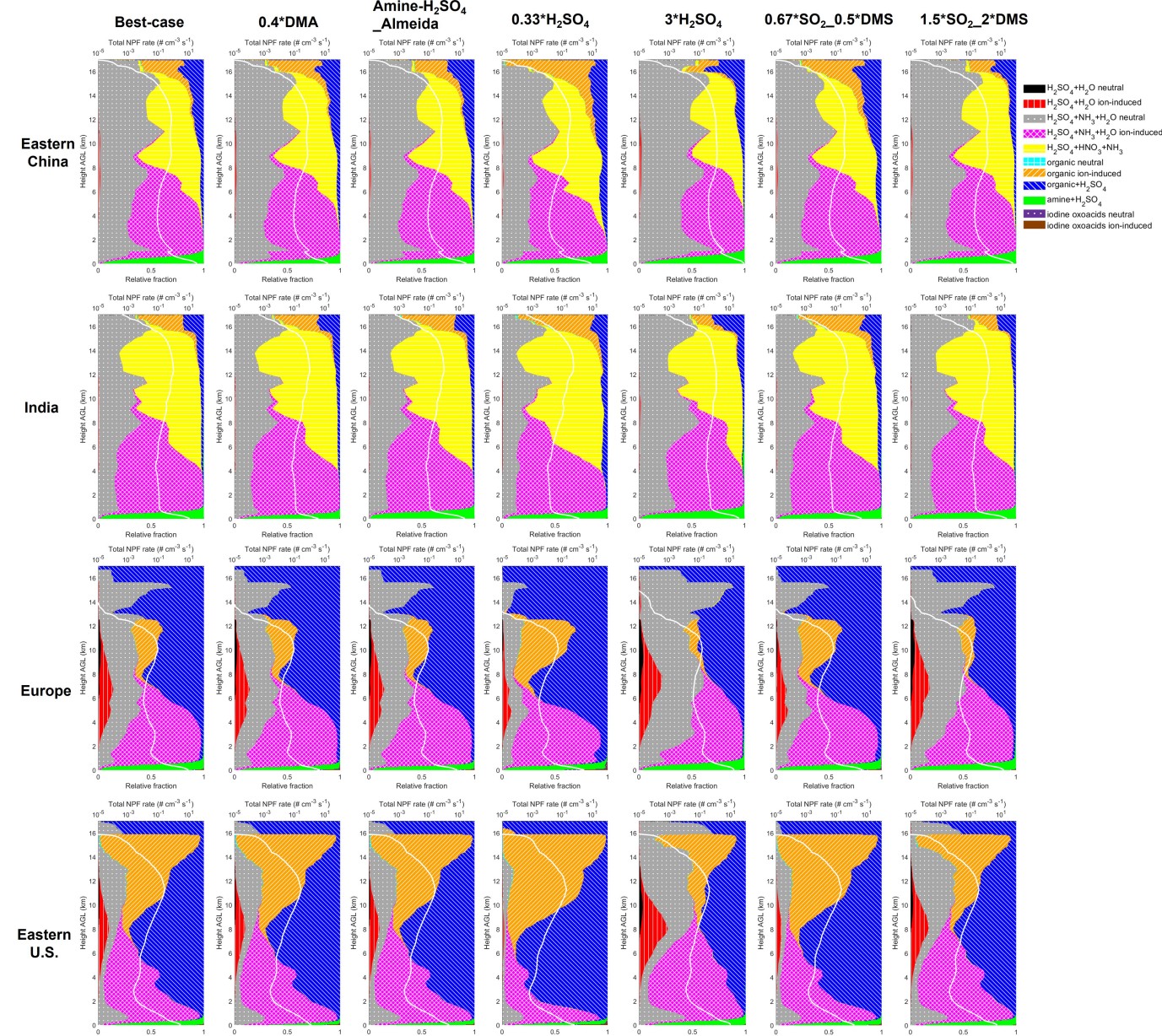

**Extended Data Fig. 6 | NPF rates as a function of height AGL over anthropogenically polluted regions under the best-case and sensitivity scenarios.** White lines represent the total NPF rates of all mechanisms at a diameter of 1.7 nm ($J_{1.7}$, on a log scale) and the coloured areas represent the relative contributions of different mechanisms, both averaged in 2016 over the regions specified in Extended Data Fig. 1b. Definitions of the sensitivity experiments are presented in Methods and Supplementary Table 1.

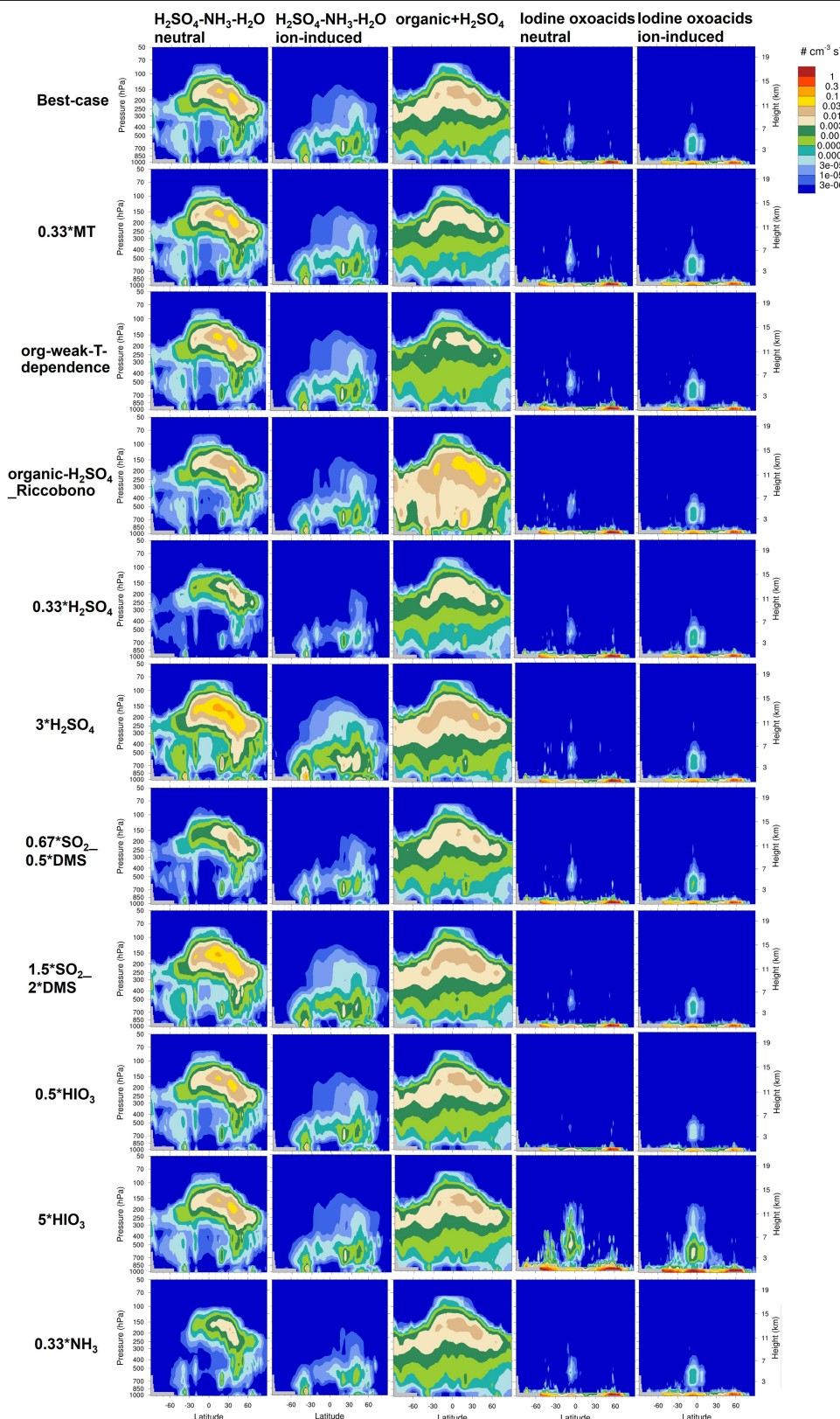

**Extended Data Fig. 7 | Zonal mean NPF rates of individual mechanisms over the Pacific Ocean (170° E–150° W) under the best-case and sensitivity scenarios in 2016.** Only five NPF mechanisms are shown because the other mechanisms are negligible in these regions. Definitions of the sensitivity experiments are presented in Methods and Supplementary Table 1.

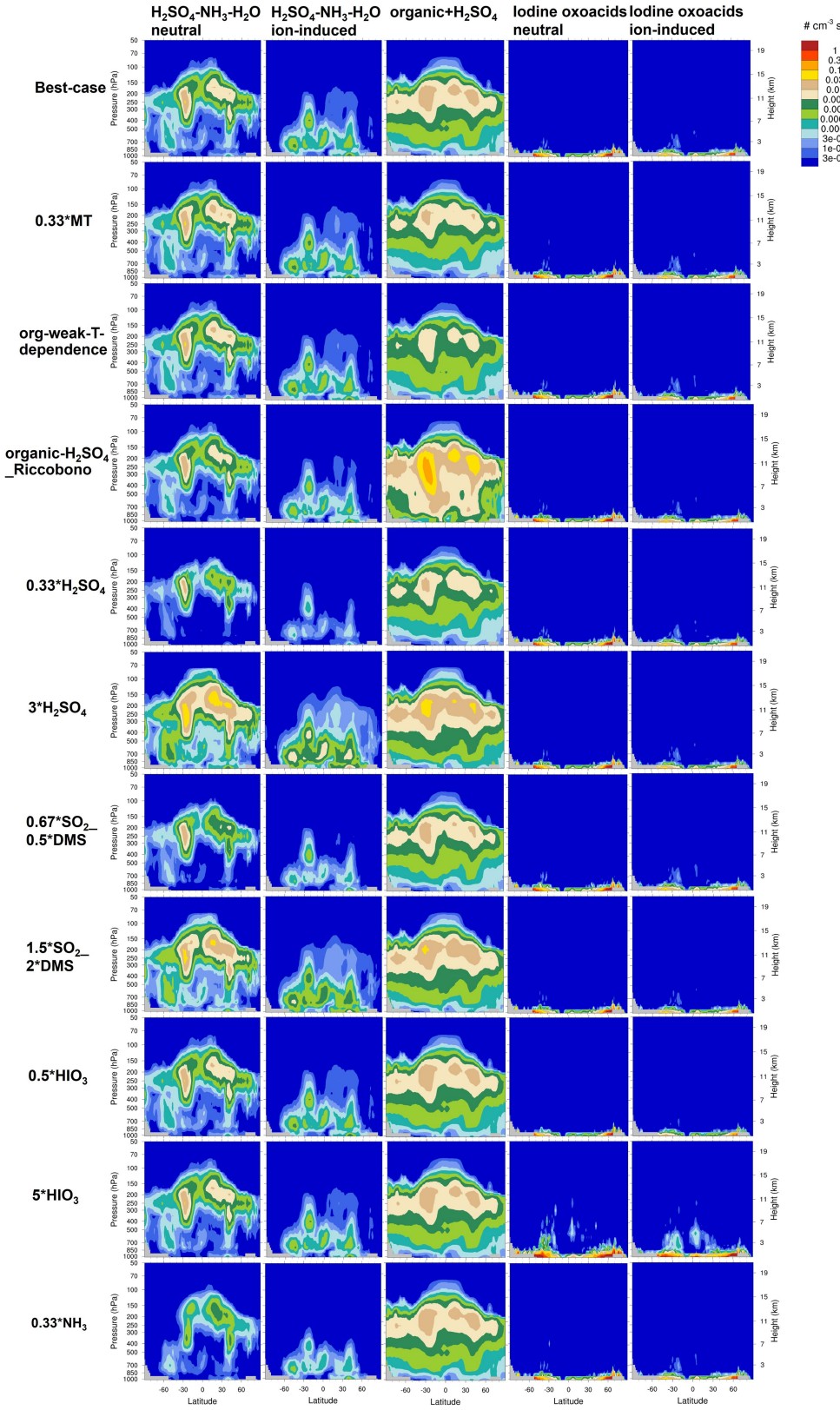

**Extended Data Fig. 8 | Same as Extended Data Fig. 7 but for the Atlantic Ocean (20° W–40° W) .**

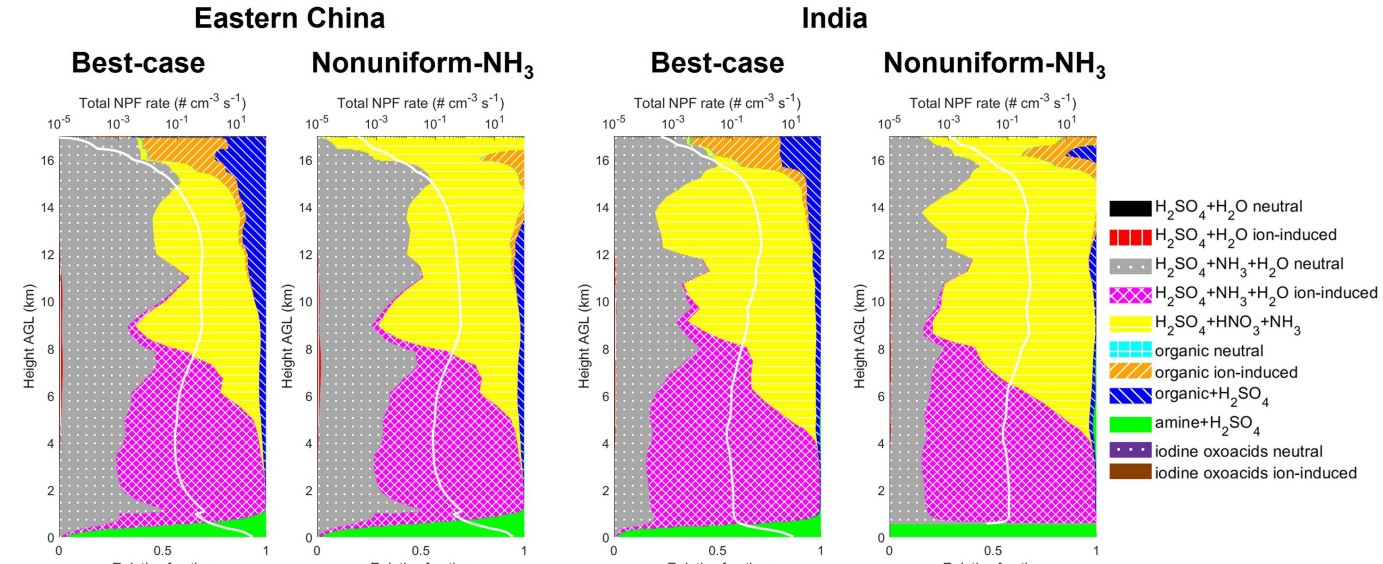

**Extended Data Fig. 9 | NPF rates as a function of height AGL over Eastern China and India, which are parts of the Asian monsoon region, under the best-case scenario and a sensitivity scenario that assumes that NH₃ concentration accumulates in a small fraction of a model grid.** White lines represent the total NPF rates of all mechanisms at a diameter of 1.7 nm ($J_{1.7}$, on a log scale) and the coloured areas represent the relative contributions of different mechanisms, both averaged in 2016 over the regions specified in Extended Data Fig. 1b. For a scenario intermediate between the best-case simulation and the above sensitivity simulation, the contribution of $H_2SO_4$–$HNO_3$–$NH_3$ nucleation could be even higher, because the limitation of $H_2SO_4$ availability may not be as strong as in the above sensitivity simulation.

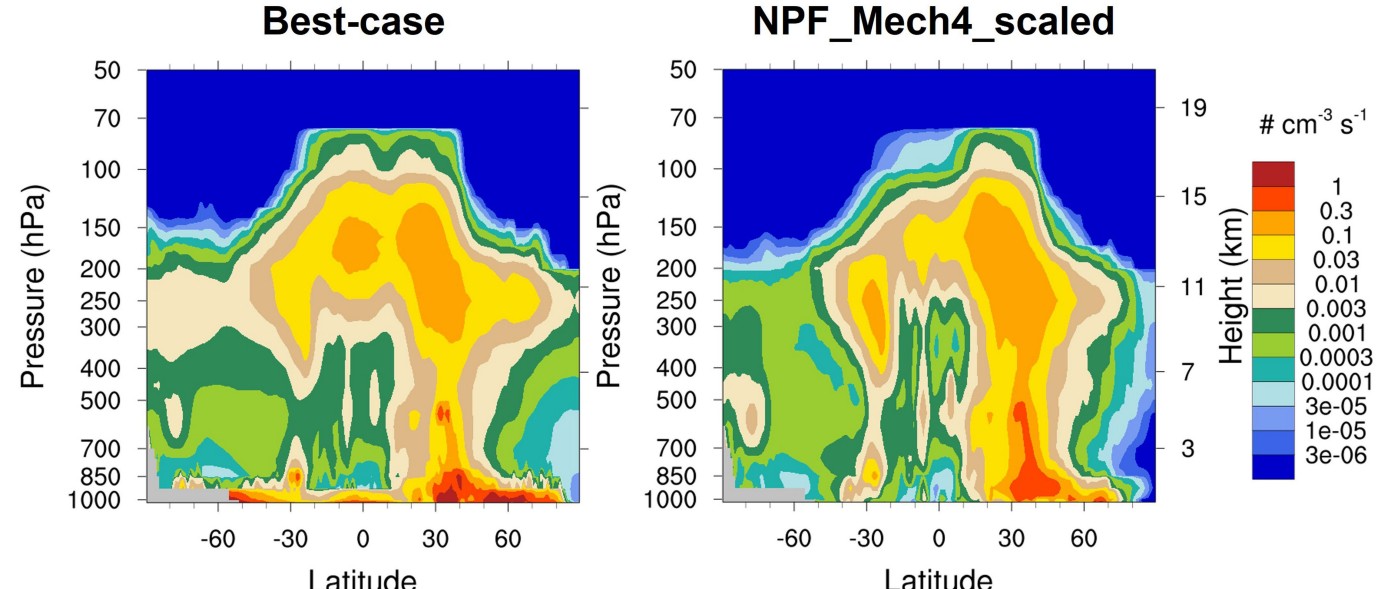

**Extended Data Fig. 10 | Comparison between simulated zonal mean NPF rates in two scenarios. a**, Best-case simulation including 11 nucleation mechanisms. **b**, A sensitivity simulation that includes only four traditional nucleation mechanisms (neutral and ion-induced $H_2SO_4$–$H_2O$ nucleation and $H_2SO_4$–$NH_3$–$H_2O$ nucleation) but scales the NPF rates of these mechanisms to match the globally averaged NPF rate of the best-case simulation.