## [Peer Review File · Nature]

Manuscript Title: Global variability in atmospheric new particle formation mechanisms

Reviewer Comments & Author Rebuttals

Reviewer Reports on the Initial Version:

Referees' comments:

Referee #1 (Remarks to the Author):

In this work, the authors implement into a global earth-system model eleven new-particle formation (NPF) mechanisms, some of which have been included in very few previous studies, and no global modelling studies to my knowledge. They examine the competition between these different NPF mechanisms in several regions, with a focus on determining which NPF mechanism dominates in each region at different altitudes. They include several sensitivity studies to examine the robustness of their conclusions to simulated concentrations of aerosol precursor gases and uncertain NPF mechanisms. Notably, they find that some of the novel NPF mechanisms dominate in certain regions, including iodine oxoacids nucleation at the ocean surface, leading to greatly increased particle number concentrations and cloud condensation nuclei concentrations compared to a case not including these novel mechanisms. This work has implications for the concentration of cloud condensation nuclei in the pre-industrial period, and thereby aerosol-cloud interactions, a large uncertainty in the anthropogenic effect on climate.

I recommend the manuscript for publication pending the resolution of my following comments:

The authors currently alternate between referring to their base case scenario as “best-case” and “NPF_Mech11”. Please choose one of the names and use it consistently throughout the document, including the Extended Data. (I prefer “best-case”, but “NPF_Mech11” would be fine as well.)

The authors describe their NPF_Mech4 experiment as resembling the NPF treatment commonly used in climate model. However, it is not uncommon for global models to scale process rates to better match observations, implicitly accounting for missing mechanisms. Would the authors like to comment on the likely results of applying a fixed scaling factor to the NPF rates in NPF_Mech4 to attempt to better match the results of NPF_Mech11?

p9, line 1: Please insert “the” between “to” and “abundance”

p26, lines 9, 16-18: “with O:C > 0.4 as inputs” If I understood correctly, the R2D-VBS was simplified to a 1D VBS with no prognostic tracking of the O:C ratio. What proportion of ULVOCS and ELVOCS was assumed to have O:C > 4? On what basis was this assumed value chosen?

p27 lines 19-20: If an equation is long enough that it needs to be split across two lines in the text, it deserves to be printed on a separate line and be formatted with an equation editor.

p30 line 16: Have the authors made an error in citing Wiedinmyer et al (2011) for this statement? I haven’t read the full reference, but it’s not obvious that the description of the FINN inventory would support this statement.

p30 lines 21-24: Zhao et al. (2020) (the authors’ reference 14) states that the SOA concentrations are sensitive to the branching ratio, which in turn depended on the O:C ratio in that study. Since the O:C was not tracked in this study, what was assumed in this study?

p34, lines 2 and 6: The modal widths for the best-case scenario and the Large_Mode_Width are both stated as being 1.6, 1.6, and 1.8. Which of these is incorrect?

Fig. 1: Is it feasible to move the colour legend so that it does not cover Fig. 1B, or to give it some transparency? Currently, it’s not clear how high in altitude the amine+H₂SO₄ proportion reaches. I assume that the region the legend covers is almost entirely organics+H₂SO₄. This doesn’t affect the conclusions of this work, but I can see it being of interest to other researchers.

Extended Data Table 2: What was the fixed fraction or constant yield used in the NPF_Mech11_constYield experiment?

Referee #2 (Remarks to the Author):

Quantification of the contribution of atmospheric new particle formation (NPF) to particle number and climatically more relevant cloud condensation nuclei (CCN) budgets in the global atmosphere is not possible without global model simulations. Such simulations have been performed a number of times during the last couple of decades, with a variable success and relatively large remaining uncertainties. A major, is not dominant, reason for these

uncertainties has been the lack of our understanding on atmospheric NPF pathways, both with regard of which pathways are important in different atmospheric environments and how individual pathways depend on precursor vapor concentrations and concentrations of ions. The work by Zhao et al. addresses these challenges by incorporating the existing laboratory-based knowledge on atmospheric NPF into a global modeling framework. The paper provides a very versatile view on atmospheric NPF: there appears to be no single dominant NPF mechanism in the global atmosphere, instead the relative roles of different mechanisms show a large geographical and altitude-dependent variability that can be related to the sources of precursor compounds, their atmospheric transport and prevailing meteorological conditions. While some indications of such versatility have been obtained from field measurements and earlier model investigations, the paper by Zhao et al. brings up this view more concretely and quantitatively than earlier studies on this subject. The results of this paper have important implications for not only our current understanding on NPF in the present-day atmosphere, but also how the role of NPF as a source of atmospheric aerosols and CCN has changed in the past and how it might change in the future as a result of anthropogenic and natural emission ranges.

The approach applied by Zhao et al. is not novel by itself, as a very similar approach was already adopted by Dunne et al. (2006). However, the current work is not just an update of earlier global simulations but a major leap forward for several reasons: 1) altogether 11 NPF mechanisms have been included, covering practically all the mechanisms thought to be atmospherically relevant at the moment, 2) the parameterized descriptions of the incorporated mechanisms are based on the latest information that is available from laboratory experiments, 3) the role of organic compounds in NPF has, for the first time, incorporated in a way that takes into account the chemistry and temperature-dependent thermodynamics of these compounds, and 4) the sensitivity of the results to the most important known uncertainties has been performed in a relatively comprehensive manner.

The paper is scientifically sound, both in terms of the approach and analysis of the results. The used model and its potential weaknesses have been adequately described. The data used for parameterized NPF mechanisms is based on accurate and well-documented laboratory experiments. The atmospheric measurement data, against which the conducted model simulations have been compared, can be considered carefully quality controlled. Besides the results from model simulations, the current work does not produce other kind of data that would require quality checking or control.

The paper does not involve statistical analyses, which is understandable for the chosen approach. Instead, uncertainties in modeling results have been investigated via sensitivity analyses. While the number of performed sensitivity simulations is by no means very large,

the selected simulations are carefully selected, reflecting existing uncertainties in model parameterizations and atmospheric chemistry in a balanced and well-designed manner.

The conclusions made from modeling results appear robust based on the available information.

While there is always room for using more data, improving the model design and performing additional simulations, I do foresee that major improvements in the results could be achieved by additional work at the moment. I do, however, think that the paper would benefit from some further discussion. First, while the authors acknowledge the possibility that current laboratory experiments may not capture all the details of NPF taking place in the atmosphere, they seem overly optimistic. The last few years have identified a number of new atmospherically relevant NPF mechanisms, and the same is certainly possible in the future as well. For example, the synergic effects of more than two vapors participating in NPF may change the picture (as shown also in the paper for a couple of mechanisms not included in earlier studies), and it is more than likely that new synergies will be identified in future experiments more accurately mimicking the atmospheric composition. Second, since our view on NPF in the atmosphere has changed quite dramatically since the first global model simulations, it would be worth having a short discussion on how our understanding on different NPF pathways and their relative roles in the atmosphere has evolved over time. I understand that there is little room for such discussion in the main text, but it could be put into the supplementary material.

In general, the paper gives credit to the earlier work on this topic to the extent possible with the limited number of references possible to be used in the main text. The results of this work could, however, be put to a broader context of earlier global model simulations on NPF (see my previous comment), especially since the number of such simulations is not too exhaustive to be discussed separately.

The main text of the paper is very well written and structured, including the introduction to the topic, main conclusions and associated implications. The abstract adequately describes the paper. Overall, the paper is of high scientific and technical quality, and it definitely provides fresh insight into atmospheric NPF that is worthy of publication.

Referee #3 (Remarks to the Author):

This manuscript examines the relative importance of different nucleation processes over

different global regions. This is an interesting work. However, the sensitivity analysis seems to be coming from a black box, without knowing/showing the precursor concentrations. Currently, there is a coherent problem with global models due to highly uncertain emissions of chemical precursors. Thus, it is difficult to conclude which mechanisms dominate in a specific region.

Author Rebuttals to Initial Comments:

Referees' comments:

Referee #1 (Remarks to the Author):

In this work, the authors implement into a global earth-system model eleven new-particle formation (NPF) mechanisms, some of which have been included in very few previous studies, and no global modelling studies to my knowledge. They examine the competition between these different NPF mechanisms in several regions, with a focus on determining which NPF mechanism dominates in each region at different altitudes. They include several sensitivity studies to examine the robustness of their conclusions to simulated concentrations of aerosol precursor gases and uncertain NPF mechanisms. Notably, they find that some of the novel NPF mechanisms dominate in certain regions, including iodine oxoacids nucleation at the ocean surface, leading to greatly increased particle number concentrations and cloud condensation nuclei concentrations compared to a case not including these novel mechanisms. This work has implications for the concentration of cloud condensation nuclei in the pre-industrial period, and thereby aerosol-cloud interactions, a large uncertainty in the anthropogenic effect on climate.

I recommend the manuscript for publication pending the resolution of my following comments:

We greatly appreciate the reviewer's valuable comments and constructive suggestions, which have helped us improve our manuscript. We have carefully revised the manuscript in accordance with those comments, and our point-by-point responses are provided below. The reviewer's comments are presented in blue text, our responses are written in black text, and quotations from our manuscript are presented in italic type.

The authors currently alternate between referring to their base case scenario as "best-case" and "NPF_Mech11". Please choose one of the names and use it consistently throughout the document, including the Extended Data. (I prefer "best-case", but "NPF_Mech11" would be fine as well.)

Following the reviewer's suggestion, we have used "best-case" throughout the manuscript as appropriate.

The authors describe their NPF_Mech4 experiment as resembling the NPF treatment commonly used in climate model. However, it is not uncommon for global models to scale process rates to better match observations, implicitly accounting for missing mechanisms. Would the authors like to comment on the likely results of applying a fixed scaling factor to the NPF rates in NPF_Mech4 to attempt to better match the results of NPF_Mech11?

We conducted a sensitivity simulation (“NPF_Mech4_scaled”) in which we applied a fixed scaling factor to the NPF rates in NPF_Mech4 such that its globally averaged NPF rate matched that of the best-case simulation (i.e., NPF_Mech11). Here, the globally averaged NPF rate is defined as the average of the NPF rates across all model grid boxes (both horizontally and vertically), weighted by the volumes of those grid boxes. Extended Data Fig. 10 (shown below) illustrates the zonal mean NPF rates in the best-case and “NPF_Mech4_scaled” scenarios. Despite using the same globally averaged NPF rates in both scenarios, the NPF rates for specific regions and altitudes differ greatly. Moreover, because the dominant NPF mechanism differs between the two scenarios, the NPF rate would respond differently to the perturbation of precursor concentrations in the future. Therefore, whilst the scaling approach might work well if we focused only on the total NPF rate at a specific location, it was critically important to develop a comprehensive model representation of NPF mechanisms, as done in this study, for the following reasons: 1) to simulate the NPF over broad regions rather than just at a specific location, 2) to elucidate the dominant NPF mechanism, and 3) to evaluate how NPF changes in response to variation in precursor emissions/concentrations. We have added the above discussion in the revised manuscript (Page 47 Lines 11–22).

Extended Data Fig. 10 Comparison between simulated zonal mean NPF rates in two scenarios. (A) best-case simulation including 11 nucleation mechanisms, and (B) a sensitivity simulation that includes only 4 traditional nucleation mechanisms (neutral and ion-induced $H_2SO_4-H_2O$ nucleation and $H_2SO_4-NH_3-H_2O$ nucleation) but scales the NPF rates of these mechanisms to match the globally averaged NPF rate of the best-case simulation.

p9, line 1: Please insert “the” between “to” and “abundance”

Revision has been made (Page 9 Line 1).

p26, lines 9, 16-18: “with O:C > 0.4 as inputs” If I understood correctly, the R2D-VBS was simplified to a 1D VBS with no prognostic tracking of the O:C ratio. What proportion of ULVOCS and ELVOCS was assumed to have O:C > 4? On what basis was this assumed value chosen?

We apologize for the confusion. The R2D-VBS was indeed condensed to an equivalent 1D-VBS when being incorporated in E3SM, but only species with an O:C ratio of >0.4 were included in this 1D-VBS. Thus, the total ULVOC and ELVOC concentrations within the condensed 1D-VBS were used to drive organic-mediated nucleation. The remaining less-oxygenated compounds (i.e., with $O:C \leq 0.4$) do not contribute to nucleation in our model, but they might still contribute to the formation of secondary organic aerosol (SOA). We simulated the SOA formation associated with those less-oxygenated compounds using the original SOA parameterization in E3SM¹, which was based on a simple 1D-VBS involving five surrogate species with saturation vapor concentrations ranging from 10^{-1} to $10^3 \mu\text{g m}^{-3}$. Then, we parameterized the SOA fraction formed from the less-oxygenated compounds by fitting a series of box-model simulations under various temperatures following Zhao et al.², and we applied the parameterized fraction below to the original SOA formation parameterization in E3SM:

$$f = 1.0 - 0.37 / (\exp(-0.0597 * T + 14.02) + 1.0)$$

where T is temperature (unit: K).

We have included the above description in the revised manuscript (Page 31 Lines 12–23).

p27 lines 19-20: If an equation is long enough that it needs to be split across two lines in the text, it deserves to be printed on a separate line and be formatted with an equation editor.

Revision has been made in accordance with the reviewer’s suggestion (Page 27 Lines 17–21). The revised text is as shown below:

“In the current study, we derived the following temperature-dependence function using the cluster kinetic model developed by Cai et al.³ and applied it to the above parameterization:

$$J_T = J_{278K} \left(1.576 \exp \left(-\left((T - 250.6) / 23.18 \right)^2 \right) + 0.6956 \exp \left(-\left((T - 273.1) / 13.01 \right)^2 \right) \right)$$

where T is temperature (unit: K)."

p30 line 16: Have the authors made an error in citing Wiedinmyer et al (2011) for this statement? I haven't read the full reference, but it's not obvious that the description of the FINN inventory would support this statement.

We are sorry that the three references cited in this sentence were incorrect owing to a technical issue with the reference management software. We have corrected the references as shown below:

"Notably, isoprene might suppress the NPF triggered by monoterpene oxidation products at low altitudes⁴⁻⁶." (Page 30 Line 24 and Page 31 Line 1)

References:

4 Kiendler-Scharr, A. et al. New particle formation in forests inhibited by isoprene emissions. *Nature* 461, 381-384, doi:10.1038/nature08292 (2009).

5 Lee, S.-H. et al. Isoprene suppression of new particle formation: Potential mechanisms and implications. *J. Geophys. Res-Atmos.* 121, 14621-14635, doi:10.1002/2016jd024844 (2016).

6 Heinritzi, M. et al. Molecular understanding of the suppression of new-particle formation by isoprene. *Atmos. Chem. Phys.* 20, 11809-11821, doi:10.5194/acp-20-11809-2020 (2020).

p30 lines 21-24: Zhao et al. (2020) (the authors' reference 14) states that the SOA concentrations are sensitive to the branching ratio, which in turn depended on the O:C ratio in that study. Since the O:C was not tracked in this study, what was assumed in this study?

In answering the reviewer's question, we first elaborate on some relevant details of the Radical Two-Dimensional Volatility Basis Set (R2D-VBS). The R2D-VBS explicitly simulates the peroxy radical (RO₂) chemistry and tracks the distribution of radical termination products within the two-dimensional space defined by saturation vapor concentration (C*) and the O:C ratio. Specifically, the reactions begin with oxidation of monoterpenes with OH, O₃, and NO₃, producing RO₂. Then, RO₂ undergoes either autoxidation or termination. Autoxidation produces a more-oxygenated RO₂, which will further undergo autoxidation or termination.

Termination proceeds through unimolecular termination or reactions with NO, HO₂, or another RO₂. The cross reactions of RO₂ produce either dimer or non-dimer products. The non-dimer products of RO₂ cross reactions, as well as the products of unimolecular termination and reaction with NO, will undergo either functionalization or fragmentation, with a branching ratio (β) between the two that depends on the O:C ratio of the RO intermediates produced in the very first step of the termination processes. It should be noted that the O:C ratio of the RO intermediates is explicitly tracked in the model. Then, we distributed the stable molecules from each of the RO₂ termination pathways to a series of species in the C*–O:C space via kernels; kernels define the rule for mapping a reactant (RO₂ in this case) with a given C* and O:C ratio to a distribution of reaction products in the C*–O:C space via a specific termination pathway. Subsequently, we summed the species in the same C* bin and different O:C bins, which equivalently condensed the R2D-VBS to a 1D-VBS. As can be seen, for the R2D-VBS, the branching ratio β depends on the O:C ratio of the RO intermediates instead of on the O:C ratio of the surrogate species within the R2D-VBS. Thus, the simplification of R2D-VBS did not affect the simulation of functionalization, fragmentation, or the branching ratio between them.

We have included the above description in the revised manuscript (Page 30 Lines 2–17).

p34, lines 2 and 6: The modal widths for the best-case scenario and the Large_Mode_Width are both stated as being 1.6, 1.6, and 1.8. Which of these is incorrect?

We apologize for the typo. The modal widths for the best-case scenario are 1.6, 1.6, and 1.8, whereas those for the “Large_Mode_Width” scenario should have been 1.8, 1.8, and 2.0. We have corrected this error in the revised manuscript (Page 35 Lines 6–8).

Fig. 1: Is it feasible to move the colour legend so that it does not cover Fig. 1B, or to give it some transparency? Currently, it's not clear how high in altitude the amine+H₂SO₄ proportion reaches. I assume that the region the legend covers is almost entirely organics+H₂SO₄. This doesn't affect the conclusions of this work, but I can see it being of interest to other researchers.

We have revised the figure in accordance with the reviewer's suggestion. The revised figure is shown below.

Fig. 1. Mechanisms of NPF and constraints from observations over rainforests. (A)

Comparison of simulated particle number concentrations with aircraft measurements obtained over the Amazon during the ACRIDICON-CHUVA campaign in September 2014. Both simulations and observations are for particles >10 nm near the surface and 20 nm above the altitude of 13.8 km, with smooth transition in between. The lines represent mean concentrations within each vertical bin and the shaded areas represent the 25th to 75th percentiles of the observations. All particle number concentrations are normalized to standard temperature and pressure (STP; 273.15 K and 101.325 kPa). Definitions of the model scenarios are given in the main text and Supplementary Table 1. Note that, in addition to the common NPF rate maximum in the upper troposphere, there is another maximum near the surface in Southeastern Asia driven by iodine oxoacids nucleation and amine–H₂SO₄ nucleation. This is because Southeastern Asia is strongly affected by oceanic and anthropogenic emissions and thus possesses some NPF features typical of oceanic and polluted regions. (B) NPF rates as a function of height above ground level (AGL) over the Central Amazon, Central Africa, and Southeastern Asia. White lines represent the total NPF rates of all mechanisms at diameter of 1.7 nm ($J_{1.7}$, on a log scale), and the colored areas represent the relative contributions of different mechanisms, both averaged in 2016 over the regions specified in Extended Data Fig. 1B.

Extended Data Table 2: What was the fixed fraction or constant yield used in the NPF_Mech11_constYield experiment?

The “fixed fractions” used here followed Gordon et al.⁷. Specifically, organic–H₂SO₄ nucleation was linked to all oxidation products of monoterpenes; in other words, the “fixed fraction” of monoterpene oxidation products used to drive organic–H₂SO₄ nucleation was 1.0. Pure-organic nucleation was assumed driven by highly oxygenated organic molecules

(HOMs), the molar yields (fixed fraction) of which were assumed to be 1.4% for the reaction of monoterpenes with O₃ and 0.6% for the reaction of monoterpenes with OH. We have clarified this in Supplementary Table 1 (i.e., Extended Data Table 2 in the original manuscript).

Referee #2 (Remarks to the Author):

Quantification of the contribution of atmospheric new particle formation (NPF) to particle number and climatically more relevant cloud condensation nuclei (CCN) budgets in the global atmosphere is not possible without global model simulations. Such simulations have been performed a number of times during the last couple of decades, with a variable success and relatively large remaining uncertainties. A major, is not dominant, reason for these uncertainties has been the lack of our understanding on atmospheric NPF pathways, both with regard of which pathways are important in different atmospheric environments and how individual pathways depend on precursor vapor concentrations and concentrations of ions. The work by Zhao et al. addresses these challenges by incorporating the existing laboratory-based knowledge on atmospheric NPF into a global modeling framework. The paper provides a very versatile view on atmospheric NPF: there appears to be no single dominant NPF mechanism in the global atmosphere, instead the relative roles of different mechanisms show a large geographical and altitude-dependent variability that can be related to the sources of precursor compounds, their atmospheric transport and prevailing meteorological conditions. While some indications of such versatility have been obtained from field measurements and earlier model investigations, the paper by Zhao et al. brings up this view more concretely and quantitatively than earlier studies on this subject. The results of this paper have important implications for not only our current understanding on NPF in the present-day atmosphere, but also how the role of NPF as a source of atmospheric aerosols and CCN has changed in the past and how it might change in the future as a result of anthropogenic and natural emission ranges.

The approach applied by Zhao et al. is not novel by itself, as a very similar approach was already adopted by Dunne et al. (2006). However, the current work is not just an update of earlier global simulations but a major leap forward for several reasons: 1) altogether 11 NPF mechanisms have been included, covering practically all the mechanisms thought to be atmospherically relevant at the moment, 2) the parameterized descriptions of the incorporated mechanisms are based on the latest information that is available from laboratory experiments, 3) the role of organic compounds in NPF has, for the first time, incorporated in a way that takes into account the chemistry and temperature-dependent

thermodynamics of these compounds, and 4) the sensitivity of the results to the most important known uncertainties has been performed in a relatively comprehensive manner.

The paper is scientifically sound, both in terms of the approach and analysis of the results. The used model and its potential weaknesses have been adequately described. The data used for parameterized NPF mechanisms is based on accurate and well-documented laboratory experiments. The atmospheric measurement data, against which the conducted model simulations have been compared, can be considered carefully quality controlled. Besides the results from model simulations, the current work does not produce other kind of data that would require quality checking or control.

The paper does not involve statistical analyses, which is understandable for the chosen approach. Instead, uncertainties in modeling results have been investigated via sensitivity analyses. While the number of performed sensitivity simulations is by no means very large, the selected simulations are carefully selected, reflecting existing uncertainties in model parameterizations and atmospheric chemistry in a balanced and well-designed manner.

The conclusions made from modeling results appear robust based on the available information.

While there is always room for using more data, improving the model design and performing additional simulations, I do foresee that major improvements in the results could be achieved by additional work at the moment. I do, however, think that the paper would benefit from some further discussion. First, while the authors acknowledge the possibility that current laboratory experiments may not capture all the details of NPF taking place in the atmosphere, they seem overly optimistic. The last few years have identified a number of new atmospherically relevant NPF mechanisms, and the same is certainly possible in the future as well. For example, the synergic effects of more than two vapors participating in NPF may change the picture (as shown also in the paper for a couple of mechanisms not included in earlier studies), and it is more than likely that new synergies will be identified in future experiments more accurately mimicking the atmospheric composition. Second, since our view on NPF in the atmospheric has changed quite dramatically since the first global model simulations, it would be worth having a short discussion on how our understanding on different NPF pathways and their relative roles in the atmosphere has evolved over time. I understand that there is little room for such discussion in the main text, but it could be put into the supplementary material.

In general, the paper gives credit to the earlier work on this topic to the extent possible with the limited number of references possible to be used in the main text. The results of this work could, however, be put to a broader context of earlier global model simulations on NPF

(see my previous comment), especially since the number of such simulations is not too exhaustive to be discussed separately.

The main text of the paper is very well written and structured, including the introduction to the topic, main conclusions and associated implications. The abstract adequately describes the paper. Overall, the paper is of high scientific and technical quality, and it definitely provides fresh insight into atmospheric NPF that is worthy of publication.

We thank the reviewer for recognizing the scientific merit of our paper. We also appreciate the reviewer's valuable suggestions, which have helped us improve the manuscript. We have carefully revised the manuscript in accordance with those comments. The reviewer's comments are presented in blue text, our responses are written in black text, and quotations from our manuscript are presented in italic type.

(1) Following the reviewer's suggestion, we have added further discussion on the continuously emerging new NPF mechanisms, especially those involving synergistic multicomponent NPF (Page 13 Line 17 to Page 14 Line 3).

"In particular, our model has not included all NPF mechanisms exhaustively. New atmospherically relevant NPF mechanisms have been identified recently, especially those involving the synergistic effects of multiple compounds⁸⁻¹¹. For example, amines and NH₃ have been found to nucleate synergistically with H₂SO₄, especially in environments with insufficient amines to fully stabilize H₂SO₄ clusters^{9,10}. HNO₃ has been found to enhance DMA-H₂SO₄ nucleation under favorable conditions with relatively high HNO₃ and DMA concentrations¹¹. It is more than likely that new synergistic effects or other new NPF mechanisms will be identified in future experiments that better mimic the atmospheric composition. Parameterizing the emerging new NPF mechanisms and incorporating them into the model can still refine and possibly modify the picture we present, and thus such work is needed in the future."

Considering the potential existence of synergistic multicomponent NPF mechanisms and other potential uncertainties, we have toned down some strong statements regarding dominant NPF mechanisms and better qualified some other statements to clarify that they were derived from our model:

*"Organic-mediated nucleation (pure-organic and organic-H₂SO₄ nucleation) consistently dominates in the upper troposphere of the three regions **according to our model.**"* (Page 6 Lines 22–23)

*"The NPF rates in these regions are highest near the surface and mainly driven by amine-H₂SO₄ nucleation **in our model.**"* (Page 8 Lines 9–10)

*“Hence, H_2SO_4 – HNO_3 – NH_3 nucleation is **probably** the leading mechanism in the upper troposphere above Eastern China and India.” (Page 9 Lines 8–9)*

*“In the upper troposphere above the Pacific and Atlantic oceans, organic– H_2SO_4 nucleation and H_2SO_4 – NH_3 – H_2O neutral nucleation are **most likely** the two dominant mechanisms of nucleation according to our model.” (Page 10 Lines 18–20)*

(2) We have also added discussion on how global model representations of NPF as well as our understanding of NPF mechanisms in the atmosphere have evolved over time. (Page 46 Line 6 to Page 50 Line 14)

[revised manuscript text omitted]

Referee #3 (Remarks to the Author):

This manuscript examines the relative importance of different nucleation processes over different global regions. This is an interesting work. However, the sensitivity analysis seems to be coming from a black box, without knowing/showing the precursor concentrations. Currently, there is a coherent problem with global models due to highly uncertain emissions of chemical precursors. Thus, it is difficult to conclude which mechanisms dominate in a specific region.

We are pleased that the reviewer is interested in our work. We also appreciate the reviewer's critical comments, which have helped us improve our manuscript. Our detailed responses are provided below. In brief, we have presented the concentrations of nucleation precursors in the best-case and sensitivity simulations. We have also conducted additional sensitivity simulations to more adequately examine the impact of uncertainties in precursor emissions on our results and conclusions; these new simulations, together with the sensitivity simulations already present in our original manuscript, suggest that our main findings regarding leading mechanisms in key regions are likely unaffected by the uncertainty of emissions. Nevertheless, we acknowledge that no model is perfect and that there might always be uncertainties beyond our current level of knowledge; therefore, we have toned down some of our statements regarding dominant nucleation mechanisms.

(1) Following the reviewer's suggestion to show precursor concentrations, we have added Supplementary Figs. 2–5 (shown below) to illustrate the concentrations of precursors directly involved in nucleation, including H_2SO_4 , NH_3 , dimethylamine (DMA), HIO_3 , and ultralow and extremely low volatility organic compounds (ULVOCs and ELVOCs, respectively) in our best-case and sensitivity simulations. These figures show precursor concentrations over the main regions of interest in this study, including rainforests, anthropogenically polluted regions, and the Pacific and Atlantic oceans, following the same format in which the NPF rates were presented in the manuscript. Note that each sensitivity experiment perturbed only one precursor or parameter; therefore, the concentrations of a precursor in most sensitivity simulations are very close to those in the best-case simulation unless its emissions/concentrations were perturbed in that simulation. To make the figures concise, we show only the concentrations of a precursor in the best-case simulation and in the sensitivity simulations where its concentrations show significant differences from the best-case scenario. Definitions of all sensitivity experiments are summarized in Supplementary Table 1 (shown below), and some of the experiments are described below.

(2) In response to the reviewer's comment on emission uncertainty, we have systematically investigated the impact of the uncertainty of precursor emissions on our findings regarding the dominant NPF mechanisms over the main regions of interest. As part of this effort, we conducted additional sensitivity experiments to quantify the impact of uncertainties in precursor emissions. Extended Data Figs. 5–8 (shown below) summarize the results of the

sensitivity simulations over rainforests, anthropogenically polluted regions, and the Pacific and Atlantic oceans. Below, we discuss separately the emission uncertainties associated with each precursor involved in nucleation, i.e., H₂SO₄, ULVOCs and ELVOCs, DMA, NH₃, and HIO₃. The descriptions and discussions have also been included in the revised manuscript (Page 39 Line 11 to Page 46 Line 5).

H₂SO₄ is a key species involved in 7 of the 11 NPF mechanisms considered in our model. H₂SO₄ is mainly formed from the oxidation of SO₂ emissions in terrestrial regions and from the oxidation of both SO₂ and dimethylsulfide (DMS) emissions in oceanic regions. The SO₂ emissions used in this study were obtained from the Coupled Model Intercomparison Project phase 6 emission dataset⁴⁶, based on the Community Emissions Data System (CEDS) emission inventory⁴⁷. Among all air pollutants, the emissions of SO₂ can be most accurately estimated, based on the mass balance of sulfur. Smith et al.⁴⁸ estimated the overall global uncertainty in SO₂ emissions of the CEDS inventory to be 8%–14% and regional uncertainties to be within 30%. Another estimate⁴⁹ based on a similar EDGARv4.3.2 emission inventory reported that SO₂ emissions have uncertainty of 14.4%–47.6% at the regional level. Overall, we conclude, with confidence, that the uncertainty of SO₂ emissions at the regional level should be well within 50%. The marine DMS emissions used in this study were obtained from Wang et al.^{50,51}, who reported an annual marine DMS emission of 20.4 TgS yr⁻¹ based on surface ocean DMS concentrations and a parameterization of sea-to-air gas transfer velocity. Using generally the same method, Lana et al.⁵² derived an annual marine DMS emission of 28.1 TgS yr⁻¹. They further showed that the uncertainty in the parameterization of sea-to-air gas transfer velocity resulted in a DMS emission uncertainty range of 17.6–34.4 TgS yr⁻¹, while the uncertainty in the underlying data used to derive surface ocean DMS concentrations resulted in a DMS emission uncertainty range of 24.1–40.4 TgS yr⁻¹. Taken together, the uncertainty of the marine DMS emission estimate should be within 100% (i.e., a factor of 2). To evaluate the impact of such uncertainty, we conducted two sensitivity experiments: the first one (“1.5*SO₂_2*DMS”), compared with the best-case simulation, increased the SO₂ and DMS emissions by a factor of 1.5 and 2, respectively, and the second one (“0.67*SO₂_0.5*DMS”) reduced the SO₂ and DMS emissions by a factor of 1.5 and 2, respectively. We also looked at the uncertainty from a different perspective. Our model evaluation showed that the simulated H₂SO₄ concentration was generally within a factor of 3 of that of the observations across various polluted regions (Extended Data Fig. 3A), and that this uncertainty range encompassed not only the uncertainty in the precursor emissions but also the uncertainty in other chemical and physical processes. Accordingly, we conducted two additional sensitivity simulations (“0.33*H₂SO₄” and “3*H₂SO₄”) by reducing and increasing the H₂SO₄ concentrations simulated in the best-case scenario by a factor of 3 to cover the uncertainty in H₂SO₄ concentration. Extended Data Figs. 5–8 show that the four sensitivity scenarios have limited impact on the relative contributions of individual NPF

mechanisms over our main regions of interest, largely because the increase or decrease in H_2SO_4 concentration causes simultaneous changes in the rates of most NPF mechanisms.

ULVOCs and ELVOCs drive organic-mediated nucleation (pure-organic and organic- H_2SO_4 nucleation), which according to our model is the dominant mechanism in the upper troposphere above three rainforest regions (i.e., the Amazon, Southeastern Asia, and Central Africa) and probably one of the two primary mechanisms in the upper troposphere above the Pacific and Atlantic oceans. ULVOCs and ELVOCs in our model are mainly formed through the oxidation of monoterpene emissions that are lifted to the upper troposphere by convection. Monoterpene emissions were calculated using the Model of Emissions of Gases and Aerosols from Nature developed by Guenther et al.⁵³, who estimated that uncertainty with a factor of 3 be associated with monoterpene emissions. To better evaluate the uncertainty in monoterpene emissions, we compared simulated monoterpene concentrations with field observations^{54,55} in two of the three main rainforest regions, i.e., the Amazon and Southeastern Asia, which are the largest sources of monoterpene globally⁵³ and key regions of interest in this work. Average simulated concentrations during the periods of the field campaigns (February–March and September–October 2014 in the Amazon, and April–July 2008 in Southeastern Asia) are 0.19 and 0.13 ppb, respectively, which are comparable to the observed values of 0.13 and 0.17 ppb; the model–observation differences are well within the uncertainty of a factor 3 estimated by Guenther et al.⁵³. To test the potential impact of emission uncertainty, we designed a sensitivity experiment (“0.33*MT”) in which the monoterpene emissions are reduced by a factor of 3; this likely represents an extreme case given the abovementioned good model–observation agreement of monoterpene concentrations. We did not test higher monoterpene concentrations because they would not challenge the leading roles of organic-mediated nucleation over the regions of interest. Extended Data Figs. 5, 7, and 8 reveal that even with reduced monoterpene emissions, organic-mediated nucleation remains the largest NPF mechanism in the upper troposphere above the three rainforest regions and one of the two primary mechanisms in the upper troposphere above the Pacific and Atlantic oceans.

DMA is a key precursor involved in amine- H_2SO_4 nucleation, which our study suggests is the leading NPF mechanism near the surface in anthropogenically polluted regions (Eastern China, India, Europe, and Eastern U.S.). Because it is difficult to directly quantify the uncertainty of DMA emissions using a bottom-up method, we instead examined the uncertainty of simulated DMA concentrations, which are more closely related to the amine- H_2SO_4 nucleation rate. Note that the concentrations of DMA near the surface in polluted regions are directly tied to local emissions because of their short lifetime. Comparison of simulated concentrations of DMA with observations in different anthropogenically polluted regions (Extended Data Fig. 3B) indicates that the simulated concentrations are within a

factor of 2.5 of the observed values. Thus, we designed a sensitivity experiment (“0.4*DMA”) that reduces the DMA concentrations by a factor of 2.5 relative to the best-case simulation. Note that this across-the-board reduction likely represents an extreme case according to the above evaluation results, because the model-to-observation ratios at different sites vary between 2.5 and 1/2.5 with less than half of the sites exhibiting an overestimation. Similar to the case of ULVOCs/ELVOCs, we did not test higher DMA concentrations because they would not alter the leading roles of amine–H₂SO₄ nucleation. Extended Data Fig. 6 illustrates that, under this sensitivity scenario, amine–H₂SO₄ nucleation remains the dominant nucleation mechanism near the surface of the four anthropogenically polluted regions. Furthermore, the results regarding the dominant role of amine–H₂SO₄ nucleation are consistent with recent observational studies that directly measured molecular clusters at a few polluted sites^{3,19,56}, which at least lends some support to our assessment results.

NPF mechanisms involving **NH₃** have been shown by our model to play key roles in the upper troposphere above the Asian monsoon regions (including Eastern China and India) and the Pacific and Atlantic oceans. Specifically, our model shows that synergistic H₂SO₄–HNO₃–NH₃ nucleation is probably a leading mechanism in the upper troposphere above Eastern China and India, whereas organic–H₂SO₄ nucleation and H₂SO₄–NH₃–H₂O neutral nucleation are the two primary NPF mechanisms in the upper troposphere above the Pacific and Atlantic oceans. In the upper troposphere of the Asian monsoon region, the simulated NH₃ concentration during summer mostly varies in the range of 10–40 ppt and it occasionally reaches 60 ppt (Extended Data Fig. 4), consistent with large-scale satellite observations (i.e., mostly 10–35 ppt, occasionally 150 ppt)^{57,58}. It should be noted that even with average NH₃ concentrations similar to those observed, our results represent a lower-limit estimate of the H₂SO₄–HNO₃–NH₃ nucleation rate, because real-world NH₃ concentration is nonuniform within a 1° × 1° model grid and because the H₂SO₄–HNO₃–NH₃ nucleation rate is much more strongly dependent on the NH₃ concentration than linear (see related sensitivity simulations in Extended Data Fig. 9). Moreover, even at grid level, the peak NH₃ concentration in the observations (~150 ppt) is somewhat higher than in our simulation (~60 ppt), potentially leading to a higher NPF rate. For these reasons, H₂SO₄–HNO₃–NH₃ nucleation is probably the leading mechanism in the upper troposphere above Eastern China and India. In the upper troposphere above the Pacific and Atlantic oceans, accurate large-scale NH₃ observations are unavailable; therefore, we evaluated the uncertainty associated with NH₃ from the perspective of emission uncertainty. In our model, the NH₃ concentrations were prescribed using the outputs of CAM-Chem⁵⁹, which were further driven by anthropogenic and marine NH₃ emission inputs. The uncertainty in anthropogenic NH₃ emissions has been estimated to be 125% at the global level and 186%–294% at the regional level^{49,60}. The marine NH₃ emissions used in the CAM-Chem simulation were obtained from Bouwman et al.⁶¹, who

reported a marine emission estimate of 8 TgN yr⁻¹ associated with an uncertainty range of a factor 2–3, although the relative source distribution might be more reliable than the absolute emissions⁶¹. Most other studies (e.g., Fowler et al.⁶², Paulot et al.⁶³, and references therein) estimated marine NH₃ emissions at 5.6–13 TgN yr⁻¹, with the most extreme estimates of 2.5 and 23 TgN yr⁻¹. Therefore, the emission estimate of Bouwman et al.⁶¹ with uncertainty of a factor 3 broadly encompasses the range of values reported in related studies. Considering the abovementioned uncertainties in anthropogenic and marine NH₃ emissions, as well as the linear relationship between NH₃ emissions and concentrations, we designed a sensitivity experiment (“0.33*NH₃”) by reducing NH₃ concentrations in the best-case simulation by a factor of 3. Similar to the discussion on organic-mediated nucleation, we did not test higher NH₃ concentrations because they would not challenge the important roles of H₂SO₄–NH₃–H₂O neutral nucleation over the regions of interest. Extended Data Figs. 7 and 8 indicate that even with low NH₃ concentration in the sensitivity experiment, H₂SO₄–NH₃–H₂O neutral nucleation remains one of the two primary NPF mechanisms in the troposphere over the Pacific and Atlantic oceans, although its contribution would be smaller than that of organic–H₂SO₄ nucleation.

Finally, **HIO₃** is the key driver of iodine oxoacids nucleation, which has been shown by our study to be the dominant NPF mechanism near the surface over oceans. HIO₃ is mainly formed from the oxidation of HOI and I₂ emissions over the oceans. Rather than evaluating the uncertainty of emissions, we directly examined the uncertainty of HIO₃ concentrations that are more closely tied to nucleation. Specifically, we compared simulated HIO₃ concentrations with observations at 10 oceanic or coastal sites distributed worldwide. The results showed that simulated HIO₃ concentrations vary between 80% below and 100% above the observed values (Extended Data Fig. 3C); the bias is slightly larger than that of H₂SO₄ and it is deemed reasonable considering that our study, to the best of our knowledge, represents the first time that HIO₃ formation chemistry has been simulated in three-dimensional models. To test the uncertainty in HIO₃ concentration, we designed two sensitivity experiments (“0.5*HIO₃” and “5*HIO₃”) that set HIO₃ concentrations to 0.5 and 5 times those of the best-case simulation results. Again, this across-the-board reduction or increase likely represents an extreme case because the biases at different locations vary between the two bounds. Extended Data Figs. 7 and 8 show that, in both sensitivity simulations, iodine oxoacids nucleation remains the dominant nucleation mechanism near the surface of the Pacific and Atlantic oceans.

In summary, our model evaluations and sensitivity simulations indicate that the uncertainties in precursor emissions and concentrations, at least those that to the best of our knowledge have been recognized, are unlikely to change our main findings regarding the leading NPF mechanisms in the main regions of interest, although they might affect the

exact magnitude of the contributions of individual mechanisms. Nevertheless, we acknowledge that there will always be uncertainties beyond our current level of knowledge; for example, there are limitations regarding observational data availability in terms of spatial and temporal coverage. Therefore, we have toned down some strong statements regarding the dominant NPF mechanisms and better qualified some other statements to clarify that they were derived from our model:

*“Organic-mediated nucleation (pure-organic and organic–H₂SO₄ nucleation) consistently dominates in the upper troposphere of the three regions **according to our model.**”* (Page 6 Lines 22–23)

*“The NPF rates in these regions are highest near the surface and mainly driven by amine–H₂SO₄ nucleation **in our model.**”* (Page 8 Lines 9–10)

*“Hence, H₂SO₄–HNO₃–NH₃ nucleation is **probably** the leading mechanism in the upper troposphere above Eastern China and India.”* (Page 9 Lines 8–9)

*“In the upper troposphere above the Pacific and Atlantic oceans, organic–H₂SO₄ nucleation and H₂SO₄–NH₃–H₂O neutral nucleation are **most likely** the two dominant mechanisms of nucleation according to our model.”* (Page 10 Lines 18–20)

We have also included the sensitivity simulations and related descriptions and discussions in the revised manuscript (Page 39 Line 11 to Page 46 Line 5).

Supplementary Table 1 Summary of model scenarios developed in this study.

Scenario	Description
Best-case	A simulation that includes all 11 NPF mechanisms and uses the R2D-VBS to simulate the nucleating organics. This is our comprehensive best-case scenario and used in most analyses in this study.
No_NPF	A simulation that does not consider any NPF.
NPF_Mech4	A simulation that considers only four traditional inorganic nucleation mechanisms, i.e., the neutral and ion-induced H ₂ SO ₄ –H ₂ O mechanisms and H ₂ SO ₄ –NH ₃ –H ₂ O mechanisms, which resembles the NPF treatment in commonly used climate models.
NPF_Mech11_constYield	A simulation that includes all 11 NPF mechanisms but assumes that pure-organic and organic–H ₂ SO ₄ nucleation is driven by a fixed fraction of the monoterpene oxidation products, following the treatment of a number of previous modeling

	studies^{26,34,36,37}. The specific “fixed fractions” used here followed Gordon et al.⁷. Specifically, organic-H₂SO₄ nucleation was linked to all oxidation products of monoterpenes; in other words, the “fixed fraction” of monoterpene oxidation products used to drive organic-H₂SO₄ nucleation was 1.0. Pure-organic nucleation was assumed driven by highly oxygenated organic molecules (HOMs), the molar yields (fixed fraction) of which were assumed to be 1.4% for the reaction of monoterpenes with O₃ and 0.6% for the reaction of monoterpenes with OH.
0.67*SO ₂ _0.5*DMS	The same as “Best-case” except that the SO ₂ and DMS emissions are reduced by a factor of 1.5 and 2, respectively.
1.5*SO ₂ _2*DMS	The same as “Best-case” except that the SO ₂ and DMS emissions are increased by a factor of 1.5 and 2, respectively.
0.33*H ₂ SO ₄	The same as “Best-case” except that the simulated H ₂ SO ₄ concentrations are reduced by a factor of 3.
3*H ₂ SO ₄	The same as “Best-case” except that the simulated H ₂ SO ₄ concentrations are increased by a factor of 3.
0.33*MT	The same as “Best-case” except that the monoterpene emissions are reduced by a factor of 3.
org-weak-T-dependence	The same as “Best-case” except that a weaker temperature dependence of pure-organic and organic-H ₂ SO ₄ nucleation rates is used.
organic-H ₂ SO ₄ _Riccobono	The same as “Best-case” except that the organic-H ₂ SO ₄ nucleation parameterization is replaced with the one reported in Riccobono et al. ³¹ .
0.4*DMA	The same as “Best-case” except that the simulated DMA concentrations are set to 0.4 times the original simulation results.
amine-H ₂ SO ₄ _Almeida	The same as “Best-case” except that the amine+H ₂ SO ₄ nucleation rate parameterization directly derived from CLOUD chamber experiments reported by Almeida et al. ⁶⁴ is used.
0.33*NH ₃	The same as “Best-case” except that the NH ₃ concentrations are reduced by a factor of 3.
nonuniform-NH ₃	The same as “Best-case” except that the NH ₃ concentration is 1 ppb (consistent with observations in the convective outflow hotspots by Höpfner et al. ⁵⁸) in [average NH ₃]/1 ppb of the area of each model grid and zero in the remaining area of the model grid in the upper troposphere. For the areas with the presence of NH ₃ , we assume that H ₂ SO ₄ is exhausted by nucleation.
0.5*HIO ₃	The same as “Best-case” except that the simulated HIO ₃ concentrations are reduced by a factor of 2.

5*HIO₃

The same as “Best-case” except that the simulated HIO₃ concentrations are increased by a factor of 5.

NPF_Mech4_scaled

In this scenario, we applied a fixed scaling factor to the NPF rates in NPF_Mech4 (which only includes four traditional nucleation mechanisms involving H₂SO₄, NH₃, and H₂O) such that its globally averaged NPF rate matched that of the best-case simulation. Here, the globally averaged NPF rate is defined as the average of the NPF rates across all model grid boxes (both horizontally and vertically), weighted by the volumes of those grid boxes.

Supplementary Fig. 2 Concentrations of precursors directly involved in nucleation as a function of altitude above ground level (AGL) over rainforest regions under the best-case and sensitivity scenarios. The concentrations are averaged in 2016 over the regions specified in Extended Data Fig. 1B. Definitions of the sensitivity experiments are presented in Methods and Supplementary Table 1. To make the figures concise, we show only the concentrations of a precursor in the best-case simulation and in the sensitivity simulations where its concentrations show significant differences from the best case.

Supplementary Fig. 3 Same as Supplementary Fig. 2 but for anthropogenically polluted regions.

Supplementary Fig. 4 Zonal mean concentrations of precursors directly involved in nucleation over the Pacific Ocean (170°E–150°W) under the best-case and sensitivity scenarios in 2016. Definitions of the sensitivity experiments are presented in Methods and Supplementary Table 1. To make the figures concise, we show only the concentrations of a precursor in the best-case simulation and in the sensitivity simulations where its concentrations show significant differences from the best case.

Supplementary Fig. 5 Same as Supplementary Fig. 4 but for the Atlantic Ocean (20°–40°W).

[revised manuscript text omitted]
 alpha-Pinene Oxidation between -50 degrees C and +25 degrees C. *Environ. Sci. Technol.* **53**, 12357-12365, doi:10.1021/acs.est.9b03265 (2019).
- Yan, C. *et al.* Size-dependent influence of NO_x on the growth rates of organic aerosol particles. *Sci. Adv.* **6**, eaay4945, doi:10.1126/sciadv.aay4945 (2020).
- Bergman, T. *et al.* Geographical and diurnal features of amine-enhanced boundary layer nucleation. *J. Geophys. Res-Atmos.* **120**, 9606-9624, doi:10.1002/2015jd023181 (2015).
- Wang, M. *et al.* Synergistic HNO₃-H₂SO₄-NH₃ upper tropospheric particle formation. *Nature* **605**, 483-489, doi:10.1038/s41586-022-04605-4 (2022).
- Dunne, E. M. *et al.* Global atmospheric particle formation from CERN CLOUD measurements. *Science*

- 354, 1119-1124, doi:10.1126/science.aaf2649 (2016).
- Feng, L. *et al.* The generation of gridded emissions data for CMIP6. *Geosci. Model. Dev.* **13**, 461-482, doi:10.5194/gmd-13-461-2020 (2020).
- Hoesly, R. M. *et al.* Historical (1750-2014) anthropogenic emissions of reactive gases and aerosols from the Community Emissions Data System (CEDS). *Geosci. Model. Dev.* **11**, 369-408, doi:10.5194/gmd-11-369-2018 (2018).
- Smith, S. J. *et al.* Anthropogenic sulfur dioxide emissions: 1850-2005. *Atmos. Chem. Phys.* **11**, 1101-1116, doi:10.5194/acp-11-1101-2011 (2011).
- Crippa, M. *et al.* Gridded emissions of air pollutants for the period 1970-2012 within EDGAR v4.3.2. *Earth System Science Data* **10**, 1987-2013, doi:10.5194/essd-10-1987-2018 (2018).
- Wang, S., Elliott, S., Maltrud, M. & Cameron-Smith, P. Influence of explicit Phaeocystis parameterizations on the global distribution of marine dimethyl sulfide. *Journal of Geophysical Research-Biogeosciences* **120**, 2158-2177, doi:10.1002/2015jg003017 (2015).
- Wang, S., Maltrud, M., Elliott, S., Cameron-Smith, P. & Jonko, A. Influence of dimethyl sulfide on the carbon cycle and biological production. *Biogeochemistry* **138**, 49-68, doi:10.1007/s10533-018-0430-5 (2018).
- Lana, A. *et al.* An updated climatology of surface dimethylsulfide concentrations and emission fluxes in the global ocean. *Global. Biogeochem. Cy.* **25**, Gb1004, doi:10.1029/2010gb003850 (2011).
- Guenther, A. B. *et al.* The Model of Emissions of Gases and Aerosols from Nature version 2.1 (MEGAN2.1): an extended and updated framework for modeling biogenic emissions. *Geosci. Model. Dev.* **5**, 1471-1492, doi:10.5194/gmd-5-1471-2012 (2012).
- Shilling, J. E. *et al.* Aircraft observations of the chemical composition and aging of aerosol in the Manaus urban plume during GoAmazon 2014/5. *Atmos. Chem. Phys.* **18**, 10773-10797, doi:10.5194/acp-18-10773-2018 (2018).
- Langford, B. *et al.* Fluxes and concentrations of volatile organic compounds from a South-East Asian tropical rainforest. *Atmos. Chem. Phys.* **10**, 8391-8412, doi:10.5194/acp-10-8391-2010 (2010).
- Cai, R. *et al.* The missing base molecules in atmospheric acid-base nucleation. *Natl. Sci. Rev.* **9**, nwac137, doi:10.1093/nsr/nwac137 (2022).
- Hoepfner, M. *et al.* First detection of ammonia (NH₃) in the Asian summer monsoon upper troposphere. *Atmos. Chem. Phys.* **16**, 14357-14369, doi:10.5194/acp-16-14357-2016 (2016).
- Hoepfner, M. *et al.* Ammonium nitrate particles formed in upper troposphere from ground ammonia sources during Asian monsoons. *Nat. Geosci.* **12**, 608-612, doi:10.1038/s41561-019-0385-8 (2019).
- Buchholz, R. R., Emmons, L. K., Tilmes, S. & The CESM2 Development Team. CESM2.1/CAM-chem Instantaneous Output for Boundary Conditions, available at <https://doi.org/10.5065/NMP7-EP60>. (UCAR/NCAR - Atmospheric Chemistry Observations and Modeling Laboratory, 2019).
- McDuffie, E. E. *et al.* A global anthropogenic emission inventory of atmospheric pollutants from sector- and fuel-specific sources (1970-2017): an application of the Community Emissions Data System (CEDS). *Earth System Science Data* **12**, 3413-3442, doi:10.5194/essd-12-3413-2020 (2020).
- Bouwman, A. F. *et al.* A global high-resolution emission inventory for ammonia. *Global. Biogeochem. Cy.* **11**, 561-587, doi:10.1029/97gb02266 (1997).
- Fowler, D. *et al.* Effects of global change during the 21st century on the nitrogen cycle. *Atmos. Chem. Phys.* **15**, 13849-13893, doi:10.5194/acp-15-13849-2015 (2015).
- Paulot, F. *et al.* Global oceanic emission of ammonia: Constraints from seawater and atmospheric observations. *Global. Biogeochem. Cy.* **29**, 1165-1178, doi:10.1002/2015gb005106 (2015).
- Almeida, J. *et al.* Molecular understanding of sulphuric acid-amine particle nucleation in the atmosphere. *Nature* **502**, 359-363, doi:10.1038/nature12663 (2013).

Reviewer Reports on the First Revision:

Referees' comments:

Referee #1 (Remarks to the Author):

The authors have satisfactorily responded to my comments from the first round of review, and they have, in my judgment, thoroughly and satisfactorily responded to the comments of the other two reviewers. The additional sensitivity studies and expanded discussion of previous work are well appreciated. In reading the revised manuscript, I did not find additional issues that needed correction, either introduced or existing. I recommend the article for publication.

+++++

Additional comments in response to referee #3's remaining concerns

One of the concerns stated by referee #3 was whether or not NPF takes place via H_2SO_4 -ions in the free troposphere over Europe and Eastern US due to uncertainties in emissions of H_2SO_4 precursors. This is specifically addressed in extended data Fig. 6, through two different sensitivity studies.

Referee #3 also expresses concerns whether or not NPF takes place about via H_2SO_4 - HNO_3 - NH_3 in the upper troposphere over Asia, due to uncertainties in emissions of H_2SO_4 , HNO_3 , and NH_3 precursors. The authors address the concentrations of NH_3 through extended data Fig. 4 and the related discussion. They address the sensitivity to the concentrations of H_2SO_4 and H_2SO_4 precursors in extended data Fig. 6. I don't currently see any sensitivity to uncertainties in the HNO_3 concentration, but I don't believe that referee #3 was suggesting that nucleation occurs over Asia via H_2SO_4 - NH_3 - H_2O instead of H_2SO_4 - HNO_3 - NH_3 . Additionally, Wang et al. (2022) (the authors' reference 15) states that the nucleation rate depends more weakly on HNO_3 concentrations than either H_2SO_4 or NH_3 concentrations (their eq. 6, square dependence vs. cubic or quartic).

I therefore feel that it would be legitimate to ask the authors to comment on why a sensitivity study wasn't necessary for the HNO_3 concentrations. Given the lower sensitivity of NPF to HNO_3 , and I suspect a lower uncertainty in HNO_3 concentrations, I do not expect this to alter the authors' conclusions.

Regarding the necessity of observations to challenge the model: The authors state that "our model evaluation over key regions suggests that these increased number concentrations are realistic, but further evaluation studies using more observations would be invaluable." This is the invitation I was looking for, but an additional sentence specifying the regions most in need of observational campaigns would be helpful.

Referee #2 (Remarks to the Author):

In their revised manuscript, the authors have added a few more relevant sensitivity simulations and

enhanced the discussion on how the current paper compares/adds compared with earlier research on atmospheric new particle formation. In my opinion, the authors have successfully answered the main critics by reviewers, thereby further improving the already high quality of the submitted paper. As a result, I recommend acceptance of this paper for publication in its present form.

Referee #3 (Remarks to the Author):

I am still not convinced with the paper's conclusions for the same reasons I mentioned previously - due to high uncertainties in emissions of precursors in models. I think with the current status of the emission inventories, we cannot model the precursor concentrations. I am skeptical that NPF takes place via $\text{H}_2\text{SO}_4\text{-HNO}_3\text{-NH}_3$ in the upper troposphere over Asia, via H_2SO_4 -ions in the free troposphere over Europe and Eastern US, for example. Currently, there are no atmospheric measurements that can be used to validate the model simulations. How do atmospheric dynamics and transport affect the precursors in the upper atmosphere from the tropics to high latitudes? In laboratory chamber settings, depending on temperature precursors, one can simulate different NPF processes, but applying them to the model with highly uncertain precursors will produce grandiose but highly uncertain results.

Author Rebuttals to First Revision:

**Referees' comments:**

**Referee #1 (Remarks to the Author):**

The authors have satisfactorily responded to my comments from the first round of review, and
they have, in my judgment, thoroughly and satisfactorily responded to the comments of the other
two reviewers. The additional sensitivity studies and expanded discussion of previous work are
well appreciated. In reading the revised manuscript, I did not find additional issues that needed
correction, either introduced or existing. I recommend the article for publication.

We thank the referee for supporting the publication of our manuscript. We also greatly appreciate
the constructive comments of the referee in response to Referee #3's remaining concerns, which
were very helpful in guiding our revision. We have carefully revised the manuscript in
accordance with those comments, and our responses are provided below. The comments of the
referee are presented in blue text, our responses are written in black text, and quotations from our
manuscript are presented in italic type.

++++++

*Additional comments in response to referee #3's remaining concerns*

One of the concerns stated by referee #3 was whether or not NPF takes place via H_2SO_4 -ions
in the free troposphere over Europe and Eastern US due to uncertainties in emissions of H_2SO_4
precursors. This is specifically addressed in extended data Fig. 6, through two different
sensitivity studies.

Referee #3 also expresses concerns whether or not NPF takes place about via H_2SO_4 - HNO_3 - NH_3
in the upper troposphere over Asia, due to uncertainties in emissions of H_2SO_4 , HNO_3 , and NH_3
precursors. The authors address the concentrations of NH_3 through extended data Fig. 4 and the
related discussion. They address the sensitivity to the concentrations of H_2SO_4 and H_2SO_4
precursors in extended data Fig. 6. I don't currently see any sensitivity to uncertainties in the
HNO_3 concentration, but I don't believe that referee #3 was suggesting that nucleation occurs
over Asia via H_2SO_4 - NH_3 - H_2O instead of H_2SO_4 - HNO_3 - NH_3 . Additionally, Wang et al. (2022)
(the authors' reference 15) states that the nucleation rate depends more weakly on HNO_3
concentrations than either H_2SO_4 or NH_3 concentrations (their eq. 6, square dependence vs. cubic
or quartic).

I therefore feel that it would be legitimate to ask the authors to comment on why a sensitivity
study wasn't necessary for the HNO_3 concentrations. Given the lower sensitivity of NPF to
HNO_3 , and I suspect a lower uncertainty in HNO_3 concentrations, I do not expect this to alter the
authors' conclusions.

We thank the referee for identifying the remaining uncertainty in the rate of H_2SO_4 - HNO_3 - NH_3
nucleation, linked to variability in HNO_3 concentration. To address this uncertainty, we evaluated

the simulated concentrations of HNO₃ against observations from the Microwave Limb Sounder
(MLS) aboard the Aura satellite¹. We used the level 3 monthly HNO₃ product for the evaluation².
Supplementary Fig. 11 (shown below) displays simulated and observed 2016 mean HNO₃
concentrations at 150 hPa (approximately 13 km), corresponding broadly to the location with the
highest NPF rate in our model. It also represents one of the few vertical levels provided by the
Aura MLS level 3 product within the upper troposphere where NPF rates are notably high. The
simulations generally agree well with the observations, with average concentrations of 0.562 and
0.544 ppb, respectively, over the Asian monsoon region (8°–40°N, 60°–130°E). Given that the
H₂SO₄–HNO₃–NH₃ nucleation rate has relatively weak (quadratic) dependence on HNO₃
concentration, the uncertainty in HNO₃ concentration should not change our conclusion
regarding the role of H₂SO₄–HNO₃–NH₃ nucleation. We have incorporated the aforementioned
results and discussion into the revised manuscript (Page 45 Line 23 to Page 46 Line 12).

**Supplementary Fig. 11 Comparison of simulated 2016 mean HNO₃ concentrations in the upper troposphere**
**(150 hPa, approximately 13 km) over the Asian monsoon region with observations from the Microwave Limb**
**Sounder (MLS) aboard the Aura satellite.**

Regarding the necessity of observations to challenge the model: The authors state that "our
model evaluation over key regions suggests that these increased number concentrations are
realistic, but further evaluation studies using more observations would be invaluable." This is the
invitation I was looking for, but an additional sentence specifying the regions most in need of
observational campaigns would be helpful.

After reviewing the availability of observational data in different regions, we think that
observational campaigns that simultaneously measure nucleation precursors and particle size
distributions are greatly needed, especially in the upper troposphere above Southeastern Asia,
Central Africa, the Asian monsoon region (including Eastern China and India), the Eastern
United States, and Europe. Furthermore, the detection of molecular clusters using state-of-the-art
techniques such as chemical ionization mass spectrometry could provide direct insights into the
precursors involved in nucleation; however, this has been accomplished only in limited regions³⁻⁶.
We suggest that such measurements should be obtained in the boundary layer of the Eastern
United States, India, and the Pacific and Atlantic oceans, as well as in the free troposphere over

all types of regions of interest. We have added the following sentence to the revised manuscript:

*“Simultaneous measurements of nucleation precursors and particle size distributions are greatly*
*needed, especially in the upper troposphere above Southeastern Asia, Central Africa, the Asian*
*monsoon regions, the Eastern United States, and Europe. Furthermore, direct detection of*
*molecular clusters in various regions of the world is encouraged.”* (Page 14 Lines 9–12)

**Referee #2 (Remarks to the Author):**

In their revised manuscript, the authors have added a few more relevant sensitivity simulations
and enhanced the discussion on how the current paper compares/adds compared with earlier
research on atmospheric new particle formation. In my opinion, the authors have successfully
answered the main critics by reviewers, thereby further improving the already high quality of the
submitted paper. As a result, I recommend acceptance of this paper for publication in its present
form.

We thank the referee for supporting the publication of our manuscript, and we are greatly
appreciative of the valuable time and the considerable effort that the referee committed to the
review process.

**Referee #3 (Remarks to the Author):**

I am still not convinced with the paper's conclusions for the same reasons I mentioned previously
- due to high uncertainties in emissions of precursors in models. I think with the current status of
the emission inventories, we cannot model the precursor concentrations. I am skeptical that NPF
takes place via H₂SO₄-HNO₃-NH₃ in the upper troposphere over Asia, via H₂SO₄-ions in the free
troposphere over Europe and Eastern US, for example. Currently, there are no atmospheric
measurements that can be used to validate the model simulations. How do atmospheric dynamics
and transport affect the precursors in the upper atmosphere from the tropics to high latitudes? In
laboratory chamber settings, depending on temperature precursors, one can simulate different
NPF processes, but applying them to the model with highly uncertain precursors will produce
grandiose but highly uncertain results.

We thank the referee for his/her additional comments, which helped us further improve the
quality of our manuscript. Our detailed responses to the specific concerns of the referee are given
below. In brief, to address the referee's doubt concerning the NPF mechanisms in several regions
attributable to uncertainties in precursor emissions/concentrations, we have added further model
evaluations and sensitivity simulation results to show that our main findings are likely unaffected
by such uncertainties in precursors. We have also shown through sensitivity simulations that the
uncertainties in atmospheric dynamics and transport are unlikely to change our main findings
regarding NPF mechanisms in the main regions of interest. Meanwhile, we recognize that the
aforementioned uncertainties, while not altering our main conclusions, might affect the precise
values of the contributions from individual NPF mechanisms, and that there could be

uncertainties beyond those currently known.

There is also a philosophical nature to this discussion. We agree that precursor emissions often
have considerable uncertainty. However, claiming that the precursor emissions are so uncertain
that there is little value in conducting model simulations is unwarranted in our opinion. We
suggest that there is still great value in exploring the current state of our understanding, in this
case regarding a growing suite of NPF mechanisms, to elucidate what we do know about the
relative importance of those mechanisms; this was our objective. However, we completely agree
that careful consideration of the uncertainties, including those derived from precursor emissions,
is an important element of this presentation. Therefore, we have more clearly stated the
uncertainties/limitations of our study in the revised manuscript.

First, the referee expressed doubt about the NPF mechanisms in the upper troposphere over Asia.
We will briefly recap our main results in this regard. Our model showed that synergistic H₂SO₄–
HNO₃–NH₃ nucleation is important and often dominant in the upper tropospheric above Eastern
China and India, while H₂SO₄–NH₃–H₂O neutral nucleation plays a secondary but sometimes
comparable role (Fig. 2B). Furthermore, the actual contribution of H₂SO₄–HNO₃–NH₃
nucleation is probably even higher than that of the above baseline simulation result because the
real-world NH₃ concentration is highly nonuniform within a 1° × 1° model grid, and because the
H₂SO₄–HNO₃–NH₃ nucleation rate is much more strongly dependent on the NH₃ concentration
than linear (Extended Data Fig. 9 and related discussion in Page 44 Lines 3–23 of the revised
manuscript); this makes H₂SO₄–HNO₃–NH₃ nucleation even more likely to be the leading
mechanism in this region. Concerning the potential uncertainties stemming from precursor
emissions/concentrations, we previously addressed the uncertainty of H₂SO₄ and H₂SO₄
precursors (SO₂ and DMS) by performing four sensitivity simulations (Extended Data Fig. 6 and
related discussion in Page 39 Line 19 to Page 41 Line 3 of the revised manuscript). Additionally,
we have addressed the uncertainty of NH₃ by comparing simulated NH₃ concentrations in this
region with observations and by conducting a sensitivity simulation (Extended Data Figs. 4 and 9
and related discussion in Page 44 Lines 3–23 of the revised manuscript). The results indicated
that our major conclusions are unlikely to be affected by uncertainties in H₂SO₄ and NH₃. Here,
we further evaluate the simulated concentrations of the third precursor, HNO₃, against
observations from the Microwave Limb Sounder (MLS) aboard the Aura satellite¹. We used the
level 3 monthly HNO₃ product for the evaluation². Supplementary Fig. 11 (shown below)
displays simulated and observed 2016 mean HNO₃ concentrations at 150 hPa (approximately 13
33 km), corresponding broadly to the location with the highest NPF rate in our model. It also
represents one of the few vertical levels provided by the Aura MLS level 3 product within the
upper troposphere where NPF rates are notably high. The simulations generally agree well with
observations, with average concentrations of 0.562 and 0.544 ppb, respectively, over the Asian
monsoon region (8°–40°N, 60°–130°E). Given that the H₂SO₄–HNO₃–NH₃ nucleation rate has
relatively weak (quadratic) dependence on HNO₃ concentration⁷, the uncertainty in HNO₃
concentration should not change our conclusion about the main nucleation mechanism. We have

incorporated the aforementioned results and discussion into the revised manuscript (Page 45 Line
23 to Page 46 Line 12).

**Supplementary Fig. 11 Comparison of simulated 2016 mean HNO₃ concentrations in the upper troposphere**
**(150 hPa, approximately 13 km) over the Asian monsoon region with observations from the Microwave Limb**
**Sounder (MLS) aboard the Aura satellite.**

Regarding the free troposphere over Europe and the Eastern United States, the referee is
skeptical that NPF occurs via H₂SO₄-ion nucleation. It is noted that H₂SO₄-H₂O ion-induced
nucleation makes minimal contribution at any height in these regions; therefore, we believe that
the referee was referring to the large contribution of H₂SO₄-NH₃-H₂O ion-induced nucleation at
heights of 0.7–4.5 km over these two regions (Fig. 2 in the manuscript). First, we would like to
clarify that we did not specifically discuss the NPF mechanisms within the above height range,
and nor did we state that H₂SO₄-NH₃-H₂O ion-induced nucleation was the dominant mechanism
within that range. Instead, given the word count limitation, we emphasized the NPF mechanisms
over particle hotspots; for human-polluted regions including Europe and the Eastern United
States, these hotspots are located in the boundary layer and in the upper troposphere, rather than
at heights of 0.7–4.5 km. Nevertheless, our model does quantify NPF mechanisms over all
regions and heights globally, and Fig. 2 shows that H₂SO₄-NH₃-H₂O ion-induced nucleation is
an important NPF mechanism at heights of 0.7–4.5 km and predominates within certain height
ranges over Europe and the Eastern United States.

Here, given the referee's concerns, we examine the sensitivity of these results to the uncertainties
of precursor emissions. In our previous response letter, we described four sensitivity scenarios
that perturbed either H₂SO₄ concentration or emissions of H₂SO₄ precursors between their
upper/lower bounds, and the results presented in Extended Data Fig. 6 showed that, under any
sensitivity scenario, H₂SO₄-NH₃-H₂O ion-induced nucleation remains an important nucleation
mechanism at heights of 0.7–4.5 km and the dominant one at certain heights. We also conducted
another sensitivity simulation ("0.33*NH₃") by reducing NH₃ concentrations in the best-case
simulation by a factor of 3, given that the uncertainties in anthropogenic and marine NH₃
emissions are both within a factor of 3 (see explanations in Page 45 Lines 5–13 of the revised
manuscript). Results shown in Figure R1 below indicate that even with lower-bound NH₃

emissions, our finding regarding the important role of $\text{H}_2\text{SO}_4\text{-NH}_3\text{-H}_2\text{O}$ ion-induced nucleation
 remains valid. Therefore, in the free troposphere at heights of 0.7–4.5 km above Europe and the
 Eastern United States, $\text{H}_2\text{SO}_4\text{-NH}_3\text{-H}_2\text{O}$ ion-induced nucleation very likely represents an
 important nucleation mechanism and the dominant one within certain height ranges, although its
 exact quantitative contribution fluctuates to some extent under different sensitivity scenarios.

 **Figure R1. NPF rates as a function of height above ground level (AGL) over Europe and the Eastern United**
 **States under the best-case scenario and a sensitivity scenario that reduces NH_3 concentration by a factor of 3.**
 White lines represent the total NPF rates of all mechanisms at diameter of 1.7 nm ($J_{1.7}$, on a log scale), and the
 colored areas represent the relative contributions of different mechanisms, both averaged for 2016 over the regions
 specified in Extended Data Fig. 1B.

 The referee also highlighted the scarcity of observations available for validation of the model
 simulations. It should be noted that we tried our best to evaluate the simulated precursor
 concentrations and/or particle size distributions over key regions, including the Amazon,
 Southeastern Asia, Eastern China, India, the United States, Europe, the Pacific Ocean, and the
 Atlantic Ocean (Figs. 1–3, Extended Data Figs. 3–4, Supplementary Fig. 11, Page 7 Lines 18–21,
 and Page 9 Lines 7–10); the overall good consistency between the simulations and the
 observations enhanced our confidence in the discussion on NPF mechanisms. For example, in
 the upper troposphere over Asia, as mentioned by the referee, we have evaluated the simulated
 NH_3 and HNO_3 concentrations against the observations and revealed reasonably good model–
 observation agreement. However, we do acknowledge that more observational data and further
 model evaluation against observations would be invaluable. Again, in our opinion, the scarcity of
 observations supports the publication of our manuscript, not the reverse. Our work represents the
 best effort to represent state-of-the-art understanding in the presence of unknowns, but it
 certainly also brings to light where those unknowns are especially vexing. Following the
 comment of Referee #1 on the concerns raised by Referee #3, we have specified those regions
 most in need of additional observational studies. Specifically, observational campaigns that

simultaneously measure nucleation precursors and particle number size distributions are needed
in the upper troposphere above Southeastern Asia, Central Africa, the Asian monsoon region
(including Eastern China and India), the Eastern United States, and Europe. Furthermore, the
detection of molecular clusters using state-of-the-art techniques such as chemical ionization mass
spectrometry could provide direct insights into the precursors involved in nucleation; however,
this has been achieved only in limited regions³⁻⁶. We suggest that such measurements should be
obtained in the boundary layer of the Eastern United States, India, and the Pacific and Atlantic
oceans, as well as in the free troposphere over all types of regions of interest. Our study
integrated existing experimental and observational data to build and evaluate the model, which in
turn, has been used to inform more targeted observational efforts for the future. We believe this
reinforces the value of this study in advancing the understanding of NPF mechanisms. We have
incorporated the above recommendation for more observational studies and further model
evaluations into the revised manuscript (Page 14 Lines 9–12).

The referee also raised questions regarding the influence of atmospheric dynamics and transport
on precursors and NPF mechanisms in the upper atmosphere. The transport of precursors to the
upper troposphere occurs via two major processes in the model: large-scale transport resolved by
the model grids and unresolved convective transport that must be parameterized. To accurately
simulate large-scale transport, we nudged the wind fields to the Modern-Era Retrospective
analysis for Research and Applications, version 2 (MERRA-2) reanalysis data, following Sun et
al.⁸ Reanalysis data represent a blend of observations and weather simulations realized through
data assimilation, and they provide the most accurate complete picture of past weather and
climate that is currently available. Sun et al.⁸ showed that nudging E3SM simulations to the
MERRA-2 reanalysis data could produce simulated grid-scale winds that closely resemble those
of MERRA-2, with spatial and temporal correlations of >0.9 for both the lower and the upper
troposphere. Therefore, we believe that the simulation of large-scale transport, based on the best
available and highly mature method, should not cause large uncertainty in precursor
concentrations and NPF mechanisms.

The simulation of subgrid convective transport is comparatively uncertain because of the
difficulty in directly applying observational constraints. In the E3SM model, deep convection
and the associated convective transport were simulated using the ZM convection scheme⁹. We
employed an improved version, as described by Wang et al.^{10,11}, which uses a unified treatment
of the convective transport of gases and aerosols. To understand the uncertainty associated with
the deep convection scheme, Qian et al.¹² performed numerous sensitivity simulations where
seven main parameters of the ZM convection scheme and certain other parameters in E3SM were
perturbed simultaneously within their possible ranges using the Latin hypercube sampling
method. They showed that over 85% of the total variance of precipitation, an indicator of
convection development that controls convective transport, could generally be explained by two
out of the seven ZM parameters: 1) the time scale for the consumption rate of convective
available potential energy (hereafter, denoted by “tau”) and 2) the fractional mass entrainment

rate (hereafter, denoted by “dmpdz”), defined as the fractional air mass flux entrained into a
volume of cloudy air per unit height¹³. Yang et al.¹⁴ reached almost the same conclusion
regarding the governing parameters for the ZM convection scheme implemented in the
Community Atmosphere Model version 5. To better address the reviewer’s concern, we further
performed two sensitivity simulations (“upper_tau” and “lower_tau”) that set the value of tau to
the upper and lower bounds (14,400 and 1,800 s, respectively) of the possible range specified by
Qian et al.¹², as compared to the optimized value of 3,600 s in the best-case simulation. Similarly,
we conducted two sensitivity simulations for dmpdz (“upper_dmpdz” and “lower_dmpdz”),
which changed the value of dmpdz from $0.7 \times 10^{-3} \text{ m}^{-1}$ in the best-case simulation to 2.0×10^{-3}
m^{-1} and $0.1 \times 10^{-3} \text{ m}^{-1}$, respectively. Supplementary Figs. 6–9 summarize the contributions of
different NPF mechanisms in these sensitivity simulations over the main regions of interest in
this study, including rainforests, anthropogenically polluted regions, and the Pacific and Atlantic
oceans. The results indicate that the sensitivity scenarios have limited influence on the relative
contributions of individual NPF mechanisms in the upper troposphere over these regions, largely
because the perturbation of convective transport simultaneously changes the concentrations of
multiple nucleation precursors, which subsequently causes simultaneous change in the rates of
most NPF mechanisms. We have included the new results and relevant discussion in the revised
manuscript (Page 47 Line 1 to Page 48 Line 18).

In summary, our model evaluations and sensitivity simulations, including those in the previous
version of the manuscript and those newly added here, indicate that the currently recognized
uncertainties in precursor emissions/concentrations and atmospheric transport are unlikely to
change our main findings regarding the leading NPF mechanisms in the main regions of interest.
Nevertheless, we acknowledge that the uncertainties might affect the exact quantitative
contributions of individual mechanisms, both in the above key regions and in other areas not
discussed in detail individually. Meanwhile, there might be uncertainties beyond our current
level of knowledge. Therefore, the revised manuscript includes clearer statements regarding the
uncertainties/limitations of our study, as listed below.

In the Introduction section:

*“Our systematic sensitivity analysis suggests that the presently quantifiable uncertainties are*
*unlikely to change our main findings but might affect the exact quantitative contributions of*
*individual mechanisms. Additionally, potential uncertainties beyond our current knowledge*
*might further refine and possibly modify the findings we present.”* (Page 6 Lines 4–8)

In the “Global overview and sensitivity analysis” section:

*“Results indicate that our main findings regarding the leading NPF mechanisms hold true under*
*these sensitivity simulations in all key regions of interest. Nevertheless, these sources of*
*uncertainty might affect the precise quantitative contributions of individual mechanisms, both in*
*the above key regions and in other areas not discussed in detail individually.”* (Page 12 Lines 3–
7)

In the Discussion section:

*“Our sensitivity experiments showed that the findings are robust across 13 key sources of*
*uncertainty, but uncertainty might remain in aspects not covered by these experiments. ... Our*
*model evaluation over key regions suggests that the changes in number concentrations are*
*realistic, but further evaluation using more observations would be invaluable. In particular,*
*simultaneous measurements of nucleation precursors and particle size distributions are greatly*
*needed, especially in the upper troposphere above Southeastern Asia, Central Africa, the Asian*
*monsoon regions, the Eastern United States, and Europe. Furthermore, direct detection of*
*molecular clusters in various regions of the world is encouraged.”* (Page 13 Line 13 to Page 14
Line 2, Page 14 Lines 7–12)

 **Supplementary Fig. 6** NPF rates as a function of height above ground level (AGL) over rainforests under the
 **best-case scenario and sensitivity scenarios perturbing key parameters in the ZM deep convection scheme.**
 White lines represent the total NPF rates of all mechanisms at diameter of 1.7 nm ($J_{1.7}$, on a log scale), and the
 colored areas represent the relative contributions of different mechanisms, both averaged in 2016 over the regions
 specified in Extended Data Fig. 1B. Definitions of the sensitivity experiments are presented in Methods and
 Supplementary Table 1.

1

2 Supplementary Fig. 7 Same as Supplementary Fig. 6 but for anthropogenically polluted regions.

 **Supplementary Fig. 8 Zonal mean NPF rates of individual mechanisms over the Pacific Ocean (170°E–150°W)**
 **under the best-case scenario and sensitivity scenarios perturbing key parameters in the ZM deep convection**
 **scheme in 2016.** Only five NPF mechanisms are shown because the other mechanisms are negligible in these
 regions. Definitions of the sensitivity experiments are presented in Methods and Supplementary Table 1.

Supplementary Fig. 9 Same as Supplementary Fig. 8 but for the Atlantic Ocean ($20^\circ\text{--}40^\circ\text{W}$).

References

- Livesey, N. J., Van Snyder, W., Read, W. G. & Wagner, P. A. Retrieval algorithms for the EOS Microwave Limb Sounder (MLS). *IEEE. T. Geosci. Remote.* **44**, 1144-1155, doi:10.1109/tgrs.2006.872327 (2006).
- Livesey, N. J. *et al.* Earth Observing System (EOS) Aura Microwave Limb Sounder (MLS) Version 5.0x Level 2 and 3 data quality and description document. (Jet Propulsion Laboratory, 2022).
- Yao, L. *et al.* Atmospheric new particle formation from sulfuric acid and amines in a Chinese megacity. *Science* **361**, 278-281, doi:10.1126/science.aao4839 (2018).
- Cai, R. *et al.* Sulfuric acid-amine nucleation in urban Beijing. *Atmos. Chem. Phys.* **21**, 2457-2468, doi:10.5194/acp-21-2457-2021 (2021).
- Baccarini, A. *et al.* Frequent new particle formation over the high Arctic pack ice by enhanced iodine emissions (vol 11, 4924, 2020). *Nat. Commun.* **11**, 5557, doi:10.1038/s41467-020-19533-y (2020).
- Bianchi, F. *et al.* New particle formation in the free troposphere: A question of chemistry and timing. *Science* **352**, 1109-1112, doi:10.1126/science.aad5456 (2016).
- Wang, M. *et al.* Synergistic HNO₃-H₂SO₄-NH₃ upper tropospheric particle formation. *Nature* **605**, 483-489, doi:10.1038/s41586-022-04605-4 (2022).
- Sun, J. *et al.* Impact of Nudging Strategy on the Climate Representativeness and Hindcast Skill of Constrained EAMv1 Simulations. *J. Adv. Model. Earth Syst.* **11**, 3911-3933, doi:10.1029/2019ms001831 (2019).
- Zhang, G. J. & McFarlane, N. A. Sensitivity of climate simulations to the parameterization of cumulus convection in the Canadian climate centre general circulation model. *Atmos.-Ocean* **33**, 407-446, doi:10.1080/07055900.1995.9649539 (1995).
- Wang, H. *et al.* Sensitivity of remote aerosol distributions to representation of cloud-aerosol interactions in a global climate model. *Geosci. Model. Dev.* **6**, 765-782, doi:10.5194/gmd-6-765-2013 (2013).
- Wang, H. *et al.* Aerosols in the E3SM Version 1: New Developments and Their Impacts on Radiative Forcing. *J. Adv. Model. Earth Syst.* **12**, e2019MS001851, doi:10.1029/2019ms001851 (2020).
- Qian, Y. *et al.* Parametric Sensitivity and Uncertainty Quantification in the Version 1 of E3SM Atmosphere Model Based on Short Perturbed Parameter Ensemble Simulations. *J. Geophys. Res-Atmos.* **123**, 13046-13073, doi:10.1029/2018jd028927 (2018).
- Xu, X. *et al.* Factors Affecting Entrainment Rate in Deep Convective Clouds and Parameterizations. *J. Geophys. Res-Atmos.* **126**, e2021JD034881, doi:10.1029/2021jd034881 (2021).
- Yang, B. *et al.* Uncertainty quantification and parameter tuning in the CAM5 Zhang-McFarlane convection scheme and impact of improved convection on the global circulation and climate. *J. Geophys. Res-Atmos.* **118**, 395-415, doi:10.1029/2012jd018213 (2013).
- Elm, J. *et al.* Modeling the formation and growth of atmospheric molecular clusters: A review. *J. Aerosol. Sci.* **149**, 105621, doi:10.1016/j.jaerosci.2020.105621 (2020).
- Yin, R. *et al.* Acid-Base Clusters during Atmospheric New Particle Formation in Urban Beijing. *Environ. Sci. Technol.* **55**, 10994-11005, doi:10.1021/acs.est.1c02701 (2021).
- Glasoe, W. A. *et al.* Sulfuric acid nucleation: An experimental study of the effect of seven bases. *J. Geophys. Res-Atmos.* **120**, 1933-1950, doi:10.1002/2014jd022730 (2015).
- Liu, L. *et al.* Rapid sulfuric acid-dimethylamine nucleation enhanced by nitric acid in polluted regions. *P. Natl. Acad. Sci. USA.* **118**, e2108384118, doi:10.1073/pnas.2108384118|lof6 (2021).
- Lee, S.-H. *et al.* New Particle Formation in the Atmosphere: From Molecular Clusters to Global Climate. *J. Geophys. Res-Atmos.* **124**, 7098-7146, doi:10.1029/2018jd029356 (2019).
- Kulmala, M. *et al.* Is reducing new particle formation a plausible solution to mitigate particulate air pollution in Beijing and other Chinese megacities? *Faraday. Discuss.* **226**, 334-347, doi:10.1039/d0fd00078g (2021).
- Williamson, C. J. *et al.* A large source of cloud condensation nuclei from new particle formation in the tropics. *Nature* **574**, 399-403, doi:10.1038/s41586-019-1638-9 (2019).

Reviewer Reports on the Second Revision:

Referees' comments:

Referee #1 (Remarks to the Author):

I am fully satisfied with the authors' response to the comments of referee #3, and my comments on referee #3's comments. I strongly feel that the current manuscript merits publication.

Referee #3 (Remarks to the Author):

The second revision is still not convincing with the conclusion (for the reasons stated below in detail), and it raises many more questions. A recent perspective article by Kirkby et al. (2023) in Nature Geoscience summarized the key findings from CLOUD chamber experiments and gave “qualitative” predictions of NPF processes in different regions of the atmosphere. It is assuring to know how different nucleation processes can be applied to or can take place in different regions under different conditions. However, the present study does not show anything newer than the Kirkby et al. (not cited in the manuscript) in a scientific context and does not have a novelty for publication in Nature. The conclusion feels like an overstatement for the reasons stated below.

This manuscript stresses multiple times that sensitivity analysis does not change their conclusion, and in fact, the figures (Extended Figures 5-8 and Supplementary Figures 3-5) show very little changes during the sensitivity analysis. This is not surprising and it is expected because the paper chose to use extremely narrow ranges of precursor concentrations (all within the same order), while their concentrations can vary over many orders of magnitude in the atmosphere. The paper states that SO₂ emission inventory has a maximum less than 50% uncertainty, this is a very optimistic view even for SO₂ in global models. For other species, such as DMA, monoterpene and HIO₃ over open ocean, and HNO₃ in the upper troposphere, their atmospheric concentrations can vary by orders of magnitude. Therefore, the sensitivity analysis (conducted by varying a single precursor concentrations with a factor of 3-5) does not provide compelling confidence in the conclusion. Considering nucleation is a non-linear process, sensitivity analysis should produce huge differences in the outcome, and the reason it shows consistent results is because of the unreasonably narrow range of concentrations tested. Therefore, the sensitivity analysis is not vigorous or statistically useful.

In more detail:

1. It is not convincing that pure organic nucleation is dominant over SE Asia and the Amazon in the upper troposphere. Because sulfuric acid is so dominant in nucleation, even slight SO₂ (convected from the surface sources or long-range transported from polluted regions) can overwhelm pure biogenic nucleation. So, under present-day conditions, pure biogenic nucleation is unlikely to take place in the real atmosphere. In fact, until now, pure biogenic new particle formation has not been observed in the real atmosphere (with extremely few exceptions, only one or two cases for an extremely short period).
2. It is not clear why in the Amazon, the surface nucleation rate is so low, with abundant

monoterpene and high oxidants, whereas in SE Asia (similar conditions), the surface nucleation rates are high. If this is due to high sulfuric acid or SO₂ in SE Asia, then why the upper troposphere in both regions is dominated by pure biogenic nucleation?

3. As mentioned previously, the paper did not provide convincing evidence that H₂SO₄-NH₃-HNO₃ nucleation is important in the upper troposphere over China and India (and if that is true, then why not over SE Asia, which is more or less similarly polluted?). The H₂SO₄-NH₃-HNO₃ nucleation process requires high concentrations of NH₃ and HNO₃ and low temperatures all simultaneously, and such a condition is unlikely to exist in the real atmosphere. This is the limitation of laboratory simulation experiments, which are often conducted under conditions that do not mimic the real atmospheric conditions.

4. Similarly, pure HIO₃ nucleation also requires high concentrations and very low temperatures together, so this process will unlikely occur over the open ocean from the tropics, sub-tropics to low latitudes, in an expansive global/regional scale.

5. Studies in Chinese mega-cities (including references cited in this paper) have shown that DMA alone (with sulfuric acid) cannot explain the measured new particle formation observed in Beijing using the current CLOUD nucleation algorithms, because DMA concentrations were too low in Beijing. So how did the model produce nucleation in the entire East China with a much lower DMA that should be expected than in Beijing, and how did it reproduce aerosol size distributions measured in Beijing?

The paper states that previous modeling studies underestimated nucleation rates. This paper overestimates nucleation rates due to excessive precursors. For example, DMA concentrations in East China are overestimated (assuming DMA is equivalent to 10% of ammonia, based on coastal site measurements) – which is very unlikely in most areas. The same overestimation for HNO₃ in the upper troposphere, HIO₃ over open ocean, and ELVOC and ULVOC in the upper atmosphere, as stated earlier.

It is not clear how a large fraction of CCN are produced from nucleated particles, for example, only from HIO₃. For the 2-nm particles to grow to become CCN, it requires high growth rates, and thus high concentrations of precursors are needed to condense on newly formed particles to overcome the Kelvin effects. In Beijing, sulfuric acid and DMA were not sufficient to explain the measured nucleation rate and growth rate of new particles as stated previously. The modeling study incorporated current nucleation processes based on the CLOUD experiments, but to infer nucleation to CCN at the global scale, the model also needs growth rates and condensable species are needed. So, the connection between nucleation rate and CCN (hence climate) may be present in the paper yet is not strong and clear.

Author Rebuttals to Second Revision:

Referee #1 (Remarks to the Author):

I am fully satisfied with the authors' response to the comments of referee #3, and my comments on referee #3's comments. I strongly feel that the current manuscript merits publication.

We thank the referee for supporting the publication of our manuscript and greatly appreciate the considerable amount of effort expended by the referee in reviewing the manuscript.

Referee #3 (Remarks to the Author):

The second revision is still not convincing with the conclusion (for the reasons stated below in detail), and it raises many more questions. A recent perspective article by Kirkby et al. (2023) in Nature Geoscience summarized the key findings from CLOUD chamber experiments and gave “qualitative” predictions of NPF processes in different regions of the atmosphere. It is assuring to know how different nucleation processes can be applied to or can take place in different regions under different conditions. However, the present study does not show anything newer than the Kirkby et al. (not cited in the manuscript) in a scientific context and does not have a novelty for publication in Nature. The conclusion feels like an overstatement for the reasons stated below.

This manuscript stresses multiple times that sensitivity analysis does not change their conclusion, and in fact, the figures (Extended Figures 5-8 and Supplementary Figures 3-5) show very little changes during the sensitivity analysis. This is not surprising and it is expected because the paper chose to use extremely narrow ranges of precursor concentrations (all within the same order), while their concentrations can vary over many orders of magnitude in the atmosphere. The paper states that SO₂ emission inventory has a maximum less than 50% uncertainty, this is a very optimistic view even for SO₂ in global models. For other species, such as DMA, monoterpene and HIO₃ over open ocean, and HNO₃ in the upper troposphere, their atmospheric concentrations can vary by orders of magnitude. Therefore, the sensitivity analysis (conducted by varying a single precursor concentrations with a factor of 3-5) does not provide compelling confidence in the conclusion. Considering nucleation is a non-linear process, sensitivity analysis should produce huge differences in the outcome, and the reason it shows consistent results is because of the unreasonably narrow range of concentrations tested. Therefore, the sensitivity analysis is not vigorous or statistically useful.

We thank the referee for his/her additional comments on our manuscript. We have provided point-by-point responses to those comments below and have revised the manuscript as needed. In the following, the referee's comments are presented in blue text, our responses are written in black text, and quotations from our manuscript are presented in italic type.

We fully acknowledge the value of the perspective article by Kirkby et al.¹ and have cited it in the revised manuscript (Page 25 Line 8–11). Kirkby et al.¹ reviewed the current understanding of atmospheric new particle formation (NPF) derived from laboratory measurements at the CLOUD chamber. However, there is a large gap between laboratory studies of NPF processes and the mechanisms/impacts of NPF in various regions of the world, partly because of the enormous variations in precursor concentrations worldwide and the sparse measurements of these concentrations. Whilst we may infer NPF mechanisms at specific sites based on laboratory experiments and available precursor measurements, modeling remains the only viable method to study the regional/global mechanisms of NPF and quantify the role of NPF in climate change. Actually, Kirkby et al.¹ repeatedly highlighted the current underrepresentation of NPF in global climate models and the great need for mechanistic parameterizations of NPF in these models. Our study, by developing a comprehensive model of NPF processes and the complex chemical transformation of precursor gases, represented a major advance in the model simulation of NPF and a meaningful step toward accurate climate change assessment. Contrary to the referee's conjecture regarding the novelty of our work, our findings provide important new insights that are difficult to infer from existing knowledge, including those derived from laboratory experiments. For example, our results reveal that iodine oxoacid nucleation dominates NPF in the vast oceanic boundary layers, which cannot be inferred from previous studies. As another example, we show that amine-H₂SO₄ nucleation dominates across the urban and rural boundary layers of various populous regions, including Eastern China, the United States, Europe, India, and South Africa, whereas previous studies only suggested the role of amine-H₂SO₄ nucleation at a couple of the most polluted city-center sites. More importantly, we report the exciting revelation that the dominant NPF mechanisms are essentially unique in all parts of the world, and vary dramatically with region and altitude, a finding that would not have been elucidated by laboratory experiments alone. Hopefully, the major advances in modeling and the new scientific findings justify the publication of this manuscript in *Nature*.

Regarding the referee's comments on the uncertainty of precursor emissions/concentrations, we respectfully suggest that the referee might have confused the range of atmospheric concentrations with the range of uncertainty. We agree with the referee that the concentrations of many precursors can vary over several orders of magnitude in the atmosphere; capturing such variations is actually a strength of our model. The range of uncertainty, however, indicates the degree to which the estimates of emissions/concentrations might deviate from true values. We quantified the uncertainties of emissions/concentrations with well-established methods by either examining the model-measurement concentration differences at a sufficient number of sites or by synthesizing specialized studies on emission uncertainties, which should be able to capture the range of uncertainty currently recognized. Moreover, it should be noted that the analysis in our manuscript focuses on NPF mechanisms over various spatial regions (e.g., rainforests, anthropogenically polluted regions, and oceans), not at individual model grids; this reduces the impact of the uncertainty in emissions/concentrations associated with their spatial variability. In summary, we consider that our sensitivity analysis provides reasonable estimates of the impact of the uncertainties in precursor emissions/concentrations presently recognized.

In more detail:

1. It is not convincing that pure organic nucleation is dominant over SE Asia and the Amazon in the upper troposphere. Because sulfuric acid is so dominant in nucleation, even slight SO₂ (convected from the surface sources or long-range transported from polluted regions) can overwhelm pure biogenic nucleation. So, under present-day conditions, pure biogenic nucleation is unlikely to take place in the real atmosphere. In fact, until now, pure biogenic new particle formation has not been observed in the real atmosphere (with extremely few exceptions, only one or two cases for an extremely short period).

Two recent modeling studies of the Amazon^{2,3} consistently showed that the H₂SO₄ concentration is far too low to explain the observed particle numbers in the upper troposphere and pure-organic nucleation is the dominant local nucleation mechanism. Our global study confirmed their findings regarding the upper troposphere of the Amazon, and further revealed that pure-organic nucleation also dominates in the upper troposphere of Southeastern Asia and Central Africa.

To directly observe pure-organic nucleation in the atmosphere, the detection of molecular clusters using state-of-the-art techniques such as chemical ionization mass spectrometry is needed. However, the detection of molecular clusters has only been achieved in limited regions and, to the best of our knowledge, has not been realized in the upper troposphere. This explains why pure-organic nucleation, which is greatly enhanced at low temperatures, has seldom been observed. To directly test our findings regarding NPF mechanisms in the upper troposphere of rainforests, local molecular cluster detection should be performed. We have incorporated in our manuscript specific recommendation for direct detection of molecular clusters in various regions of the world, especially in the upper troposphere. This is an example of how our modeling study could help guide more targeted observational efforts in the future.

“Furthermore, direct detection of molecular clusters is encouraged in various regions of the world, especially in the upper troposphere.” (Page 14 Line 11–13)

2. It is not clear why in the Amazon, the surface nucleation rate is so low, with abundant monoterpene and high oxidants, whereas in SE Asia (similar conditions), the surface nucleation rates are high. If this is due to high sulfuric acid or SO₂ in SE Asia, then why the upper troposphere in both regions is dominated by pure biogenic nucleation?

In the surface layer, the low NPF rate in the Amazon is attributable to the low SO₂/H₂SO₄ and high temperatures that cause low rates of organic nucleation; this is consistent with observations showing rare NPF events in the pristine Amazon boundary layer^{4,5}. The high surface NPF rate in Southeastern Asia is driven by iodine oxoacids nucleation (typical of oceanic regions) and amine–H₂SO₄ nucleation (typical of anthropogenically polluted regions), as shown in Fig. 1B. This is because Southeastern Asia is affected by not only biogenic emissions but also oceanic emissions of iodine species and anthropogenic emissions of SO₂ and amines; thus, it possesses some NPF features typical of oceanic and polluted regions.

In the upper troposphere of the Amazon and Southeastern Asia, pure-organic nucleation is the dominant NPF mechanism because its rate is greatly enhanced at low temperatures due to the dramatic decrease of volatility and increase of cluster stability. Nucleation of H₂SO₄ and NH₃

contributes <2% of the NPF rate in the upper troposphere of the Amazon, but contributes a larger fraction of 10%–15% in the upper troposphere of Southeastern Asia owing to more abundant H_2SO_4 , as suggested by the referee. However, according to our model, this mechanism still cannot compete with organic-mediated nucleation; this conclusion remains unchanged under all sensitivity simulations perturbing precursor emissions/concentrations or NPF parameterizations in this study.

We have included the following description regarding the differences among rainforest regions:

“Figure 1 shows that, near the surface of rainforests, the NPF rates are low in the Amazon, but are high in Southeastern Asia and moderate in Central Africa. The low NPF rate in the Amazon is due to low $\text{SO}_2/\text{H}_2\text{SO}_4$ and high temperatures that cause low organic nucleation rates; this is consistent with observations showing rare NPF events in the pristine Amazon boundary layer^{4,5}. The high surface NPF rate in Southeastern Asia is driven by iodine oxoacids nucleation (typical of oceanic regions) and amine– H_2SO_4 nucleation (typical of anthropogenically polluted regions), as shown in Fig. 1B. This is because Southeastern Asia is affected by not only biogenic emissions but also oceanic emissions of iodine species and anthropogenic emissions of SO_2 and amines, and thus it possesses some NPF features typical of oceanic and polluted regions. Central Africa is more affected by anthropogenic and oceanic emissions than the Amazon but is less affected than Southeastern Asia, leading to the moderate NPF rate there.

In the upper troposphere of all three rainforest regions, pure-organic nucleation is the dominant NPF mechanism because its rate is greatly enhanced at low temperatures due to the dramatic decrease of volatility and increase of cluster stability. It is noted that, nucleation of H_2SO_4 and NH_3 contributes <2% of the NPF rate in the upper troposphere of the Amazon, but contributes a larger fraction of 10%–15% in the upper troposphere of Southeastern Asia owing to more abundant H_2SO_4 from anthropogenic sources. However, according to our model, this mechanism still cannot compete with organic-mediated nucleation.” (Page 49 Line 6–24)

3. As mentioned previously, the paper did not provide convincing evidence that $\text{H}_2\text{SO}_4\text{-NH}_3\text{-HNO}_3$ nucleation is important in the upper troposphere over China and India (and if that is true, then why not over SE Asia, which is more or less similarly polluted?). The $\text{H}_2\text{SO}_4\text{-NH}_3\text{-HNO}_3$ nucleation process requires high concentrations of NH_3 and HNO_3 and low temperatures all simultaneously, and such a condition is unlikely to exist in the real atmosphere. This is the limitation of laboratory simulation experiments, which are often conducted under conditions that do not mimic the real atmospheric conditions.

$\text{H}_2\text{SO}_4\text{-HNO}_3\text{-NH}_3$ nucleation is important in the upper troposphere over the Asian monsoon region (spanning China and India) but not over Southeastern Asia mainly because of the much larger NH_3 concentration in the former than the latter, given that $\text{H}_2\text{SO}_4\text{-HNO}_3\text{-NH}_3$ nucleation rate is highly sensitive to NH_3 concentration. Observations (Figs. 4 and 5 of Höpfner et al.⁶ and Supplementary Fig. 5 of Höpfner et al.⁷) and our simulation (Extended Data Fig. 4) both revealed striking hotspots of NH_3 concentrations (10–40 ppt, occasionally >60 ppt) in the upper troposphere of the Asian monsoon region in summer, but not over other regions or during different seasons. This is because of the following: 1) NH_3 emissions in China and India (23.0 Mt in 2014⁸) are much larger than those in Southeastern Asia (4.7 Mt in 2014⁸), and 2) the Asian summer monsoon is especially favorable for

upward transport of NH_3 to the heights of interest. For similar reasons, the mean H_2SO_4 concentrations in the upper troposphere of the Asian monsoon region (0.2–0.5 ppt) are also larger than those over Southeastern Asia (0.04–0.1 ppt), as shown in Supplementary Figs. 2–3, further contributing to the high H_2SO_4 – HNO_3 – NH_3 nucleation rate over the Asian monsoon region.

According to observations and our simulations, high concentrations of NH_3 (10–40 ppt, occasionally >60 ppt) and HNO_3 (0.15–1.2 ppb) and low temperatures (205–230 K) do coexist in the upper troposphere of the Asian monsoon region, with good agreement between the simulated and observed NH_3 and HNO_3 concentrations (see Extended Data Fig. 4, Supplementary Fig. 10, and related descriptions).

The parameterization of the H_2SO_4 – HNO_3 – NH_3 nucleation rate was derived from CLOUD chamber experiments reported in a previous *Nature* paper by Wang et al.⁹ The experiments were conducted at 223 K with H_2SO_4 , NH_3 , and HNO_3 concentrations of 0.05–0.5 ppt, 19–80 ppt, and 0.03–0.2 ppb, respectively, i.e., all similar to real-world conditions in the upper troposphere of the Asian monsoon region. Therefore, the laboratory experiments closely mimic the real atmospheric conditions in this case.

We have added the following descriptions and discussion to the revised manuscript:

“Our results show that H_2SO_4 – HNO_3 – NH_3 nucleation is important in the upper troposphere over the Asian monsoon region (spanning China and India, see Fig. 2B) but not over other regions with notable anthropogenic pollution such as Southeastern Asia. This is mainly because of the much larger NH_3 concentration in the former than the latter. Observations (Figs. 4 and 5 of Höpfner et al.⁶ and Supplementary Fig. 5 of Höpfner et al.⁷) and our simulation (Extended Data Fig. 4) both revealed striking hotspots of NH_3 concentrations (10–40 ppt, occasionally >60 ppt) in the upper troposphere of the Asian monsoon region in summer, but not over other regions or during different seasons. This is because of the following: 1) NH_3 emissions in China and India (23.0 Mt in 2014⁸) are much larger than those in Southeastern Asia (4.7 Mt in 2014⁸), and 2) the Asian summer monsoon is especially favorable for upward transport of NH_3 to the heights of interest. For similar reasons, the mean H_2SO_4 concentrations in the upper troposphere of the Asian monsoon region (0.2–0.5 ppt) are also larger than those over Southeastern Asia (0.04–0.1 ppt), as shown in Supplementary Figs. 2–3, further contributing to the high H_2SO_4 – HNO_3 – NH_3 nucleation rate over the Asian monsoon region.” (Page 50 Line 1–14)

“For the synergistic H_2SO_4 – HNO_3 – NH_3 mechanism, Wang et al.⁹ conducted experiments in the CLOUD chamber at 223 K, which is a temperature typical of the upper troposphere in the Asian monsoon region (205–230 K according to our model). They used H_2SO_4 , NH_3 , and HNO_3 concentrations of 0.05–0.5 ppt, 19–80 ppt, and 0.03–0.2 ppb, respectively, i.e., all similar to real-world conditions in the upper troposphere of the Asian monsoon region according to observations and our model simulations (Extended Data Fig. 4, Supplementary Figs. 2–3, and Supplementary Fig. 10).” (Page 27 Line 21 to Page 28 Line 3)

4. Similarly, pure HIO₃ nucleation also requires high concentrations and very low temperatures together, so this process will unlikely occur over the open ocean from the tropics, sub-tropics to low latitudes, in an expansive global/regional scale.

Recent CLOUD chamber experiments showed that, at +10 °C, iodine oxoacids ion-induced nucleation proceeds at high rates of 0.1–200 cm⁻³ s⁻¹ under atmospherically relevant HIO₃ concentrations¹⁰. Therefore, low temperatures are not prerequisite for iodine oxoacids nucleation. Our parameterizations of iodine oxoacids nucleation were primarily based on these experimental data.

Moreover, HIO₃ concentrations simulated by our model vary between 80% below and 100% above the observed values at 10 oceanic or coastal sites worldwide (Extended Data Fig. 3C), which is deemed reasonable performance. The combination of experiment-based parameterizations and reasonable concentration simulations likely results in reasonable estimates of NPF rates. Our sensitivity simulations that perturbed domain-wide HIO₃ concentrations within the largest range of the simulation biases did not change our conclusion regarding the role of iodine oxoacids nucleation, increasing the credibility of our results.

5. Studies in Chinese mega-cities (including references cited in this paper) have shown that DMA alone (with sulfuric acid) cannot explain the measured new particle formation observed in Beijing using the current CLOUD nucleation algorithms, because DMA concentrations were too low in Beijing. So how did the model produce nucleation in the entire East China with a much lower DMA that should be expected than in Beijing, and how did it reproduce aerosol size distributions measured in Beijing?

Most recent studies concluded that the nucleation of H₂SO₄ with amines, with dimethylamine (DMA) being the key species and a proxy for all amines, is the dominant nucleation mechanism in Chinese megacities such as Beijing¹¹⁻¹³. Two studies, as cited in our paper and mentioned by the referee, suggested the potential role of other precursors such as NH₃ and HNO₃^{14,15}. Specifically, Yin et al.¹⁴ characterized molecular clusters during NPF periods in urban Beijing, and showed that clusters with three or fewer H₂SO₄ molecules are almost exclusively stabilized by amines, whereas NH₃ helps stabilize larger clusters with four or more H₂SO₄ molecules. This is actually consistent with the notion that amine–H₂SO₄ nucleation dominates nucleation in Beijing; a subsequent paper by the same group stated the dominant role of DMA and H₂SO₄ in nucleation more clearly¹². Liu et al.¹⁵ showed that HNO₃ might enhance DMA–H₂SO₄ nucleation in certain megacities under favorable conditions with relatively high HNO₃ and DMA concentrations. However, the quantum chemistry calculation methods used in that study were rather uncertain, and the DMA concentrations used in the calculations were approximately one order of magnitude larger than values observed in Beijing. Moreover, HNO₃-containing clusters have not been detected in previous field observations conducted in megacities, leaving the results of Liu et al.¹⁵ unconfirmed by measurements. For these reasons, we did not consider the enhancement of amine–H₂SO₄ nucleation by HNO₃ in our simulation, but we acknowledged its potential role in the Discussion section (Page 13 Line 19–20).

Regarding the simulation results over the entire region of Eastern China, we agree that the amine–H₂SO₄ nucleation rates in many rural areas are lower than those in Beijing, but other competing mechanisms such as H₂SO₄–NH₃–H₂O nucleation are also lower in those rural areas because H₂SO₄

concentrations are lower and NH_3 largely co-varies with amines owing to their similar emission sources. Therefore, our model shows that amine- H_2SO_4 nucleation dominates the average NPF rate near the surface of Eastern China, although H_2SO_4 - NH_3 - H_2O nucleation also contributes a few percent of the total NPF rate.

The paper states that previous modeling studies underestimated nucleation rates. This paper overestimates nucleation rates due to excessive precursors. For example, DMA concentrations in East China are overestimated (assuming DMA is equivalent to 10% of ammonia, based on coastal site measurements) – which is very unlikely in most areas. The same overestimation for HNO_3 in the upper troposphere, HIO_3 over open ocean, and ELVOC and ULVOC in the upper atmosphere, as stated earlier.

We respectfully suggest that this criticism lacks solid foundation based on evidence. Our model evaluation against available observations showed that simulated concentrations of DMA, HNO_3 , HIO_3 , and monoterpenes (precursors to ELVOC and ULVOC) agree reasonably well with observations (Extended Data Fig. 3, Supplementary Fig. 10, and Page 7 Line 18–21). Our sensitivity simulations further showed that the differences between simulated and observed precursor concentrations are unlikely to change our main findings about the dominant NPF mechanisms (Extended Data Figs. 5–8).

We did not assume that DMA is equivalent to 10% of NH_3 , nor did we state this in the manuscript. Instead, we used source-specific DMA/ NH_3 emission ratios of 0.0070, 0.0018, 0.0015, 0.0100, 0.0009, and 0.0144 for chemical-industrial, other industrial, agricultural, residential, transportation, and maritime sources, respectively, based on source apportionment analysis using terrestrial observations of amines and NH_3 ¹⁶ as well as oceanic measurements of amines and NH_3 ¹⁷ (Page 32 Line 6–12). All these ratios are much lower than the ratio of 10% mentioned by the referee.

It is not clear how a large fraction of CCN are produced from nucleated particles, for example, only from HIO_3 . For the 2-nm particles to grow to become CCN, it requires high growth rates, and thus high concentrations of precursors are needed to condense on newly formed particles to overcome the Kelvin effects. In Beijing, sulfuric acid and DMA were not sufficient to explain the measured nucleation rate and growth rate of new particles as stated previously. The modeling study incorporated current nucleation processes based on the CLOUD experiments, but to infer nucleation to CCN at the global scale, the model also needs growth rates and condensable species are needed. So, the connection between nucleation rate and CCN (hence climate) may be present in the paper yet is not strong and clear.

To account for the particle growth due to condensation, we dynamically simulated the condensation of H_2SO_4 and, more importantly, organic vapors across the entire volatility range, with the latter being a major contributor to the growth of newly formed particles to CCN size. To address the referee's concern, we have included a more detailed description of the particle growth treatment in the revised manuscript:

“Condensation is a key process driving the growth of newly formed particles to CCN size. The model explicitly represents the condensation of H_2SO_4 and organic vapors across the entire volatility range

(including ULVOC and ELVOC). The condensation of H_2SO_4 is treated dynamically as an irreversible process, using standard mass transfer expressions that are integrated over the size distribution of each mode¹⁸. The condensation and evaporation of organic vapors are treated dynamically as reversible processes and calculated using a semi-implicit Euler approach with adaptive time stepping based on Zaveri et al.¹⁹. The Kelvin effect is accounted for in the calculation of condensation rates. Condensation can result in smaller-mode particles growing into the size range of the next larger mode. Thus, after condensation is calculated, the renaming module reallocates the number and mass concentrations of the subset of smaller-mode particles that exceed a specified threshold diameter to the next larger mode. The threshold diameter is defined as the geometric mean of the characteristic diameters of two neighboring modes, where the characteristic diameter of a mode was assumed to be the nominal volume mean diameter (determined by the nominal number median diameter and the geometric standard deviation) for that mode^{20,21}." (Page 34 Line 13 to Page 35 Line 3)

References

- Kirkby, J. *et al.* Atmospheric new particle formation from the CERN CLOUD experiment. *Nat. Geosci.* **16**, 948-957, doi:10.1038/s41561-023-01305-0 (2023).
- Zhao, B. *et al.* High concentration of ultrafine particles in the Amazon free troposphere produced by organic new particle formation. *P. Natl. Acad. Sci. USA.* **117**, 25344-25351, doi:10.1073/pnas.2006716117 (2020).
- Wang, X., Gordon, H., Grosvenor, D. P., Andreae, M. O. & Carslaw, K. S. Contribution of regional aerosol nucleation to low-level CCN in an Amazonian deep convective environment: results from a regionally nested global model. *Atmos. Chem. Phys.* **23**, 4431-4461, doi:10.5194/acp-23-4431-2023 (2023).
- Wimmer, D. *et al.* Ground-based observation of clusters and nucleation-mode particles in the Amazon. *Atmos. Chem. Phys.* **18**, 13245-13264, doi:10.5194/acp-18-13245-2018 (2018).
- Franco, M. A. *et al.* Occurrence and growth of sub-50nm aerosol particles in the Amazonian boundary layer. *Atmos. Chem. Phys.* **22**, 3469-3492, doi:DOI 10.5194/acp-22-3469-2022 (2022).
- Hoepfner, M. *et al.* First detection of ammonia (NH₃) in the Asian summer monsoon upper troposphere. *Atmos. Chem. Phys.* **16**, 14357-14369, doi:10.5194/acp-16-14357-2016 (2016).
- Hoepfner, M. *et al.* Ammonium nitrate particles formed in upper troposphere from ground ammonia sources during Asian monsoons. *Nat. Geosci.* **12**, 608-612, doi:10.1038/s41561-019-0385-8 (2019).
- Hoesly, R. M. *et al.* Historical (1750-2014) anthropogenic emissions of reactive gases and aerosols from the Community Emissions Data System (CEDS). *Geosci. Model. Dev.* **11**, 369-408, doi:10.5194/gmd-11-369-2018 (2018).
- Wang, M. *et al.* Synergistic HNO₃-H₂SO₄-NH₃ upper tropospheric particle formation. *Nature* **605**, 483-489, doi:10.1038/s41586-022-04605-4 (2022).
- He, X.-C. *et al.* Role of iodine oxoacids in atmospheric aerosol nucleation. *Science* **371**, 589-595, doi:10.1126/science.abe0298 (2021).
- Cai, R. *et al.* Sulfuric acid-amine nucleation in urban Beijing. *Atmos. Chem. Phys.* **21**, 2457-2468, doi:10.5194/acp-21-2457-2021 (2021).
- Cai, R. *et al.* The missing base molecules in atmospheric acid-base nucleation. *Natl. Sci. Rev.* **9**, nwac137, doi:10.1093/nsr/nwac137 (2022).
- Yao, L. *et al.* Atmospheric new particle formation from sulfuric acid and amines in a Chinese megacity. *Science* **361**, 278-281, doi:10.1126/science.aao4839 (2018).
- Yin, R. *et al.* Acid-Base Clusters during Atmospheric New Particle Formation in Urban Beijing. *Environ. Sci. Technol.* **55**, 10994-11005, doi:10.1021/acs.est.1c02701 (2021).

- Liu, L. *et al.* Rapid sulfuric acid-dimethylamine nucleation enhanced by nitric acid in polluted regions. *P. Natl. Acad. Sci. USA*. **118**, e2108384118, doi:10.1073/pnas.2108384118|1of6 (2021).
- Mao, J. *et al.* High-resolution modeling of gaseous methylamines over a polluted region in China: source-dependent emissions and implications of spatial variations. *Atmos. Chem. Phys.* **18**, 7933-7950, doi:10.5194/acp-18-7933-2018 (2018).
- Chen, D. *et al.* Mapping gaseous dimethylamine, trimethylamine, ammonia, and their particulate counterparts in marine atmospheres of China's marginal seas - Part 1: Differentiating marine emission from continental transport. *Atmos. Chem. Phys.* **21**, 16413-16425, doi:10.5194/acp-21-16413-2021 (2021).
- Liu, X. *et al.* Toward a minimal representation of aerosols in climate models: description and evaluation in the Community Atmosphere Model CAM5. *Geosci. Model. Dev.* **5**, 709-739, doi:10.5194/gmd-5-709-2012 (2012).
- Zaveri, R. A., Easter, R. C., Fast, J. D. & Peters, L. K. Model for Simulating Aerosol Interactions and Chemistry (MOSAIC). *J. Geophys. Res-Atmos.* **113**, D13204, doi:10.1029/2007jd008782 (2008).
- Zhang, K., Sun, J. & Ma, P. L. Modal Aerosol Module with nucleation mode (MAM5) in E3SM version 1: model description and evaluation. *under review* (2023).
- Liu, X. *et al.* Description and evaluation of a new four-mode version of the Modal Aerosol Module (MAM4) within version 5.3 of the Community Atmosphere Model. *Geosci. Model. Dev.* **9**, 505-522, doi:10.5194/gmd-9-505-2016 (2016).